# Inhibitory PD-1 axis maintains high-avidity stem-like CD8⁺ T cells

Jyh Liang Hor[1✉], Edward C. Schrom[1], Abigail Wong-Rolle[1,2], Luke Vistain[1], Wanjing Shang[1], Qiang Dong[3], Chen Zhao[1,2], Chengcheng Jin[3] & Ronald N. Germain[1,4✉]

Stem-like progenitors are self-renewing cytotoxic T cells that expand as effector cells during successful checkpoint immunotherapy[1,2]. Emerging evidence suggests that tumour-draining lymph nodes support the continuous generation of these stem-like cells that replenish tumour sites and are a key source of expanded effector populations[3–6], underlining the importance of understanding what factors promote and maintain activated T cells in the stem-like state. Here, using advanced three-dimensional multiplex immunofluorescence imaging, we identify antigen-presentation niches in tumour-draining lymph nodes that support the expansion, maintenance and affinity evolution of TCF-1⁺PD-1⁺SLAMF6ʰⁱᵍʰ stem-like CD8⁺ T cells. Contrary to the prevailing view that persistent T cell receptor (TCR) signalling drives terminal effector differentiation, prolonged antigen engagement days beyond initial priming sustains the proliferation and self-renewal of these stem-like T cells in vivo. The inhibitory PD-1 pathway has a central role in this process through fine-tuning the TCR signal input that enables the selective expansion of high-affinity TCR stem-like clones as a renewable source of effector cells. PD-1 blockade disrupts this tuning, leading to terminal differentiation or death of the most avid anti-tumour stem-like cells. Our results therefore reveal a relationship between TCR ligand affinity recognition, a key negative-feedback regulatory loop and T cell stemness programming. Furthermore, these findings raise questions about whether anti-PD-1 blockade during cancer immunotherapy provides a short-term anti-tumour effect at the cost of diminishing efficacy due to progressive loss of these critical high-affinity precursors.

Inhibitory molecules acting in *cis* (within a T cell) and *trans* (between cells) are central to the complex regulatory circuit that fine-tunes the functional output of TCR signalling. PD-1 and its ligands, PD-L1 and PD-L2, represent a key *trans*-acting inhibitory pathway that attenuates lymphocyte activation, restrains effector differentiation, and limits pathology in autoimmune diseases and chronic infections[7]. Checkpoint inhibitory pathways have demonstrated their capacity to promote anti-tumour T cell responses[8], but only a subset of patients respond favourably to checkpoint immunotherapy and many patients eventually experience acquired resistance to treatment and tumour relapse[9].

Stem-like progenitor CD8⁺ T ($T_{SL}$) cells represent a subset of activated cytotoxic T lymphocytes that retains high proliferative potential and self-renewal capacity[10]. This subset of cells or their immediate progeny is primarily responsible for the proliferative burst that gives rise to a large pool of functional effector cells during anti-PD-1 checkpoint immunotherapy in chronic viral infections and tumour-bearing hosts[2,11]. Emerging evidence suggests that tumour-draining lymph nodes (tdLNs) serve as reservoirs from which freshly generated $T_{SL}$-like cells or their progeny continuously replenish the tumour microenvironment (TME)

and provide a source of functionally competent effector cells during checkpoint therapy[3–6].

Questions remain unresolved of how activated T cells remain in a stem-like state in lymphoid tissues during ongoing antigen stimulation and inflammation, especially under persistent antigen encounter in the case of tumours, and whether T cell fate is imprinted during early priming events or continues to be dynamically shaped by diverse inputs present in the lymph node microenvironment. Here we combined a multiplex three-dimensional (3D) volumetric imaging method with conventional immunological assays to approach these questions.

## PD-1⁺SLAMF6⁺ $T_{SL}$ cells in cDC1-cell-rich niches

To map the spatial distribution and functional states of tumour antigen-specific CD8⁺ T cells in the tdLN, we performed 3D tissue imaging[12] of optically cleared day 8 tdLN slices (around 300 μm thickness, comprising about 25–30% of the whole tdLN volume) derived from mice adoptively transferred with a physiological frequency of naive OT-I precursors (500 cells)[13] and intradermally implanted with an ovalbumin-expressing *Kras*^G12D/+*Trp53*^−/− lung adenocarcinoma line[14]

[1]Lymphocyte Biology Section, Laboratory of Immune System Biology, NIAID, NIH, Bethesda, MD, USA. [2]Thoracic and GI Malignancies Branch, Center for Cancer Research, NCI, NIH, USA. [3]Perelman School of Medicine, University of Pennsylvania, Philadelphia, PA, USA. [4]Center for Advanced Tissue Imaging, NIAID/NCI, NIH, Bethesda, MD, USA. ✉e-mail: jyhliang.hor@nih.gov; rgermain@niaid.nih.gov

(KP-OVA). Fluorescent XCR1 reporter mice were used as recipients[15]. We developed and optimized a 3D tissue staining protocol together with a distributed computational pipeline that enables efficient single-cell-level analysis of large 3D volume imaging datasets comprising >1 million cells of interest (Extended Data Fig. 1a and Supplementary Video 1). This provides a much more complete view of the cell populations in the tdLN than single-cell studies that often only examine a fraction of this number of cells, while also adding high-resolution spatial information.

Naive antigen-specific T cells undergo intense antigen-driven proliferation and functional diversification after the first 24–36 h of initial priming[16]. The day 8 tdLN therefore represents a substantially late phase of expansion and differentiation distinct from both the initial T cell activation or the terminal exhaustion phase that occurs many days or weeks later in tumour-bearing or chronically infected hosts. By day 8, the antigen-activated OT-I population extensively infiltrated the lymph node T cell zone, interfollicular regions and medullary sinuses (Fig. 1a). Using a combination of labelled antibodies to visualize stem-like markers (TCF-1 and SLAMF6), activation state-associated proteins (BATF), a key inhibitory molecule (PD-1) and a feature of proliferating cells (Ki-67), we mapped the spatial distribution of expanded OT-I cells to their respective functional states (Fig. 1b–e). A small subset of OT-I cells formed distinct clusters in close association with XCR1+ type 1 conventional dendritic (cDC1) cells in the T cell zone. These clustering T cells expressed high levels of stem-like (TCF-1 and SLAMF6) and activation (PD-1 and BATF) markers (Fig. 1b).

T cell–dendritic cell clustering is a classic feature of early T cell priming[16]. As naive T cells also express high amounts of TCF-1, we examined whether these clustering TCF-1+ cells belonged to a late infiltrating wave of naive OT-I cells that had only recently experienced initial antigen exposure. Using proliferation dye staining and either flow cytometry or imaging, we failed to detect residual dye signal among the adoptively transferred antigen-specific OT-I cells in the tdLN of tumour-bearing mice (Extended Data Fig. 1b–f), indicating that these clustering TCF-1+PD-1+SLAMF6+ cells at day 8 represented a subset of activated and extensively divided OT-I cells.

We then performed subset classification based on TCF-1 and PD-1 expression (Fig. 1c,d, Extended Data Fig. 2a and Supplementary Video 2), confirming that the TCF-1+PD-1+ subset is enriched with cells also showing high SLAMF6 and BATF expression and positioned in close proximity to dense regions of cDC1 (Fig. 1e–g and Extended Data Fig. 2b,c and Supplementary Video 3). We defined TCF-1+ cells as $T_{SL}$ cells and further delineated this population into two separate subsets: a PD-1$^{high}$SLAMF6$^{high}$ subset (SLAMF6+ $T_{SL}$) and a PD-1$^{int}$SLAMF6$^{int}$ $T_{SL}$ subset (SLAMF6− $T_{SL}$). A previous study associated the latter SLAMF6− subset with a lower proliferative capacity and lower Ki-67 staining than their SLAMF6+ $T_{SL}$ cell counterparts[4], consistent with both our imaging and flow cytometry data (Extended Data Fig. 2c–e). Spatial analysis showed that TCF-1+ $T_{SL}$ cells preferentially localized to the T cell zone whereas TCF-1− T effector ($T_{eff}$) cells concentrated in the lymph node periphery (Fig. 1d and Supplementary Video 2), consistent with previous observations[17].

## Late antigen signalling among SLAMF6+ $T_{SL}$ cells

As PD-1 upregulation is driven by TCR signalling, the pattern of elevated PD-1 (as well as BATF) expression among clustering SLAMF6+ $T_{SL}$ cells suggested ongoing antigen signalling events occurring days beyond initial antigen contact. To examine whether active TCR signalling was occurring within the niches containing SLAMF6+ $T_{SL}$ and cDC1 cells, we stained the tdLN tissues with antibodies against the transcription factor NFAT1, which translocates from the cytoplasm of T lymphocytes into the nucleus after TCR engagement and NFAT dephosphorylation[18] (Fig. 1h). We have previously shown that nuclear NFAT staining of T lymphocytes is highly specific for antigen-stimulated T cells in contact with antigen-presenting cells[19]. Nuclear NFAT staining is mostly observed among clustered OT-I $T_{SL}$ cells in close spatial proximity to cDC1 cells, whereas OT-I cells distant

from cDC1 cells largely lacked nuclear NFAT (Fig. 1h,i and Extended Data Fig. 2f–i). We also observed similar findings in the draining lymph nodes from mice immunized with OVA protein and poly(I:C) (Extended Data Fig. 2j,k), suggesting that this late antigen presentation phase is probably a fundamental characteristic of the primary adaptive immune response involving activated CD8+ T cells exhibiting a stem-like phenotype. Overall, these findings show that, days after the initiation of TCR signalling and extensive cell division, the most stem-like population of CD8+ T cells remains associated with antigen-presenting cDC1 cells and responds to such antigen display with active TCR-dependent signalling. Such results contrast with the prevailing view that prolonged antigen signalling necessarily drives T cells to an effector state.

## Preferential SLAMF6+ $T_{SL}$ cell retention in the lymph node

We next examined the spatial distribution of polyclonal activated CD8+ T cell subsets using the same 3D microscopy technique as in Fig. 1 with optically cleared tdLN tissues from day 8 tumour-bearing mice without adoptive transfer of OT-I cells. This volumetric imaging technique enabled the detection of relatively rare cellular niches that are often missed with thin tissue cryosections. Using a combination of protein markers to delineate activated Ki-67+CD8+ cells, we observed the formation of CD8+TCF-1+ $T_{SL}$ cell clusters scattered throughout the T cell zone, albeit at a lower frequency and with a smaller cluster size than in the transgenic model (Fig. 2a–c and Extended Data Fig. 3a,b). These clustering polyclonal CD8+ T cells also exhibited high PD-1 and SLAMF6 expression, similar to what was observed with OT-I. A density map of activated Ki-67+CD8+ T cells revealed enrichment of staining for PD-1 and SLAMF6 in association with cDC1 within discrete foci of the tdLN (Fig. 2a and Extended Data Fig. 3a). Similar observations were also made examining the tdLN of MC38-OVA tumour-bearing mice, although with a lower frequency of $T_{SL}$ cell clusters (Extended Data Fig. 3c). Thus, clustering SLAMF6+ $T_{SL}$ cell niches is a distinct spatial feature that occurs even with natural precursor frequencies and with different tumour and vaccine models.

These observations prompted us to examine whether the SLAMF6+ $T_{SL}$ cell subset was uniquely localized to the antigen-rich tdLN or also trafficked to other sites, including the TME. We used flow cytometry and H-2K$^b$–SIINFEKL tetramer staining to detect OVA-specific CD8+ T cells in the tumour-draining inguinal lymph nodes, non-draining contralateral brachial lymph nodes, the spleens and tumours at day 10 after tumour induction (Supplementary Fig. 1). Whereas TCF-1+ $T_{SL}$ and TCF-1− $T_{eff}$ cells were readily identified in all tissues examined, PD1$^{high}$SLAMF6$^{high}$ $T_{SL}$ cells were exclusively detected in the tdLN (Fig. 2d–g). As expected, expression of PD-1 and SLAMF6 was highly correlated among TCF-1+ $T_{SL}$ cells in the tdLN (Extended Data Fig. 4a). While SLAMF6 expression was elevated in all subsets of activated CD8+ T cells, PD1$^{high}$ $T_{SL}$ cells had the highest SLAMF6 levels among all of the subsets, with $T_{SL}$ cells in the tdLN showing a 3.5-fold greater expression than naive CD8+ T cells (Extended Data Fig. 4b,c). Notably, in contrast to in the tdLN, PD-1+ $T_{SL}$ cells in the tumour did not show high expression of SLAMF6 (Fig. 2f,g and Extended Data Fig. 4b). We found that the SLAMF6+ $T_{SL}$ cell subset also expressed higher amounts of the anti-apoptotic protein BCL-2 and the inhibitory molecule CD200—a marker associated with stem-like cells[3,20] (Extended Data Fig. 4d,e). A recent study linked histone 2A Lys119 ubiquitination (H2AK119Ub) to a stem-like epigenetic profile in CD8+ T cells[21]. Indeed, we found a strong correlation between SLAMF6 expression and H2AK119Ub level (Extended Data Fig. 4f,g). Collectively, these observations revealed a strong association of SLAMF6+ $T_{SL}$ cells uniquely localized to the tdLN with stem-like/quiescent T cell features.

High PD-1 expression outside of the tdLN was observed only in the tumour (Fig. 2d,e), in agreement with published findings[22]. $T_{eff}$ cells in both tdLN and non-tdLN had low PD-1 levels, while expressing the effector and exhaustion markers CX3CR1 and CD39 (refs. 23,24)

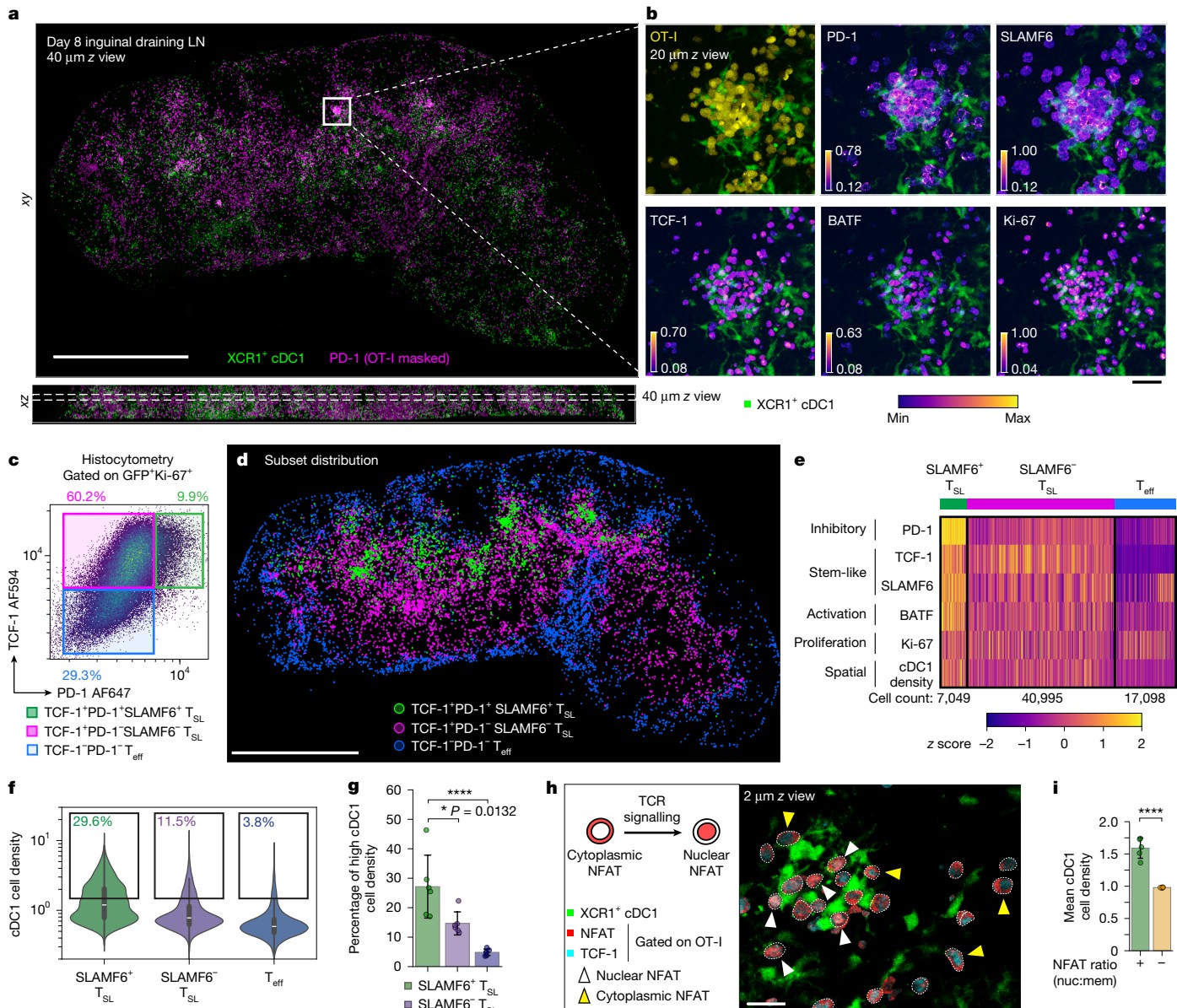

**Fig. 1 | 3D multiplexed tissue imaging reveals late antigen presentation niches in the tdLN. a**, A cross-section (thickness, 40 μm) from a 300-μm-thick optically cleared day 8 inguinal tdLN slice, showing the spatial localization of expanded OT-I cells (masked PD-1 intensity in magenta) and XCR1+ cDC1 cells (green). The white box indicates the magnified region displayed in **b**. Scale bar, 1,000 μm. **b**, Magnified image of a cDC1–T cell clustering niche with the protein markers expressed by OT-I cells displayed with a perceptually uniform colourmap, with minimum (min)/maximum (max) values scaled for each individual marker (1.0 = maximum image bit-depth). Scale bar, 30 μm. **c**, Representative histocytometry gating strategy based on TCF-1 and PD-1 expression. **d**, The spatial distribution of OT-I subsets from **a** (40 μm z thickness) based on the gating strategy in **c**. Scale bar, 1,000 μm. **e**, Normalized z scores for each parameter sorted by individual subsets. The heat map contains the entire OT-I population of 65,142 cells. **f**, Quantification of the OT-I cell subset proximity to dense cDC1 cell regions based on the cDC1 cell spatial density (the fluorescent XCR1 channel was processed with a Gaussian filter with a kernel

of $\sigma = 3.27$ μm). T cells register a higher cDC1 cell density value when positioned in denser cDC1 regions. The gates denote the cDC1 cell density values above the threshold of mean + 1 s.d. The box plots show the median (centre line), first and third quartiles (box limits) and 1.5 × interquartile range (whiskers). **g**, Subset frequency with high cDC1 cell density as gated in **f**. Data are pooled from three independent experiments. $n = 6$. **h**, NFAT1 staining masked on OT-I cells from a magnified region in day 8 tdLN, showing OT-I cells (dotted white outlines) clustering with cDC1 cells (green). The white arrowheads indicate cells with nuclear NFAT localization (red) and their co-localization with nuclear TCF-1 stain (cyan). The yellow arrowheads show examples of OT-I cells with cytoplasmic NFAT localization. Scale bar, 20 μm. **i**, Quantification of the mean cDC1 cell density value of OT-I cells with a positive (nuclear, nuc) or negative (membrane, mem) NFAT localization ratio. Data are pooled from two independent experiments. $n = 5$. Data are mean ± s.d. Statistical analysis was performed using one-way analysis of variance (ANOVA) with Tukey's multiple-comparison test (**g**) and unpaired two-tailed t-tests (**i**); *$P < 0.05$, ****$P < 0.0001$.

(Extended Data Fig. 4h–k). This is consistent with the notion that PD-1$^{high}$ T$_{SL}$ cells involved in responses to late ongoing antigen presentation were preferentially retained in the tdLN, while PD-1$^{low}$ T$_{SL}$ and T$_{eff}$ cells that had egressed from the tdLN did not encounter cognate antigen-presenting cells until arriving at the tumour site.

## Late antigen-driven T$_{SL}$ cell expansion

We next explored the functional role of late antigen presentation within the tdLN. The XCR1-DTR transgenic mouse enables selective ablation of cross-presenting cDC1 cells using diphtheria toxin administration

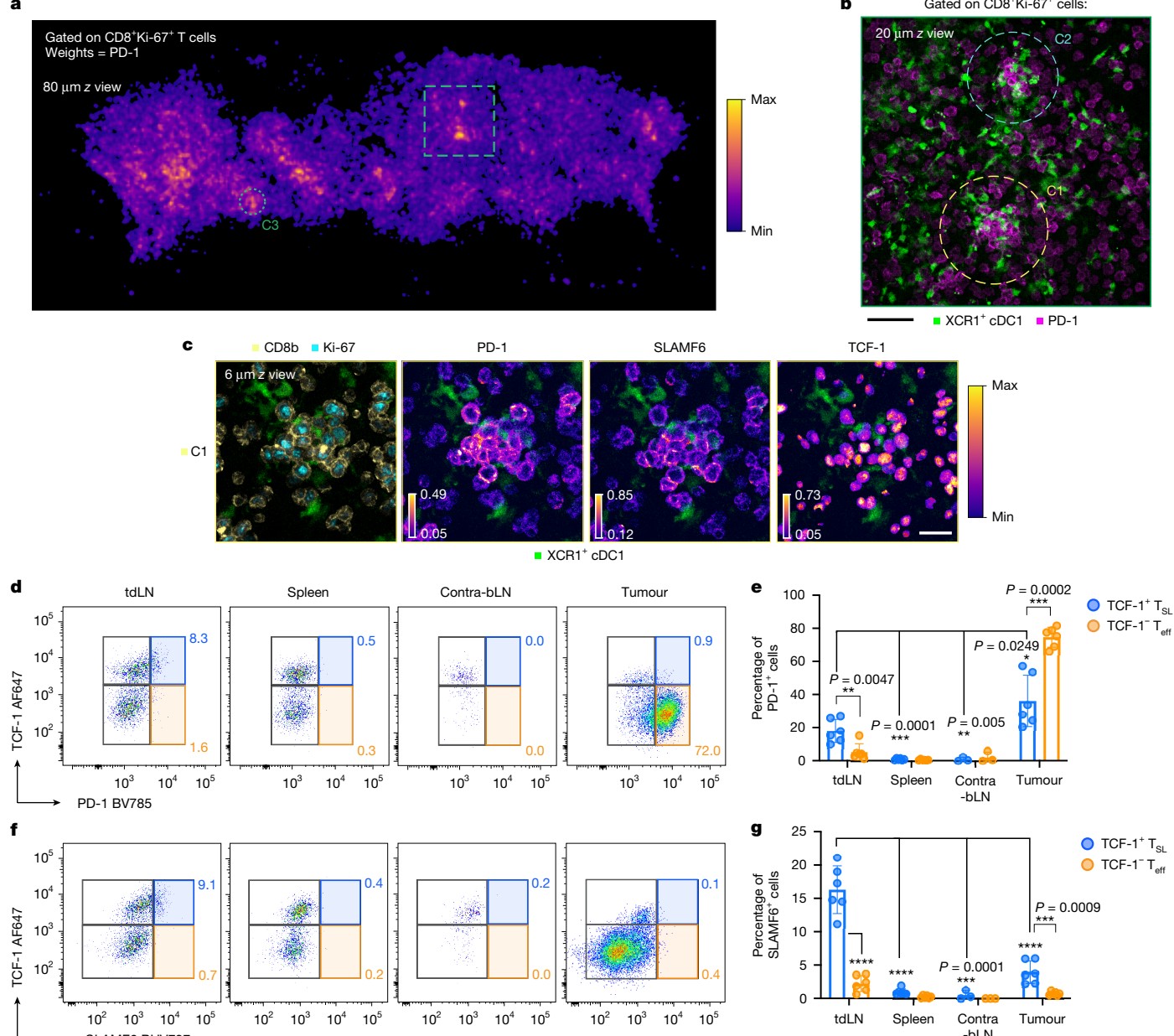

**Fig. 2 | PD-1⁺SLAMF6⁺ T_SL cells are uniquely retained in the tdLN.**

a–c, Optically cleared tdLN slice from a day 8 tumour-bearing mouse without adoptive transfer of transgenic T cells. Activated polyclonal CD8⁺ T cells are gated based on co-expression of CD8b and Ki-67. **a**, Day 8 inguinal draining lymph node kernel density map showing foci of activated polyclonal CD8⁺ T cells. The density was estimated using a Gaussian kernel of 6 μm bandwidth, weighted on normalized PD-1 expression. The green box indicates the region magnified in **b**, in which T cell–cDC1 cell clusters are further enlarged and displayed with individual protein markers: C1 (**c**) and C2 (Extended Data Fig. 3b). Data are representative of two independent experiments. $n = 5$. Scale bars, 50 μm (**b**)

and 20 μm (**c**). **d**,**f**, Representative gating strategy of OVA–tetramer⁺CD8⁺ T cells recovered from tissues collected from day 10 tumour-bearing mice and analysed using flow cytometry, based on TCF-1 and PD-1 (**d**) or SLAMF6 (**f**). **e**,**g**, Quantification of the proportions of PD-1⁺ (**e**) and SLAMF6⁺ (**g**) cells of the TCF-1⁺ T_SL and TCF-1⁻ T_eff cell subsets. Data are pooled from two independent experiments. $n = 6$ (tdLN, spleen, tumour) and $n = 3$ (contralateral brachial lymph node (contra-bLN)). Data are mean ± s.d. Statistical analysis was performed using unpaired two-sided $t$-tests; $*P < 0.05$, $**P < 0.01$, $***P < 0.001$, $****P < 0.0001$.

at various times after tumour implantation[15]. We determined that two successive diphtheria toxin doses are required to achieve >90% depletion in the lymph node (Supplementary Fig. 2a,b), although we observed that rare PD-1⁺ T_SL OT-I clusters still formed around the very few remaining cDC1 cells (Supplementary Fig. 2c).

Next, we initiated cDC1 cell depletion in KP-OVA tumour-bearing mice beginning at day 5 after induction (Fig. 3a), when late antigen presentation clusters were first observed (data not shown). Using H-2K^b–SIINFEKL tetramers to detect OVA-specific CD8⁺ T cells by flow

cytometry at day 10.5 (5.5 days after depletion), we found that late cDC1 cell depletion significantly reduced OVA-specific CD8⁺ T cell expansion in the tdLN (Fig. 3b). Nearly all of the reduction was accounted for by a decline in the TCF-1⁺ T_SL cell compartment (Fig. 3c). Gating on the SLAMF6⁺ T_SL cell subset similarly revealed a marked reduction compared with non-depleted mice (Fig. 3d and Extended Data Fig. 5a). The T_SL cell compartment of cDC1-cell-depleted tdLN also had a substantially diminished PD-1^high subset (Extended Data Fig. 5b), with the mean PD-1 intensity of T_SL cells decreasing to that of their T_eff cell

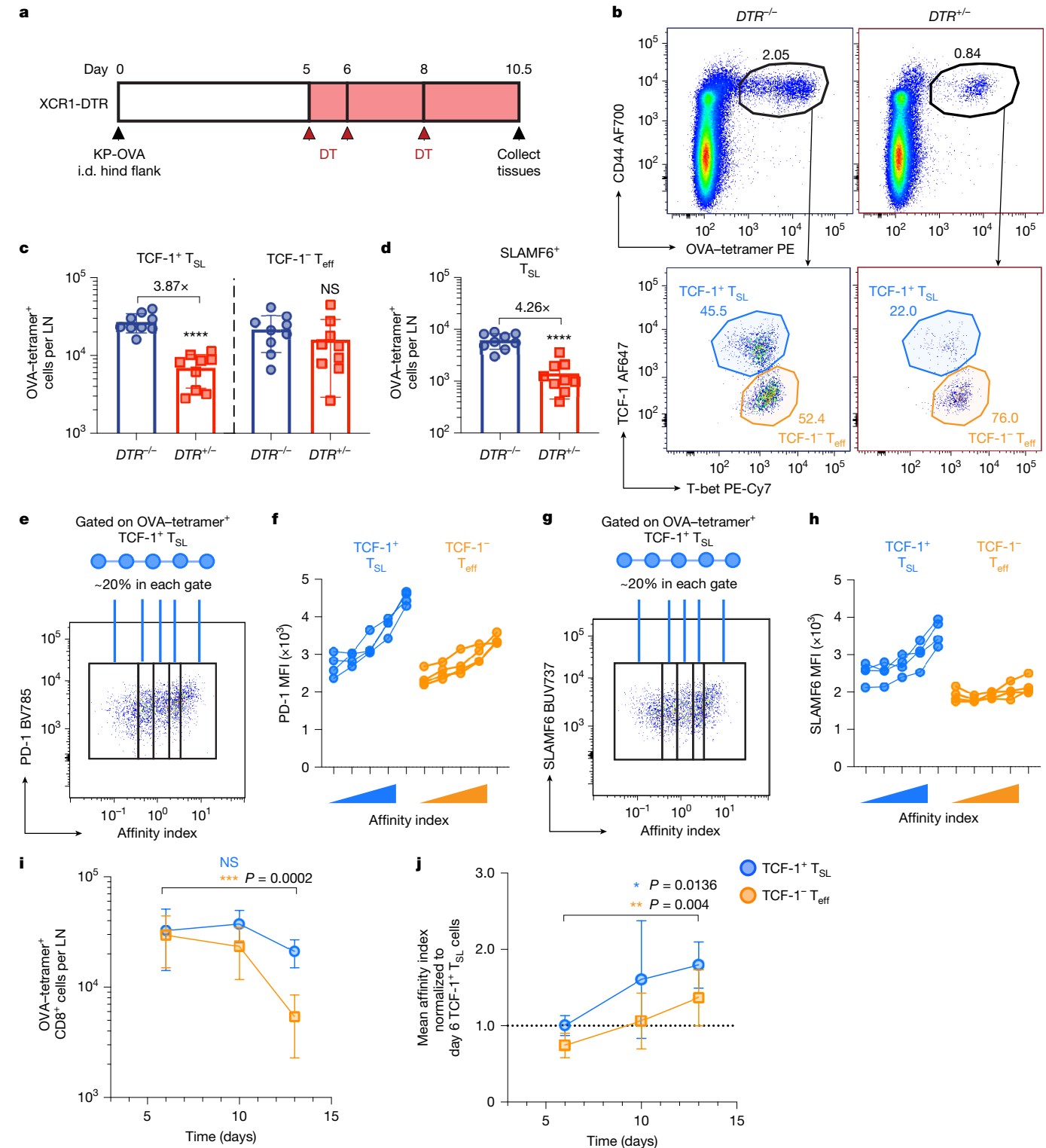

**Fig. 3** | See next page for caption.

counterparts (Extended Data Fig. 5c). This suggests that ablation of antigen-presenting cDC1 cells preferentially eliminated late $T_{SL}$ cell antigen signalling. By contrast, $T_{eff}$ cell expansion was only marginally affected in both the tdLN and the spleen, with negligible effect in the tumour (Fig. 3c and Extended Data Fig. 5d,e), showing that late antigen presentation by cDC1 cells in the tdLN is necessary for maintaining an expanded population of antigen-specific SLAMF6$^+$ $T_{SL}$ cells.

## Enrichment of high-affinity SLAMF6$^+$ $T_{SL}$ cells

An important question is whether the T cells that adopt the PD-1$^+$SLAMF6$^+$ $T_{SL}$ cell fate in the tdLN are a functionally selected subset of all the antigen-reactive T cells, or whether they are simply a cross-section of all such cells that, by virtue of their positioning within $T_{SL}$–cDC1 cellular niches, have taken on this phenotype. To determine

whether TCR dictates cell fate[25], we performed further analysis of H-2K^b–SIINFEKL tetramer-stained cells and found that tetramer binding was strongly correlated with PD-1 expression. As tetramer binding intensity alone can vary based on the total amount of TCR expressed on the cell surface, we normalized tetramer staining intensity to CD3 intensity (tetramer:CD3 staining ratio) to derive an imputed TCR affinity index of the T cells (a detailed explanation of and rationale for this approach is provided in the Methods). Next, because individual adult mice each possess a unique T cell repertoire spanning a range of TCR affinities, we further subdivided the OVA-specific CD8⁺ T cells from each mouse's tdLN samples into five separate bins of imputed affinity indices, with each containing 20% of the total OVA–tetramer⁺ population in that tdLN (Fig. 3e). We found that the compartment containing the highest imputed TCR affinity index was also associated with the highest mean expression of PD-1 within the $T_{SL}$ cell subset (Fig. 3f).

**Fig. 3 | Antigen-driven expansion and TCR affinity evolution of TCF-1⁺ $T_{SL}$ cells in the tdLN. a**, Schematic of the cDC1 cell depletion strategy in KP-OVA-bearing XCR1-DTR transgenic mice. DT, diphtheria toxin; i.d., intradermal. **b**, Representative flow cytometry plot showing the gating strategy for OVA–tetramer⁺CD8⁺ T cells (top) from day 10 tdLN and further gated on the TCF-1⁺ $T_{SL}$ and TCF-1⁻ $T_{eff}$ cell subsets (bottom). PE, phycoerythrin. **c,d**, Quantification of OVA-specific TCF-1⁺ $T_{SL}$ and TCF-1⁻ $T_{eff}$ cells (**c**) and TCF-1⁺SLAMF6⁺ $T_{SL}$ cells (**d**) recovered from the tdLN. Data are from three independent experiments. $n = 9$ per group. LN, lymph node. **e,g**, Representative flow cytometry plots showing PD-1 (**e**) and SLAMF6 (**g**) expression against imputed TCR affinity index (tetramer:CD3 staining ratio) of OVA–tetramer⁺TCF-1⁺ $T_{SL}$ cells from day 10 tdLNs, and the gating strategy that divides each population into five separate bins, each comprising around 20% of the total subset population. **f,h**, The mean

fluorescence intensity of PD-1 (**f**) and SLAMF6 (**h**) of the binned subsets from **e** and **g**. MFI, mean fluorescence intensity. Data are representative of two independent experiments. $n = 4$ per experiment. **i**, The total number of OVA–tetramer⁺CD8⁺TCF-1⁺ $T_{SL}$ and TCF-1⁻ $T_{eff}$ cells recovered from the tdLN across different timepoints. **j**, Imputed TCR affinity index plot of OVA–tetramer⁺TCF-1⁺ $T_{SL}$ and TCF-1⁻ $T_{eff}$ cells in the tdLN showing the evolution of average TCR affinity indices from day 6 to day 13 after tumour induction. Imputed affinity indices were normalized to the mean imputed affinity index of day 6 TCF-1⁺ $T_{SL}$ cells of the same experiment. Data are pooled from two independent experiments. $n = 8$ (days 6 and 13) and $n = 7$ (day 10). Data are mean ± s.d. Statistical analysis was performed using unpaired two-tailed $t$-tests (**c** and **d**) and one-way ANOVA with Tukey's multiple-comparison test (**i** and **j**); NS, not significant; *$P < 0.05$, **$P < 0.01$, ***$P < 0.001$, ****$P < 0.0001$.

Similarly, SLAMF6 protein expression was highly correlated with both PD-1 expression and the imputed TCR affinity index of $T_{SL}$ cells (Fig. 3g,h). As SLAMF6⁺ $T_{SL}$ cells express higher levels of the co-receptor CD8 than the other subsets (data not shown) and such expression is known to enhance tetramer binding/functional ligand avidity of a T cell[26], we gated on an intermediate band of CD8-expressing $T_{SL}$ cells (CD8^int) to more directly assess tetramer binding. PD-1 expression remained positively correlated with imputed TCR affinity even under such constraints (Extended Data Fig. 5f). Moreover, we observed a similar PD-1 and imputed TCR affinity correlation in the draining lymph node of mice immunized with OVA protein (Extended Data Fig. 5g,h). Importantly, the positive correlation between PD-1 and imputed TCR affinity is seen uniquely in the antigen-draining lymph node and is not observed in the spleen and other non-antigen-presenting tissues (Extended Data Fig. 5h). These data collectively suggest that, in a polyclonal CD8⁺ T cell setting, T cells with higher TCR affinity are selectively enriched among cells adopting the PD-1⁺SLAMF6⁺ $T_{SL}$ cell fate.

## Antigen-driven affinity evolution of $T_{SL}$ cells

The presence of distinct cellular niches in a physiological context suggests the possibility of a selection mechanism that can be biologically relevant. The positive correlation of PD-1 and SLAMF6 expression with imputed TCR affinity, combined with the observation of enriched PD-1⁺SLAMF6⁺ polyclonal $T_{SL}$ cell clusters around cDC1 cells (Fig. 2a–c), suggests that higher-affinity OVA-specific $T_{SL}$ cell clones are predisposed towards occupying the late $T_{SL}$–cDC1 cell antigen-presentation niches.

We hypothesized that these late antigen presentation niches drive affinity evolution of antigen-specific CD8⁺ T cell responses over time through selective enrichment and expansion of higher TCR affinity clones. Importantly, owing to the precursor–product relationship between the less differentiated $T_{SL}$ and the more terminally differentiated $T_{eff}$ cell subsets, such selective expansion of CD8⁺ T cell clones at the $T_{SL}$ progenitor stage ensures the generation of high-affinity

progenitors and, eventually, amplification of their progeny as high-affinity effector cells.

This model predicts a progressive increase in the average imputed TCR affinity of both $T_{SL}$ and $T_{eff}$ cell subsets in the tdLN over time. To test this hypothesis, we examined the imputed TCR affinity indices of OVA-specific CD8⁺ $T_{SL}$ and $T_{eff}$ cells from tdLNs collected at different timepoints after tumour induction. Indeed, despite the number of $T_{SL}$ cells remaining relatively constant throughout, the average imputed TCR affinity of $T_{SL}$ and $T_{eff}$ cell subsets steadily increased over time from day 6 to day 13 (Fig. 3i,j and Extended Data Fig. 5i–k). Temporally, the imputed TCR affinity of $T_{eff}$ cells lagged slightly behind those of $T_{SL}$ cells, agreeing with the notion that the enrichment of higher-affinity $T_{SL}$ cells subsequently gave rise to the higher-affinity $T_{eff}$ cells. Together, our findings suggest that late antigen presentation is critical in facilitating the affinity evolution of CD8⁺ $T_{SL}$ clones in a spatiotemporal manner.

## PD-1 action maintains $T_{SL}$ stem-like state

The preceding results raise an important question, namely, how high-TCR-affinity CD8⁺ $T_{SL}$ cells retain their stemness while undergoing continued antigen receptor signalling and proliferation. Previous studies have established that TCR signalling strength is closely associated with CD8⁺ T cell differentiation fate, with stronger, more prolonged TCR signalling driving effector differentiation genes (for example, *TBX21* (encoding T-bet) and *PRDM1* (encoding BLIMP1)), whereas intermediate and weaker signalling promoted a memory phenotype (for example, *BCL6* and *EOMES*) (reviewed previously[27]). These past observations predict that sustained late antigen signalling by high-affinity CD8⁺ $T_{SL}$ cells should prime these cells towards terminal effector differentiation, in contrast to our observations.

On the basis of the elevated level of inhibitory PD-1 receptor expression among SLAMF6⁺ $T_{SL}$ cells, and some evidence that this molecule can contribute to preventing terminal differentiation of CD8⁺ T cells[28–30], we surmised that PD-1 inhibitory function in the CD8⁺ $T_{SL}$ cells could have a role in attenuating TCR signals, enabling the preservation of stem-like state. We conducted high-resolution 3D imaging of tdLN slices stained with non-blocking anti-mouse PD-1 antibodies (clone RMP1-30)[31,32] to enable in situ detection of PD-1 engaged with its ligands. Clustered OT-I $T_{SL}$ cells interacting with cDC1 cells demonstrated polarization of PD-1 towards the T cell–dendritic cell synaptic interface (Fig. 4a). These features resembled the formation of PD-1 microclusters previously reported to mediate inhibitory signalling through the recruitment of the phosphatase SHP2 (ref. 33). We included polyclonal anti-mouse PD-L1 antibodies in subsequent tissue staining experiments, which revealed the co-localization of PD-1 on OT-I cells and PD-L1 expressed on cDC1 cells (Fig. 4b and Extended Data Fig. 6a), indicating active engagement of PD-1 and its ligands during late antigen presentation.

Using flow cytometry to measure the co-stimulatory and co-inhibitory ligands expressed by dendritic cells in the tdLN, we found that, whereas PD-L1 was highly expressed among migratory dendritic

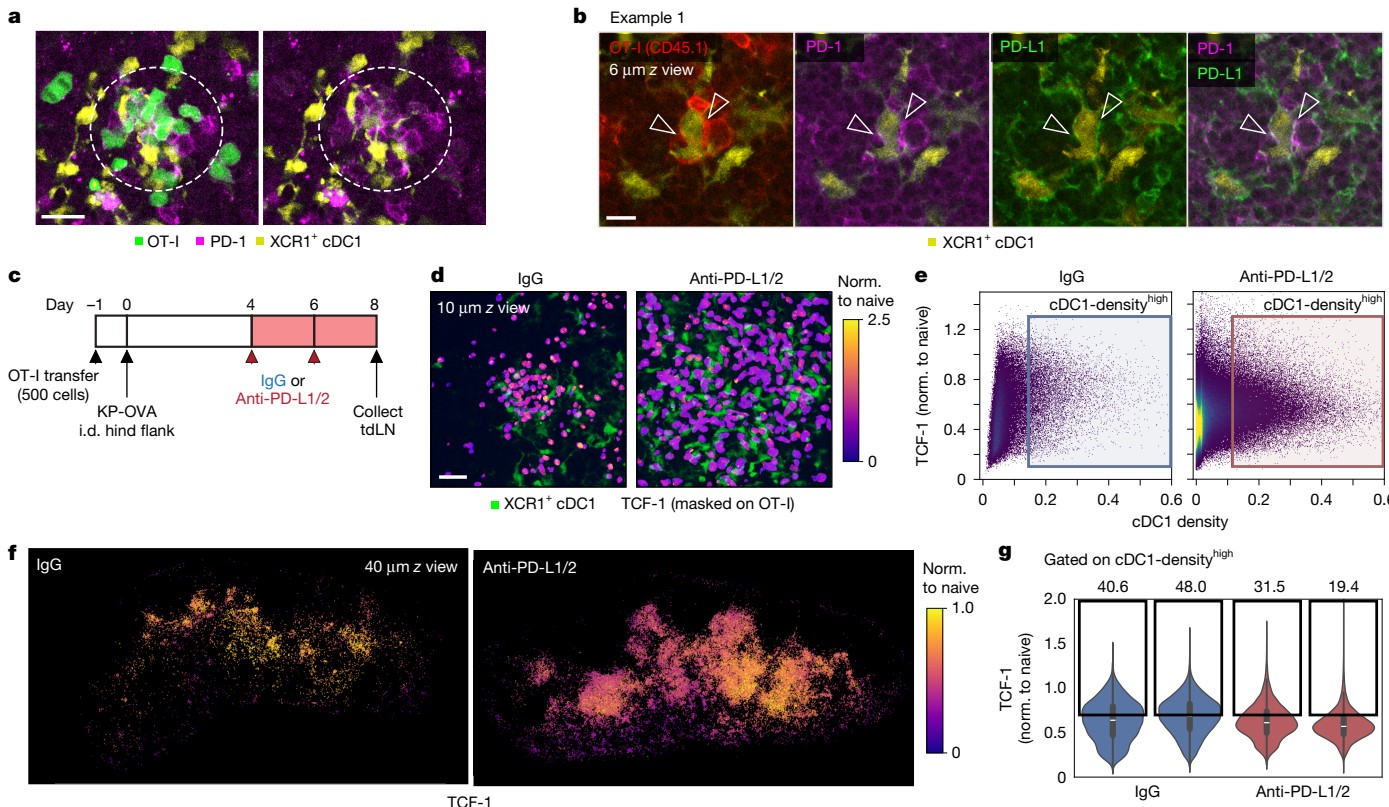

**Fig. 4 | PD-1 signalling sustains and regulates the stem-like state of CD8⁺TCF-1⁺ T_SL cells.** **a**, High-resolution 3D scans of cleared day 8 tdLN stained with non-blocking anti-PD-1 antibody. A magnified example of TCF-1⁺ OT-I T_SL cells clustered around a cDC1 cell network, showing punctate microclusters of PD-1 proteins polarized towards the T cell–dendritic cell synaptic interface. Right, image without the OT-I GFP channel. Data are representative of at least two independent experiments. $n = 2$ per experiment. Scale bar, 20 μm. **b**, Representative images of an example of clustering OT-I cells and the co-localization of PD-1 (magenta) and OT-I CD45.1 (red), with PD-L1 (green) expressed on cDC1 (yellow). A thin cross-section of a 3D stack is shown (4 μm z thickness) and the arrowheads indicate polarization and punctate microcluster formation of PD-1. Note the co-localization of PD-1 and PD-L1 blending into white pixels, indicating receptor–ligand engagement. Data are representative of at least two independent experiments. $n = 4$. Scale bar, 10 μm. **c**, Experimental schematic for **d**–**g**, with tumour-bearing mice receiving either IgG (control) or anti-PD-L1/PD-L2 blocking antibodies from day 4 onwards. **d**, Representative magnified images of TCF-1 expression (masked on OT-I cells) within a dense cDC1 region. Scale bar, 30 μm. **e**,**f**, Histocytometry plots (**e**) showing the distribution of OT-I cells in day 8 tdLN and their relative expression of TCF-1 (normalized (norm.) to naive T cells) and cDC1 cell density (proximity to dense cDC1 cell region). The cDC1-density^high subset is thresholded on mean + 1 s.d. above cDC1 cell density (normalized to a maximum cDC1 cell density value of OT-I) and their spatial distribution as well as relative TCF-1 expression are shown (**f**) as a 40-μm cross-section view of the tdLN. **g**, Quantification of TCF-1 expression among cDC1-density^high OT-I cells (**e** and **f**). The gates show the relative frequencies of cDC1-density^high OT-I cells with TCF-1 expression (normalized to naive T cells) of >0.7. The box plots show the median (centre line), first and third quartiles (box limits), and 1.5 × interquartile range (whiskers). Data are representative of two independent experiments. $n = 4$ per group.

cell subsets even at steady state, a marked increase in PD-L2 (another ligand of PD-1) can be observed in the day 8 tdLN, and especially among the cross-presenting cDC1 cell subset (Supplementary Fig. 3). This suggests that both PD-1 ligands can potentially contribute to PD-1-mediated TCR signal attenuation during late antigen presentation.

To investigate the role of PD-1 inhibitory signalling in promoting the expansion/survival of T_SL cells, we performed checkpoint co-blockade with anti-PD-L1 and anti-PD-L2 (Fig. 4c). Day 8 tdLN revealed a proliferative burst of OT-I after checkpoint blockade, as anticipated from the loss of PD-1 inhibition (Fig. 4d). Whereas OT-I cells spatially associated with the cDC1 cell network in the T cell zone displayed higher TCF-1 expression in control animals, an extensively divided OT-I cell population with heterogeneous TCF-1 levels occupied the dense cDC1 cell region after checkpoint blockade (Fig. 4d). Gating on OT-I cells in close spatial proximity to cDC1 cells revealed a lower mean TCF-1 expression compared with OT-I cells from control animals (Fig. 4e–g). Checkpoint blockade led to intensified cleaved caspase-3 staining across the tdLN, including OT-I cells that formed clusters with cDC1 (Extended Data Fig. 6b,c). Increased cleaved caspase-3 expression is

associated with T cell contraction accompanied by apoptosis and cell death[34]. Strong antigen signalling, especially in the absence of PD-1 inhibition, is known to cause activated-induced cell death[35]. Using flow cytometry, OVA–tetramer⁺SLAMF6⁺ T_SL cells in the tdLN showed significantly higher expression of this apoptosis-associated marker in anti-PD-L1/PD-L2-treated mice (Extended Data Fig. 6d). These data indicate that disruption of PD-1 signalling drives TCF-1 downregulation among T cells receiving strong antigen signalling, together with an increase in apoptosis.

## PD-1 blockade reduces high-affinity T_SL cells

Our findings suggested that blockade of PD-1 inhibitory signalling axis can disrupt maintenance of the stem-like state and/or drive antigen-engaged T_SL cells towards cell death, potentially dysregulating the clonal landscape of responding T cells. We therefore investigated the effects of PD-1 checkpoint blockade on polyclonal OVA-specific T cells from KP-OVA tumour-bearing mice, collecting tdLNs at day 12 after tumour induction (8 days after initial checkpoint blockade

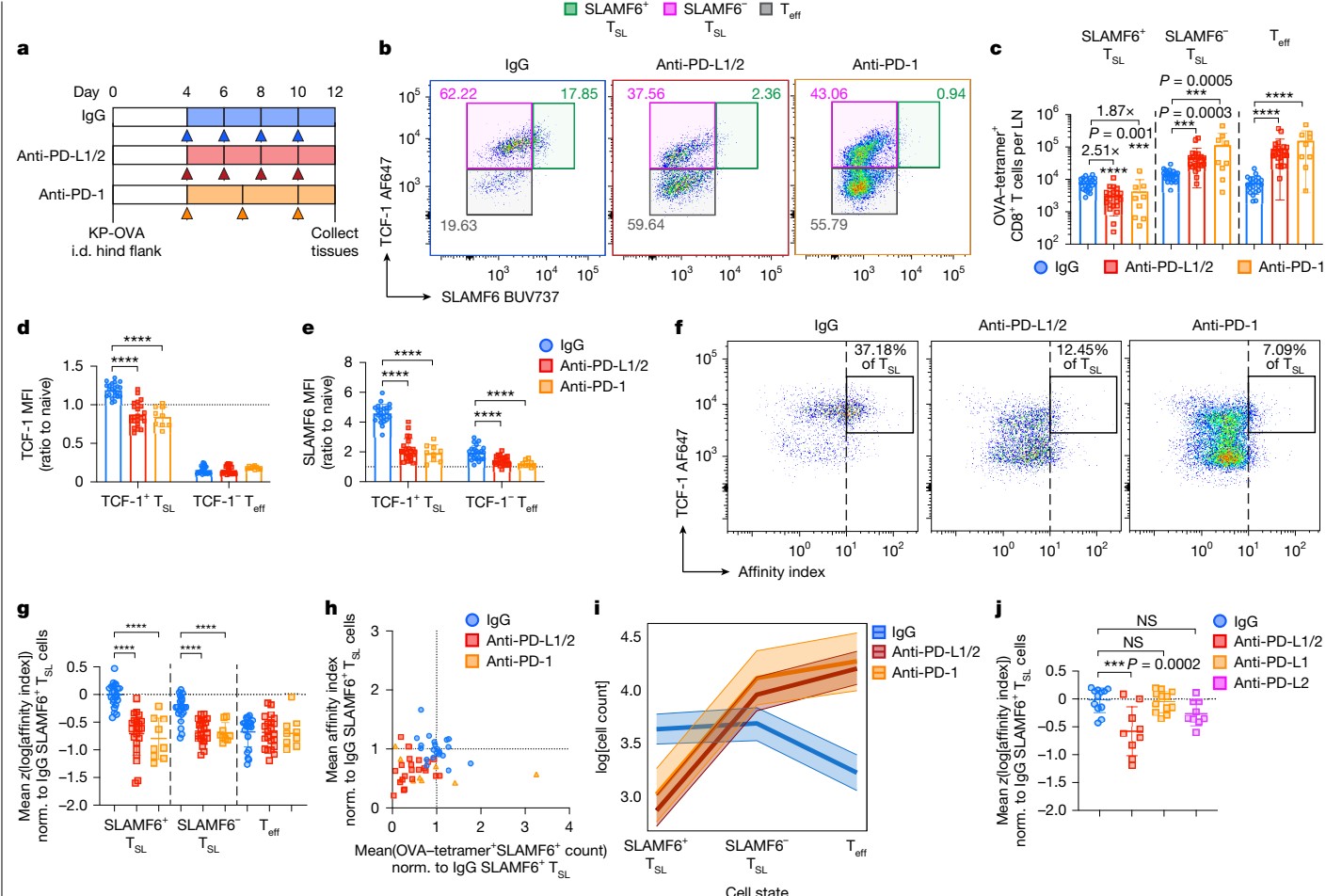

**Fig. 5 | PD-1 checkpoint blockade disrupts the accumulation and survival of high-affinity CD8⁺ T_SL cells. a**, Schematic of the checkpoint blockade treatment strategy. **b,c**, Representative flow cytometry plots (**b**) and quantification (**c**) of OVA–tetramer⁺CD8⁺ T cell subsets from day 12 tdLN from mice that had received IgG (control), anti-PD-L1/PD-L2 or anti-PD-1 checkpoint blockade treatment. The gating strategy defines the three major T_SL and T_eff cell subsets. The total numbers of each subset are given in **c**. **d,e**, The mean fluorescence intensity of TCF-1 (**d**) and SLAMF6 (**e**) of each subset of OVA–tetramer⁺CD8⁺ T cells, normalized to naive (CD44⁻TCF-1⁺PD-1⁻) CD8⁺ T cells. **f**, Representative flow cytometry plots showing TCF-1 expression against imputed TCR affinity indices (tetramer:CD3 staining ratio) of OVA–tetramer⁺CD8⁺ T cells of each treatment group. The box indicates a high-affinity TCF-1⁺ T_SL cell subpopulation based on an arbitrarily defined imputed affinity index. **g,j**, Standardized log-transformed imputed TCR affinity indices of each treatment group. Each datapoint represents the mean subset value from each animal.

Imputed affinity indices were normalized to the mean of the log-transformed imputed affinity index of IgG (control) group SLAMF6⁺ T_SL cells from the same experimental cohort. **h**, 2D plot projecting the relationship between the SLAMF6⁺ T_SL cell TCR affinity index versus SLAMF6⁺ T_SL cell count from each animal. The imputed TCR affinity index and cell counts were normalized to the averages of the IgG (control) group from the same experimental cohort. **i**, Bayesian multilevel linear modelling showing the estimated high-affinity (defined as above mean imputed TCR affinity threshold) cell counts of each subset on a logarithmic scale. Data are from 7 independent experiments (n = 22 (IgG and anti-PD-L1/2) and n = 9 (anti-PD-1)), with the anti-PD-1 group from two independent experiments. Data from **j** are pooled from 3–4 independent experiments (n = 12 (IgG and anti-PD-L1) and n = 9 (anti-PD-L1/2 and anti-PD-L2)). Data are mean ± s.d. Statistical analysis was performed using one-way ANOVA with Tukey's multiple-comparison test; ***P < 0.001, ****P < 0.0001.

treatment) (Fig. 5a). Mice treated with a combination of anti-PD-L1 and anti-PD-L2 showed robust proliferation of both OVA-specific T_SL and T_eff cells as anticipated (Fig. 5b,c and Extended Data Fig. 7c), as well as partial suppression of tumour growth (Extended Data Fig. 7a). However, the proliferative bursts were accompanied by a phenotypic shift among the T_SL cell compartment, characterized by downregulation of stem-like markers (TCF-1 and SLAMF6) (Fig. 5b,d,e) and a decrease in SLAMF6⁺ T_SL cell numbers compared with non-treated tdLN (Fig. 5c).

We also examined the effect of PD-1 blockade on the imputed affinity of the T_SL cells. We found that the average imputed TCR affinity of the remaining SLAMF6⁺ T_SL and SLAMF6⁻ T_SL cells was substantially lower after PD-1 blockade (Fig. 5f,g and Extended Data Fig. 7b). Similar findings were also obtained when mice were treated with an anti-PD-1 blocking antibody clone[36,37] (Fig. 5b–g and Extended Data Fig. 7a,b),

indicating that the observed phenotypic changes were directly related to the PD-1 inhibitory signalling rather than the impaired function of PD-L1 and PD-L2, which can ligate with other proteins such as CD80 (ref. 38) and RGMb[39] to mediate inhibitory functions.

Owing to a few outliers in the dataset (Fig. 5h), we conducted further analysis using Bayesian linear modelling to control for potential experimental variability (Fig. 5i and Methods). This modelling revealed that the loss of high-affinity SLAMF6⁺ T_SL cells after checkpoint blockade was associated with a corresponding increase in high-affinity SLAMF6⁻ T_SL and T_eff cells, although the expansion of lower-affinity T_eff cells also occurred at a much greater rate (Fig. 5i). This supports the hypothesis that the inhibitory PD-1 pathway regulates the T cell clonal response to antigens. Blocking this pathway causes the high-affinity clones to either differentiate into T_eff cells or undergo apoptosis (Extended Data Fig. 6b–d), allowing lower-affinity clones more opportunity to engage

with antigens and to do so when relieved of their lower but significant PD-1 inhibition, resulting in their selective expansion.

As many factors influence tetramer-binding ability, we performed additional control experiments in which mice transferred with monoclonal OT-I T cells (OT-I.Rag1) were subjected to the same anti-PD-L1/PD-L2 checkpoint treatment strategy. OT-I cells showed a similar decline in TCF-1 and SLAMF6 expression after checkpoint blockade (Extended Data Fig. 7d–f) but, as expected for a monoclonal TCR population, the imputed TCR affinity remained similar among cells from the different treatment groups (Extended Data Fig. 7g,h). PD-1 blockade can lead to downregulation of the CD8 co-receptor, influencing tetramer binding[26]. We therefore also measured CD8 downregulation in SLAMF6$^+$ T$_{SL}$ OT-I cells from treated mice and confirmed that although OT-I experienced CD8 downregulation as did cells in the polyclonal antigen-specific repertoire, their imputed affinity index remained unaffected (Extended Data Fig. 7i,j). To further examine whether TCR downregulation explains the differential tetramer binding, we subdivided TCF-1$^+$ T$_{SL}$ cell populations from OT-I and polyclonal experiments on the basis of their TCR expression. At a given TCR level, the imputed affinity remains unchanged in OT-I cells after checkpoint blockade but is significantly reduced among polyclonal OVA-specific cells (Extended Data Fig. 7k,l). Together, these findings support the conclusion that the polyclonal cells show a true shift in imputed TCR affinity as a population after checkpoint treatment.

The average imputed TCR affinity of stem-like CD8$^+$ T cells continues to evolve over days (Fig. 3j and Extended Data Fig. 5k). We next determined whether initiating blockade at a later timepoint (day 10) when the evolving T cell response is more matured would also compromise T$_{SL}$ cell clonal affinity (Extended Data Fig. 8a). Such late blockade similarly led to robust expansion of SLAMF6$^-$ T$_{SL}$ and T$_{eff}$ cells accompanied by TCF-1 and SLAMF6 downregulation (Extended Data Fig. 8b,c). Total SLAMF6$^+$ T$_{SL}$ in the control group declined substantially from day 12 to day 19.5 (Fig. 5c and Extended Data Fig. 8d), probably due to diminished antigen presentation in the tdLN during late tumour growth. Although the number of high-affinity SLAMF6$^+$ T$_{SL}$ cells (defined as above the mean affinity index of SLAMF6$^+$ T$_{SL}$ cells in the control group) was not significantly decreased, late blockade still led to markedly reduced average imputed affinity of SLAMF6$^+$ T$_{SL}$ cells, with high-affinity clones comprising only a small fraction of this subset (Extended Data Fig. 8e–h). These data reinforce the notion that PD-1 pathway continuously regulates and shapes T cell clonal competition in the antigen presentation niches and mediates the retention of the stem-like state among T cells that continue to receive late antigen signals.

We further determined whether the loss of high-affinity SLAMF6$^+$ T$_{SL}$ cells persists after checkpoint treatment has stopped by extending the end point for 9–10 days (Extended Data Fig. 9a). Most OVA-specific CD8$^+$ T cells continued to show lower TCF-1 and SLAMF6 levels and the numbers of total and high-imputed-affinity SLAMF6$^+$ T$_{SL}$ cells remained substantially reduced (Extended Data Fig. 9b–i). Thus, cessation of checkpoint treatment did not reverse the phenotypic shift towards SLAMF6$^-$ T$_{SL}$ cells nor did it replenish the high-imputed-affinity SLAMF6$^+$ T$_{SL}$ cell pool.

PD-1 is also highly expressed among activated CD4$^+$ T cells, including both conventional T helper and regulatory T cells. Both populations are known to have a major influence on CD8$^+$ T cell responses, including control of TCR affinity distribution[40]. Pre-depletion of both conventional CD4$^+$ and regulatory T cells yielded enhanced SLAMF6$^+$ T$_{SL}$ cell generation in the tdLN (Extended Data Fig. 10a,d,e). Nonetheless, checkpoint blockade still resulted in significant downregulation of TCF-1 and SLAMF6 (Extended Data Fig. 10a–c), reduced SLAMF6$^+$ T$_{SL}$ numbers (Extended Data Fig. 10d,e), reduced imputed TCR affinity (Extended Data Fig. 10f,g), as well as a similar partial inhibition of tumour growth as seen in non-CD4-depleted mice (Extended Data Fig. 10h). Thus, CD4$^+$ T cells did not appear to have a substantial role in driving the loss of stem-like state and the change in clonal repertoire of CD8$^+$ T$_{SL}$ cells reported here.

Finally, given that our blocking strategy relied on the near-complete repression of PD-1-mediated inhibition, we explored whether partial blockade with single-ligand antibody treatment (anti-PD-L1 or PD-L2 only) would be sufficient to reproduce the loss of high-affinity CD8$^+$ T$_{SL}$ cells in the tdLN. We found that anti-PD-L1 blockade alone drove moderate but less effective tumour regression as compared to co-blockade treatment (Extended Data Fig. 11a,b), along with moderate expansion of both T$_{SL}$ and T$_{eff}$ cells in the spleen (Extended Data Fig. 11d,e). Notably, in contrast to dual PD-1 ligand blockade, the numbers and imputed TCR affinity index of SLAMF6$^+$ T$_{SL}$ cells after anti-PD-L1 blockade alone were not significantly decreased (Fig. 5j and Extended Data Fig. 11c,f). However, PD-L2 blockade alone did not induce significant expansion of either T$_{SL}$ or T$_{eff}$ cells nor contribute to effective control of tumour growth (Extended Data Fig. 11a–d).

Overall, these data reveal that strongly impaired PD-1 inhibition of TCR signalling is required to promote robust effector expansion capable of curbing tumour growth, but such thorough blockade of this inhibitory pathway also caused the loss of high-affinity SLAMF6$^+$ T$_{SL}$ cells. These key precursors can be preserved with less effective interference with the PD-1 inhibitory axis, but this also substantially diminishes the resultant anti-tumour effect of therapy.

## Discussion

Contrary to the widely accepted model that prolonged antigen stimulation of high-affinity cells in the secondary lymphoid tissue drives terminal differentiation towards an effector state[27], we show here that ongoing TCR input in the tdLN promotes the expansion of high-TCR-affinity clones as stem-like progenitors owing to the action of the PD-1 inhibitory pathway that attenuates antigen-driven signalling. Antigen-bearing cDC1 cells serve as evolution sites that enable high-affinity clones to receive TCR signals, proliferate and out-compete lower-affinity clones over time. As T$_{SL}$ cells are precursors to T$_{eff}$ cells, expansion of high-affinity T$_{SL}$ cells subsequently gives rise to a large pool of high-affinity T$_{eff}$ cells. Such a mechanism is therefore congruent with earlier studies showing strong effector bias among T cells with high TCR ligand affinity. However, it changes our understanding from one in which high-affinity antigen recognition directly promotes exit from a stem-like state, to one in which high-affinity TCR signalling attenuated by a feedback inhibitory pathway (PD-1) drives the enrichment of stem-like cells, of which the progeny differentiate into effectors. Importantly, this feedback mechanism enables high-affinity clones to be preserved as T$_{SL}$ cells. The prolonged association of high-affinity T$_{SL}$ cells with cDC1 cells also helps to explain the previous observations that lower-affinity antigen-specific clones egress from lymphoid tissues earlier than their high-affinity counterparts[41,42]. A recent study found that tumour-antigen-specific T cells remained highly enriched in the tdLN as T$_{SL}$ cells and differentiate into effectors after migrating to the TME[43]. Another study reported that suboptimal TCR signalling in mouse tumour model promotes accelerated differentiation into effector cells[20]. Both reports are consistent with our findings here.

Our results implicate the PD-1 inhibitory pathway in maintaining stemness of high-affinity T$_{SL}$ cells through blunting of TCR input during prolonged antigen stimulation in the tdLN. While the locus of PD-1 inhibition[33,44,45] has been debated, recent data from this laboratory confirm the TCR as the primary target[46]. This view is also supported by data showing a bias towards terminal differentiation of CD8$^+$ T cells in PD-1 deficient mice[28,30] and fits with a negative-feedback model in which strong TCR signalling by high-affinity T$_{SL}$ cells drives strong upregulation of PD-1 (ref. 29). In acute lymphocytic choriomeningitis virus infection, PD-1 deficient progenitors preferentially upregulate exhaustion signatures, therefore reinforcing the model that PD-1 negative signalling helps to promote the retention of a stem-like state[47]. An earlier study proposed a similar affinity-biased model for control of effector CD4$^+$ T cells, in which higher-affinity T cells induce greater

TCR-dependent CTLA4 recruitment to the immune synapse, resulting in stronger inhibition of co-stimulatory signalling[48]. In vivo blockade with anti-PD-L1 prolongs effector CD4[+] T cell–dendritic cell interactions, further supporting a key role for TCR-dependent PD-1 expression in regulating the duration and extent of high antigen level-dependent signalling[49]. These linked signalling and feedback mechanisms yield a finely tuned system controlling TCR input to support $T_{SL}$ fate among an optimal cohort of T cells.

These findings raise important clinical implications regarding PD-1 checkpoint immunotherapy. Changes in the tumour-infiltrating lymphocyte clonal landscape, sometimes referred to as clonal replacement, have been reported in patients undergoing anti-PD-1 immunotherapy, with extinction of pre-existing clones and emergence of novel tumour-reactive clones[50–53]. Our preclinical model suggests that, during PD-1 blockade, biased differentiation of high-affinity $T_{SL}$ cells towards the effector state as well as increased activated-induced cell death of these cells could promote a repertoire shift by enabling lower-affinity clones to gain access to the antigen-bearing dendritic cell and expand. Such changes may compromise subsequent treatment efficacy, as higher TCR affinity and avidity CD8[+] TILs are associated with better tumour clearance in anti-PD-1-treated patients[54,55] and superior tumour infiltration and control after adoptive cell therapy[56]. Given that high-affinity CD8[+] TILs can rapidly become dysfunctional after exposure to persistent antigens in the TME, these cells may have only a narrow time window to facilitate effective tumour killing before losing functional cytotoxicity[42,57]. The ability of non-metastatic tdLNs to act as an ongoing source of TCF-1[+] $T_{SL}$ cells, as recently reported[3–6], could become dysregulated by PD-1 checkpoint immunotherapy, leading to impaired long-term maintenance of high-affinity $T_{SL}$ cell clones in the tdLNs; the loss of these cells could be permanent due to reduced naive T cell output from the adult thymus[58,59]. Recent clinical evidence reporting a declining efficacy of repetitive checkpoint treatment in patients experiencing tumour relapse fits with this view[60]. Finally, our findings indicate that partial blockade of the PD-1 pathway could avoid the loss of high-affinity $T_{SL}$ cells at the price of decreased anti-tumour efficacy in any single round of treatment, suggesting that a sweet spot could be found through careful titration of checkpoint antibody dosage.

In summary, our study provides insights into how the PD-1 inhibitory pathway regulates finely tuned processes that enable the immune system to maximize the use of its diverse TCR repertoire without rapid loss of the most antigen-responsive cells. Interference with this tuning disturbs a carefully balanced system, with potentially detrimental long-term consequences to host responses. We propose that more careful attention to the countervailing effects of checkpoint blockade will be important in maximizing the clinical benefit of such treatments going forwards.

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

## Methods

### Mice

CD45.2 (C57BL/6J) and B6.GFP (C57BL/6-Tg(UBC-GFP)30Scha/J) mice were purchased from the Jackson Laboratory (strains 000664 and 004354, respectively); CD45.1 (B6.SJL *Ptprc*[a]), OT-I.CD45.1 (B6(Ly5.1)-[Tg]TCR OT-I-[KO]RAG1) and OT-I.CD45.2 (C57BL/6NAi-[Tg]TCR OT-I-[KO]RAG1) were obtained from the NIAID-Taconic exchange programme (strains 8478, 300 and 175, respectively). XCR1-DTR (B6. Cg-Xcr1[tm2(HBEGF/Venus)Ksho]) and XCR1-venus (B6.Cg-Xcr1[tm1Ksho]) transgenic mice[15] were gifts from T. Kaisho. OT-I.GFP mice were cross-bred from OT-I.CD45.2 and B6.GFP mice and maintained as a homozygous strain in our laboratory. The majority of mice used were female and aged 6–16 weeks at the beginning of experiments, with a small number of experiments performed in male mice. No significant difference was observed between sex. Age-matched littermate mice were used to control for litter, cage and age effects. For imaging experiments, two to three mice were used per group in each experiment, and for flow cytometry experiments, three to six mice were used per group. Mice were randomly assigned for experimental groups, and animal experiments were not blinded as the procedures were performed by the same investigator. All mice were bred and maintained under specific-pathogen-free conditions at an American Association for the Accreditation of Laboratory Animal Care (AAALAC)-accredited animal facility within NIAID and were used (including compliance with tumour size limit) under study protocol LISB-4E approved by NIAID Animal Care and Use Committee (NIH).

### Tumour cell line generation

The KP-OVA tumour line was generated by lentiviral transduction of KP6-1B11 cells (a single colony subcloned from the KP1233 cell line derived from a *Kras*[G12D/+]*Trp53*[−/−] mouse[14]) with a lentivirus expressing full-length OVA (LRG-EFS-ZsGreen-P2A-OVA). pcDNA3-OVA (Addgene, 64599) was cloned into the lentiviral vector LRG-EFS-ZsGreen-P2A, a plasmid modified from LRG vector[61], a gift from J. Shi, and transfected into HEK293T cells, a gift from T. Jacks, for lentiviral production. ZsGreen[+] KP6-1B11 cells were sorted into 96-well plates containing single cells per well. Single-cell-derived clones with homogenous morphology and ZsGreen expression were selected and expanded for western blot and flow cytometry validation using anti-OVA (Invitrogen, PA1-196) and anti-mouse SIINFEKL–H-2K[b] (25-D1.16) antibodies. The KP-OVA line was cultured and passaged in RPMI medium containing 10% FCS, L-glutamine (2 mM), penicillin (100 U ml[−1]), streptomycin (0.1 mg ml[−1]), sodium pyruvate (1 mM), HEPES (10 mM) and 2-mercaptoethanol (1 mM) at 37 °C under 6.5% $CO_2$.

The MC38-OVA tumour line was generated by lentiviral transfection of MC38 cells with lentivirus expressing full-length OVA (pLV-EF1a-OVA-puro). The MC38 cell line was generously provided by M. Meier-Schellersheim (NIAID, NIH). pcDNA3-OVA (Addgene, 64599) was cloned into the pLV-EF1a-puro lentiviral vector. Transfection was performed on HEK293T cells (Takara, 632180) for lentivirus production. Infected cells were selected with puromycin (8 µg ml[−1]) and OVA expression was assessed by flow cytometry using anti-mouse SIINFEKL–H-2K[b] (BioLegend, 25-D1.16). The MC38-OVA line was cultured and passaged in DMEM medium containing 10% FCS, L-glutamine (2 mM), penicillin (100 U ml[−1]), streptomycin (0.1 mg ml[−1]), sodium pyruvate (1 mM) and HEPES (10 mM) at 37 °C under 6.5% $CO_2$. All HEK293T lines were tested negative for mycoplasma, but the tumour cell lines were not tested for mycoplasma.

### Tumour induction and protein immunization

For tumour induction, mice were anaesthetized with isoflurane inhalation (2% induction, 1–1.5% maintenance). Mice were shaved on the left flank with a Wahl clipper (Kent Scientific), depilated using Nair hair removal cream and the shaved flank was washed thoroughly with water-soaked gauze. Tumour cells (KP-OVA and MC38-OVA) were collected by washing with PBS, incubated with 0.25% trypsin/ EDTA (Thermo Fisher Scientific) at 37 °C for 5 min, then washed with prewarmed RPMI supplemented with 10% FCS. Then, $4 \times 10^5$ KP-OVA or $5 \times 10^6$ MC38-OVA cells were suspended in 20 µl Hanks' balanced saline solution (HBSS) and intradermally injected into the left hind flank skin using a 30 G needle. Tumour volumes were measured using digital callipers and the volume was estimated using the formula: volume = $((width^2 \times length)/2)$. For OVA protein immunization, 25 µg ovalbumin (OVA EndoFit, Invivogen) and 12.5 µg poly(I:C) HMW (Invivogen) were reconstituted in 20 µl PBS and were injected intradermally as above, or subcutaneously in the left footpad.

### Isolation of CD8[+] T cells and adoptive cell transfer

Naive OT-I or wild-type T cells (B6.GFP, CD45.1 or CD45.2) were isolated from spleens and lymph nodes of donor mice with magnetic cell separation (MACS) using mouse CD8a[+] T cell isolation kits (Miltenyi Biotec), according to the manufacturer's instructions. Between 500 and $2 \times 10^3$ cells were injected intravenously in 200 µl HBSS into recipient mice at least 1 day before tumour induction or immunization. For CellTrace Violet labelling, CellTrace Violet dye (Thermo Fisher Scientific) was added at a final concentration of 5 µM to $1 \times 10^7$ cells per ml suspended in 0.1% BSA-containing PBS and incubated at 37 °C for 10 min, before addition of RPMI with 10% FCS to 10× staining volume and incubated further at 37 °C for 5 min. Cells were washed in RPMI before resuspension in HBSS for adoptive transfer.

### In vivo antibody treatment

For blockade of PD-L1 and PD-L2, mice were intraperitoneally injected with 250 µg anti-mouse PD-L1 (10F.9G2, BioXCell) and anti-mouse PD-L2 (TY25, BioXCell) every 2 days starting on day 4 or day 10 after tumour induction. For blockade of PD-1, 250 µg anti-mouse PD-1 (29F.1A12, BioXCell) was injected intraperitoneally into tumour-bearing mice every 3 days starting on day 4 after tumour induction. Rat IgG2a (2A3, BioXCell) and rat IgG2b (LTF-2, BioXCell) isotype antibodies were injected at a dose of 250 µg each as control treatment. In some experiments, PBS was injected intraperitoneally into control animals. No phenotypic differences were observed between IgG-treated and PBS-treated controls.

For depletion of CD4[+] T cells, mice were intraperitoneally injected with 100 µg anti-mouse CD4 monoclonal antibody (GK1.5, BioXCell) 3 days and 1 day before tumour induction.

### XCR1–DTR depletion

For diphtheria toxin depletion of cDC1 in XCR1–DTR mice, 1 µg diphtheria toxin (Millipore Sigma, 322326) diluted in PBS was injected intraperitoneally into tumour-bearing mice on days 5, 6 and 8 after tumour induction.

### Tissue preparation for 3D imaging

Mice were euthanized with sodium pentobarbital through intraperitoneal injection, immediately followed by cardiac perfusion with 10 ml of 1% paraformaldehyde (PFA) prepared from 16% aqueous stock (Electron Microscopy Sciences). Tissues were then collected and fixed for 24 h at 4 °C on a slow shaker in 1 ml fixative buffer (BD Cytofix/Cytoperm diluted 1:4 in PBS; BD Biosciences). Fixed tissues were then washed in 2 ml PBS solution overnight and embedded in 4% UltraPure low-melting-point agarose (Thermo Fisher Scientific) prepared with PBS (cooled to and maintained at 40 °C in a water bath after boiling). Embedded tissues were allowed to solidify on ice. Then, 300 µm agarose-embedded tissue slices were cut using a VT1200S vibrating blade microtome (Leica) at a speed of 0.06 mm s[−1] and amplitude of 1.50. Tissue slices were collected into PBS-filled wells in 24-well plates and stored at 4 °C.

### Antibody conjugation

Custom antibody fluorophore labelling was performed when specific antibody–fluorophore pairs were not commercially available.

Purified antibodies were concentrated using Amicon Ultra 50k MWCO centrifugal filters (Millipore). NHS ester-dye solution (10 mM) was prepared by dissolving NHS ester-dye with DMSO (Sigma-Aldrich). The NHS ester-dye solution and concentrated antibody solution were combined at a 1:9 ratio to yield a final NHS ester-dye concentration of 1 mM, and the solution was allowed to react on ice for 60 min. The reaction mixture was then diluted and concentrated in the centrifugal filters at least three times with PBS at 14,000*g* centrifugation for 5 min to remove unbound dye. The final antibody–fluorophore conjugates were diluted in PBS and stored at 4 °C. Antibody concentration and degree-of-labelling (DOL) were determined with Nanodrop.

## Immunostaining and Ce3D clearing of tissue slices

Tissue slices were incubated in mouse BD Fc Block (1:100; BD Biosciences) prepared in 500 µl BD perm/wash buffer (1:10 diluted in distilled H$_2$O; BD Biosciences) for 24 h at room temperature. For primary antibody staining, blocking buffer was replaced with antibody cocktail prepared in 400 µl perm/wash buffer containing a mixture of fluorophore-conjugated antibodies (1:50 to 1:100 dilution) and incubated for 72 h at room temperature on a slow shaker (60–70 rpm). Stained tissues were then washed in 2 ml perm/wash buffer for another 24 h. After the wash step, post-fixation of the tissue was performed by first replacing the perm/wash buffer with 2 ml PBS for 30 min, followed by post-fixing in 500 µl 1% PFA for 15 min at room temperature, and the samples were then washed again with 2 ml PBS for 30 min. Post-fixed tissues were then transferred into Ce3D solution (see below) for clearing. All steps were performed in the dark in 24-well plates covered in aluminium foil to protect tissues and fluorophores from light exposure.

For samples with secondary antibody staining, following the wash step after primary staining, the perm/wash buffer was replaced with 500 µl perm/wash buffer containing fluorophore-conjugated secondary antibodies (1:500 dilution) and incubated for another 48 h, washed for 24 h and then post-fixed as described above. A list of the antibodies used is provided in Supplementary Table 1.

For tissue clearing, Ce3D tissue clearing solution was prepared as described previously[62] with modifications. In brief, a 10 ml clearing solution was prepared using 5.5 ml 40% (v/v) *N*-methylacetamine (diluted with PBS; Sigma-Aldrich) and 10 g Histodenz (Sigma-Aldrich) without Triton X-100 detergent. The tube containing Ce3D solution was then incubated in a heated shaker at 37 °C and 150 rpm for at least 1 h until the solid powder was fully dissolved. The prepared Ce3D solution was then stored on a slow shaker at room temperature, wrapped in aluminium foil to protect from light exposure. The refractive index of Ce3D was measured using a digital refractometer, with an expected value of about 1.52.

For tissue clearing, small chambers containing around 700 µl Ce3D solution were prepared on a glass slide (SuperFrost Plus, VWR) by stacking two CoverWell incubation chambers (0.5 mm depth with a 13 mm chamber diameter; Grace Biolabs) on top of an adhesive SecureSeal imaging spacer (Grace Biolabs) to prevent leakage between the silicone chamber and the glass slide. Post-fixed tissue slices were transferred into Ce3D-filled chambers and sealed with a glass coverslip. Tissues were cleared for at least 48–72 h on a gentle shaker (60–70 rpm) at room temperature before imaging.

Before imaging, a shallow imaging chamber was created by stacking two layers of adhesive SecureSeal imaging spacers (2 × 0.13 mm depth) on a SuperFrost Plus glass slide. Up to 4 lymph node tissue slices were placed into an imaging chamber of 20 mm × 20 mm (spacers of 20 mm circular chamber diameter trimmed to a square using a scalpel blade), arranged in a 2 × 2 grid, for batch imaging of tissues from the same experiment. The shallow chamber was then filled with Ce3D solution and gently sealed with a no. 1.5 glass coverslip (VWR).

## Laser-scanning confocal microscopy

Volumetric images were acquired using an inverted Leica Stellaris or upright TCS SP8 X confocal microscopes (Leica) equipped with a pulsed white-light laser and four tuneable spectral hybrid detectors. Images were acquired with a ×20 multi-immersion objective (with correction collar adjusted to oil immersion), numerical aperture (NA) = 0.75 and a working distance of 0.66 mm, and were captured at a digital zoom of 1.5 (0.361 µm *xy* pixel resolution) and 2 µm *z* step over the full thickness of the tissue slice. Excitation of CellTrace Violet and eFluor 450 was performed using a fixed 405 nm laser line. Twelve- and 16-bit images were typically acquired, although some experimental datasets were acquired as 8-bit images. Image tiles were taken with 5% overlap and merged using the Leica LAS X Navigator application.

For high-resolution imaging of protein co-localization, a digital zoom factor of 2.0–2.5 (0.227–0.284 µm *xy* pixel resolution) and 1.5 µm *z* step were used. For anti-NFAT1 stained tissues, images were acquired with a digital zoom factor of 2.0 (0.284 µm *xy* pixel resolution) with 2 µm *z* step to provide sufficient lateral resolution for distinguishing cytoplasmic and nuclear NFAT localization.

## Chemical bleaching of fluorophores

For imaging experiments involving IBEX iterative staining[63], imaged cleared tissue slices were first returned to wells filled with PBS for removal of clearing reagent until the tissue appearance became opaque. Tissues were then transferred into 2 ml perm/wash buffer and washed for 24 h at room temperature, then transferred into new perm/wash buffer-containing wells for two subsequent washes (24 h each) to minimize retention of clearing reagent within the tissues.

To chemically bleach fluorophores with lithium borohydride (LiBH$_4$, STREM Chemicals, 93-0397), bleaching solution was prepared at 1 mg ml$^{-1}$ LiBH$_4$ concentration in distilled water, as described previously[63]. Tissue slices were then washed in PBS for 30 min at room temperature, replaced with LiBH$_4$ solution for 45 min and placed ~30 cm from a fluorescent light source. Bleached tissues were then washed again in PBS for 30 min, and finally transferred into a new well containing antibody cocktail solution for subsequent staining steps.

## Image analysis

**Image preprocessing, segmentation and histocytometry quantification of single cells.** Detailed steps of the image processing and analysis pipeline are provided in the Supplementary Methods. In brief, a Python-based computational pipeline was developed and optimized to enable distributed processing of large volumetric datasets on the NIH HPC Biowulf cluster. Raw image data were preprocessed to compensate for spectral spillover and correction for intensity attenuation along the *z* axis.

Single-cell segmentation was performed with a modified version of Stardist3D[64] on nuclei or with Cellpose[65] on membrane markers. A small image region was cropped from a representative image and manually annotated as a training dataset, and the custom trained model was used for segmentation of all image datasets from the same experiment. In most cases, Ki-67 nuclear stains were used as the segmentation channel. The nuclear masks were then morphologically dilated to encompass the membrane/cytoplasmic region of the cells, and subtraction of both masks generated a new membrane/cytoplasmic mask. The mean intensity for each channel was determined by summing the masked voxel intensities divided by the sum of all mask voxels for every cell. Output arrays containing both cell coordinates (*x*, *y*, *z*) and mean marker intensities were exported for downstream analyses.

For histocytometry gating of single cells, Python-based scripts were used to visualize marker intensities as two-dimensional histocytometry plots and for gating on further subsets. In brief, scatter plots showing the distribution of mean intensities of protein markers were visualized using the mpl_scatter_density package, and polygonal-shaped gates were drawn using the Polygon class objects from the package shapely, specifying the vertex coordinates of the polygons. The contains method from the Polygon class was used to filter the cells contained within the

gates for downstream analysis and visualization. Donor OT-I cells were selected on the basis of Ki-67 and GFP/CD45.1 expression. For gating polyclonal activated CD8[+] T cells, Ki-67 and CD8b expression was used. Further subsets were generated based on TCF-1 and PD-1 expression. The spatial distribution of each subset was then visualized using Imaris v.10.0 (Bitplane).

Quantification of marker expression for each subset was visualized with violin plots using the Python-based seaborn visualization library. Principal component analysis was performed using the Python-based scikit-learn library on z-score-normalized marker intensity and plotted using matplotlib library to visualize relative marker expression.

For quantification of TCF-1 expression level among OT-I cells in close proximity to dense cDC1 region, the threshold for the cDC1-density[high] gate was determined using the mean + 1 s.d. of cDC1-cell density normalized to the maximum cDC1 cell density value of the OT-I cell population (see the 'cDC1 cell density' section). OT-I cells above the cDC1-dense threshold (mean + 1 s.d.) were then gated (cDC1-density[high]) and their relative TCF-1 expression values (normalized to naive T cell expression) were displayed as violin plots. Naive TCF-1 expression was determined by selecting a small patch (about 512 × 512) of the lymph node T cell zone densely populated with naive T cells expressing a high level of TCF-1 and calculating the mean intensity of the TCF-1 channel. Further gating on the TCF-1[high] subset was performed by selecting cells with a TCF-1 value (normalized to naive T cells) of >0.7.

**cDC1 cell density.** For determining spatial proximity to dense cDC1 region, a Gaussian smoothing filter of bandwidth $\sigma = 3.6$ μm (10 pixels) was applied to the XCR1 channel for each z slice. The mean intensity obtained from a cell's nuclear mask yields the spatial density of cDC1 at the cell's centroid.

**NFAT quantification.** Single-cell segmentation as well as nuclear and membrane/cytoplasmic masks were generated as described in the 'Image preprocessing, segmentation and histocytometry quantification of single cells' section. The middle z slice of each cell (the largest cross-section in a 3D cell volume) was used to calculate the ratio between the means of nuclear and membrane/cytoplasmic NFAT intensity. Low-NFAT-expressing cells could lead to false positives and were gated out from analysis. Quantification of the mean cDC1 cell density (normalized) as well as the proportion of cells above the cDC1-dense threshold (determined at mean = 1.0 of the normalized cDC1 cell density described above) was then performed on the remaining NFAT ratio[+] and NFAT ratio[−] subsets.

**Kernel density estimation.** 3D image coordinates of cells of interest were converted to world coordinates by multiplying their voxel dimensions. Kernel density estimation was then performed using the TreeKDE module from KDEpy library, with an isotropic Gaussian kernel (bandwidth $\sigma = 6$ μm) across a grid system of 10 μm interval in each axis. Weights were set to normalized protein marker expression (for example, PD-1, SLAMF6) of each cell. To generate a kernel density map for visualization, the density values were summed over a selected z-axis range comprising 80 μm thickness (out of ~300 μm full volume thickness) to reduce clutter. A contour map was then generated using the contourf function in matplotlib library set to a perceptually uniform colourmap for visualization of the probability density on a two-dimensional plot.

**Heat-map visualization.** Each parameter (protein expression, spatial density) was first standardized to obtain a z score for each cell. The cell population was then subdivided and sorted based on the manual gating strategy defining SLAMF6[+] $T_{SL}$, SLAMF6[−] $T_{SL}$ and TCF-1[−] $T_{eff}$ cells. A perceptually uniform colourmap was applied to display the relative z score of each parameter. Note that both screen and print resolutions are not sufficient to enable discrimination of single cells within the heat map, which contains the total population of OT-I recovered from the lymph node tissue slice (>65,000 cells in the dataset shown in Fig. 1e).

**Visualization.** For visualization of protein markers expressed by gated cells, segmented labels (nuclear masks) of the gated cells were processed with the morphological dilation tool (radius = 6) from scikit-image library to generate a new mask encompassing the cytoplasmic/membrane region of the cells. Individual image channels were then multiplied with the mask to generate a new image displaying only protein marker intensities masked within the cells of interest. These masked images were then visualized using Imaris v.10.0 (Bitplane). To visualize the relative protein expression level, perceptually uniform colourmaps were used and the min/max scaling (contrast and brightness) of each channel was individually adjusted to avoid under- and over-saturation of the marker intensity.

**Animation.** For animation of 3D imaging datasets, the napari-animation library was used and an animation script based on the instructions provided by the developers was made to generate keyframes specifying the camera positions and angles, image layer's colourmap, adjustments of contrast/brightness, as well as clipping planes for animating transition between layers and for focusing on thin cross-sections of the imaging volume. The output video file was further processed and edited in Davinci Resolve v.18.6 (Blackmagic Design) to include annotations with text and graphic items.

### Tissue preparation for flow cytometry

Lymphocytes were isolated from lymph nodes and spleens and made into single-cell suspensions using a syringe plunger and 100 μm or 70 μm cell strainers (MACS SmartStrainer, Miltenyi Biotec). Around $3 \times 10^6$ cells were used in subsequent staining steps for flow cytometry analysis. Dendritic cell isolation was performed as described previously[66]. In brief, lymphoid tissues were sliced into small fragments using a scalpel blade and incubated in a digestion mix of collagenase type III (Worthington, 1 mg ml[−1]) and DNase I (20 μg ml[−1]) and vigorously mixed for 25 min at room temperature, followed by addition of 0.1 M EDTA solution at 1/10 digestion volume for 5 min to dissociate lymphocytes from dendritic cells. Tissue debris was filtered out by passing the cell suspension through a 70 μm nylon mesh.

T cells from skin tumour samples were isolated as described previously[67]. In brief, a 1 cm × 1 cm tumour-containing skin patch was collected into collagenase type III (Worthington, 3 mg ml[−1]), finely chopped with scissors and incubated at 37 °C for 90 min before pressing through 70 μm cell strainers. For spleen and tumour samples, cells were also treated with red blood cell lysis buffer before staining. Cell counts in lymph node and spleen were determined using an automated Cellometer T4 cell counter (Nexcelom Bioscience).

### Flow cytometry

For detection of polyclonal OVA-specific CD8[+] T cells, cells were first incubated with phycoerythrin-conjugated H-2K[b]–SIINFEKL tetramer (1:100, NIH Tetramer Core) for 20 min at 37 °C, washed, followed by cell surface marker staining for 25 min at 4 °C. Mouse BD Fc Block (1:200, BD Biosciences) was also included during the cell surface marker staining step. A fixable LIVE/DEAD near-infrared staining dye was used for determining cell viability. For detection of intracellular proteins, stained cells were further treated with fixative and stained for antibodies against intracellular proteins using FOXP3/transcription factor staining buffer kit according to the manufacturer's protocol (eBioscience). For tumour samples, CountBright Plus Absolute beads (Thermo Fisher Scientific) were added before sample acquisition. The samples were acquired using the BD Fortessa (BD Biosciences) system and analysed using Flowjo 10 (TreeStar). A list of the antibodies used is provided in Supplementary Table 1.

## Imputed affinity index

Tetramer binding is an estimation of TCR avidity (based on the multivalent binding strength of the TCR–pMHC complex), which in itself is a function of TCR affinity (monovalent TCR–pMHC binding), the amount/density of expressed TCRs, as well as the presence of co-receptors (such as CD8) that can potentially influence binding ability in a solution-based measurement.

To estimate the TCR affinity of tetramer-stained cells, an imputed affinity index was derived by dividing the tetramer staining intensity by the TCR (CD3) staining intensity (tetramer/CD3 ratio) of the single cells. This is possible because tetramers elute from labelled T cells in the wash buffer during incubation steps in reasonable proportion to affinity, in particular the off-rate of the interaction[26,68–71]. When normalized to surface TCR expression to account for avidity differences among T cells, we can use the measured staining as a proxy for this off-rate and, therefore, affinity. Cells were post-fixed after the surface marker staining step, before intracellular antibody staining, to minimize continued elution of tetramers over time. Further controls for secondary factors affecting this off-rate such as CD8 co-receptor binding, as well as TCR level/density, were performed through direct comparison with monoclonal OT-I TCR transgenic cells under control and checkpoint treatment conditions as described in the main text.

Given that tetramer binding varies from batch to batch and is more sensitive to incubation conditions compared with antibody staining, a $z$-score for each subset from each animal was derived from the log-affinity index (log[tetramer/CD3 ratio]) of the tetramer-stained cells, standardized with reference to SLAMF6$^+$ $T_{SL}$ cells of the control IgG-treated group pooled from the same experimental cohort. The mean of the $z$ score, mean $z$(log[affinity index]), therefore estimates the distribution of TCR affinity of each subset and treatment group in relation to SLAMF6$^+$ $T_{SL}$ cells of the control IgG-treated group.

A normalized ratio was also derived from the imputed TCR affinity indices of tetramer-stained cells, by normalizing the affinity indices of each subset from each animal to the reference affinity index of SLAMF6$^+$ $T_{SL}$ cells from control IgG-treated group pooled from the same experimental cohort. This normalized ratio, mean affinity index, estimates the relative TCR binding affinity of each subset and treatment group in relation to SLAMF6$^+$ $T_{SL}$ cells of the control IgG-treated group. When specified, other reference subsets (for example, day 6 TCF-1$^+$ $T_{SL}$ cells) were also used for normalization.

To plot the mean CD8a expression versus log[affinity index] of OT-I and polyclonal T cells in Extended Data Fig. 7j, the mean CD8a expression values and the mean imputed log-affinity indices were obtained from OVA–tetramer$^+$SLAMF6$^+$ $T_{SL}$ cells of each sample and normalized to the reference mean CD8a value and mean log-affinity index, respectively, of all OVA–tetramer$^+$SLAMF6$^+$ $T_{SL}$ cells pooled from control IgG-treated animals from the same experimental cohort. As the mean affinity indices of SLAMF6$^+$ $T_{SL}$ cells in some animals in the anti-PD-L1/2-treated group were very low, which led to negative log-transformed values, the imputed affinity indices of all OVA–tetramer$^+$ cells from the same experimental cohort (pooled from both control IgG-treated and anti-PD-L1/2-treated samples) were first translated with the formula: affinity index = affinity index + (1 − min(affinity index$_{pooled}$)) to ensure that log-transformed imputed affinity indices remain positive and to allow for normalization with the control IgG-treated SLAMF6$^+$ $T_{SL}$ cells.

## Modelling of TCR affinity and cell states

In the course of these studies, we observed a number of outlier responses among the treated individual mice. Such results are not unexpected based on the known variation in TCR repertoire between inbred animals[13]. A linear modelling analysis was conducted to determine whether these outlier events affected the conclusions of our flow cytometry studies.

The TCR affinity of single cells was quantified as the log-ratio of tetramer staining to CD3 staining (log[tetramer/CD3 ratio]). Within each experiment, these log-ratios were standardized with reference to the control IgG-treated SLAMF6$^+$ $T_{SL}$ cells, to emphasize the differences in affinity between cell states and treatment groups. The three major subsets of OVA–tetramer$^+$CD8$^+$ T cells: SLAMF6$^+$TCF-1$^+$ $T_{SL}$, SLAMF6$^-$TCF-1$^+$ $T_{SL}$ and TCF-1$^-$ $T_{eff}$ cells were defined as distinct cell states. Standardization was performed separately for each experiment, to control for experiment-to-experiment staining variability, because affinity measurements differed substantially across experiments. Cellular affinity was then regressed on cell state, treatment and their interaction using a Bayesian multilevel model with a Gaussian likelihood. Hyperparameters for the effect of mouse identity on the intercept and on the cell state slopes were included to account for mouse-to-mouse variability. Predictions from this model were simulated without mouse-to-mouse variability to isolate the effects of cell state and treatment on the expected distributions of cellular affinity.

In a second model, the number of cells in each cell state from each animal was log-transformed. These log-counts were then standardized across all groups of cells. Cell counts were regressed on cell state, treatment and their interaction using a Bayesian multilevel model with a Gaussian likelihood. Hyperparameters for the effect of mouse identity on the intercept and on cell state slopes were included to account for mouse-to-mouse variability. Predictions from this model were simulated without mouse-to-mouse variability to isolate the effects of cell state and treatment on the expected cell counts.

The simulated predictions from the first model provided expected distributions of log-affinity relative to the mean of control IgG-treated SLAMF6$^+$ $T_{SL}$ cells for each combination of cell state and treatment. The fraction of cells with log-affinity greater than the mean of control IgG-treated SLAMF6$^+$ $T_{SL}$ cells was simply the fraction of cells above 0 in each distribution. These fractions were multiplied by the expected total number of cells for each combination of cell state and treatment, provided by the simulated predictions of the second model. This gave the expected number of cells greater than the mean log-affinity of control IgG-treated SLAMF6$^+$ $T_{SL}$ cells (above average TCR affinity threshold) for each combination of cell state and treatment.

## Statistics

Statistical tests were performed in Prism 9.0 software (GraphPad). Data analyses were performed using unpaired two-tailed Student's $t$-tests or one-way ANOVA with Tukey's post hoc multiple comparison test, as specified in the text or figure legends. $P < 0.05$ was considered to be significant.

## Reporting summary

Further information on research design is available in the Nature Portfolio Reporting Summary linked to this article.

## Data availability

The representative imaging dataset shown in Fig. 1 can be accessed at the Zenodo repository[72] (https://zenodo.org/records/15599322). Owing to the large file sizes, other imaging datasets will be made available on request from the corresponding authors. Source data are provided with this paper.

## Code availability

Python code generated for imaging data analysis is available at GitHub (https://github.com/jlhor/3d-imaging-pipeline).

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

**Acknowledgements** We thank T. Kaisho for the provision of transgenic mouse strains; P. L. Schwartzberg (NIAID, NIH) for discussions; and all of the members of the Lymphocyte Biology Section for their feedback and support. We acknowledge the use of computational resources of the NIH HPC Biowulf cluster for imaging data processing and analysis. This work was supported by the Division of Intramural Research of NIAID, NIH and Center for Cancer Research, NCI, NIH. This work was supported by the Division of Intramural Research of NIAID, NIH and the Center for Cancer Research, NCI, NIH. Work from C.J.'s laboratory was supported by NIH grants (NIH R00 CA226400, NIH DP2 CA280834), Emerson Collective Cancer Research Fund, W.W. Smith Charitable Trust and a Pew-Stewart Scholarship for Cancer Research. C.Z. is additionally supported by ASCO awards, a SITC-AstraZeneca Immunotherapy in Lung Cancer (Early Stage NSCLC) Clinical Fellowship Award and the NIH Bench-to-Bedside and Back Program (BtB).

**Author contributions** J.L.H. and R.N.G. conceived the study, reviewed all data and wrote the manuscript. J.L.H. designed and performed all experimental work, analysed data and developed computational pipelines for imaging analyses. E.C.S. designed and performed the TCR affinity modelling analyses. W.S., A.R.-W., Q.D., C.Z. and C.J. generated tumour lines. L.V. performed custom dye conjugation of antibodies.

**Competing interests** The authors declare no competing interests.

**Additional information**
**Correspondence and requests for materials** should be addressed to Jyh Liang Hor or Ronald N. Germain.

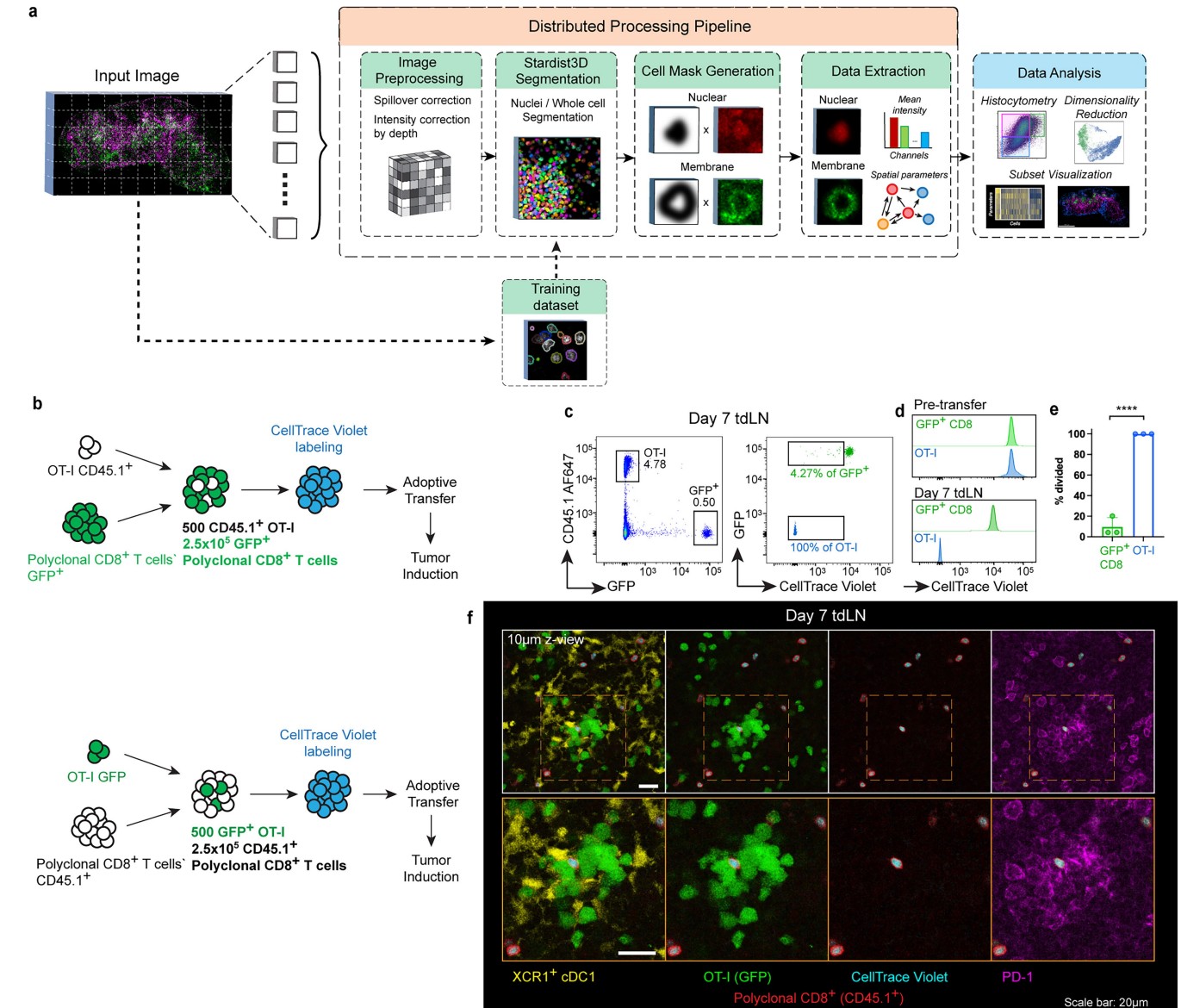

**Extended Data Fig.1 | Imaging data processing pipeline and tracking of antigen-specific CD8+T cell proliferation in the tdLN. a**, Schematics of a distributed image processing and analysis pipeline that enables fast and accurate single-cell level analysis of large 3D volumetric datasets. The raw input image is divided into sub-blocks, which are processed piece-wise in parallel to perform image corrections, cell segmentation, masks generation and data extraction. The training dataset for segmentation is generated from semi-automated label annotation from a small, cropped region of a representative image from the same experiment. Further downstream analyses are performed with extracted data. **b**, Experimental scheme of co-labelling of OT-I and polyclonal CD8+ T cells isolated from naïve mice with CellTrace Violet proliferation dye, in reciprocal formats: OT-I.CD45.1 and CD8. GFP (top: **c-e**); OT-I.GFP and CD8.CD45.1 (bottom: **f**). Cells were co-transferred into recipient mice at 1:500 ratio. **c**, Left, flow cytometry plot of donor cell

populations recovered from Day 7 tdLN and right, their corresponding CellTrace Violet dye intensity. Gates denote populations with diluted CellTrace Violet signal (CTV^low). **d**, Histograms showing CellTrace Violet signal measured from pre-transfer cell suspension (top) and those recovered from Day 7 tdLN (bottom). **e**, Quantification of proportion of divided cells (CTV^low) from Day 7 tdLN as gated in (**c**). **f**, Representative cross-section images showing clustering PD-1+ OT-I.GFP cells (green) and polyclonal non-antigen specific CD8+ T cells (CD45.1, red) in Day 7 tdLN. Proliferation dye (cyan) has been fully diluted among the antigen-specific OT-I.GFP population. Dashed box indicates the area shown in the close-up images (bottom). Data in (**b**-**f**) are representative of 2 independent experiments (in reciprocal formats), with n = 4 for both imaging and flow cytometry experiments. Error bars: means ± s.d. ****p < 0.0001; unpaired two-tailed *t*-test (**e**).

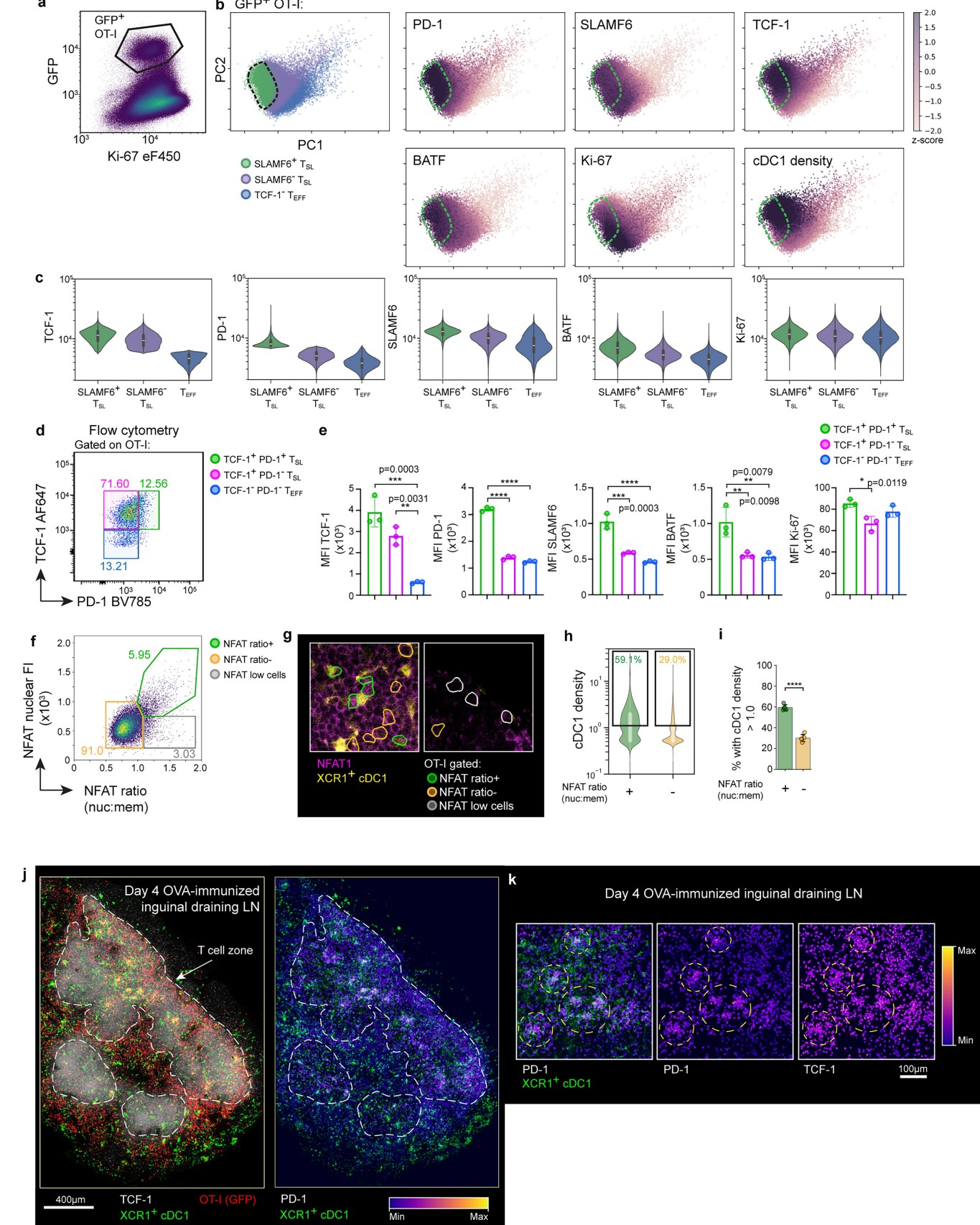

**Extended Data Fig. 2** | See next page for caption.

**Extended Data Fig. 2 | Phenotypic and functional state analysis of tumour antigen-specific OT-I cells in Day 8 tdLN. a**, Representative histo-cytometric plot showing gating strategy of OT-I cells in Day 8 tdLN based on GFP and Ki-67 co-expression. **b**, PCA plots of colour-coded OT-I subsets based on TCF-1 and PD-1 expression (Fig. 1c) (left panel) and the standardized expression for each marker (right panels). Dotted region indicates TCF-1+PD-1+SLAMF6+$T_{SL}$. **c**, Protein expression levels for each OT-I subset as defined in Fig. 1c. Data shown are representative of 2 independent experiments (n = 4). **d**, Flow cytometric plot of OT-I cells recovered from Day 7 tdLN with subsets gated similarly to histo-cytometric data from Fig. 1c. **e**, Protein expression levels for each OT-I subset as gated in (**d**). **f-i**, Quantification of NFAT localization on OT-I. **f**, Gating strategy for NFAT localization (nuclear:membrane) ratio, with low NFAT expressing but positive ratio cells filtered out as false positive events (grey). **g**, Close up representative images showing examples of OT-I cell boundaries gated from (**f**) and the actual NFAT signal. **h, i**, Quantification of OT-I subset proximity to dense cDC1 regions based on cDC1 spatial density. Gates denote proportion of cells with cDC1 density value above the mean threshold (normalized value = 1.0). Data pooled from 2 independent experiments (n = 5). **j-k**, Day 4 inguinal draining lymph node (LN) from mice transferred with 500 OT-I.GFP and immunized with OVA/poly(I:C) on the flank skin. **j**, Cross-section image showing the demarcation of T cell zone (dashed boundaries) based on TCF-1 staining (left) and PD-1 intensity (masked on OT-I.GFP cells) (right). **k**, Close-up images showing co-localization of PD-1+ OT-I cells with cDC1, PD-1 intensity only and TCF-1 intensity only (both masked on OT-I cells) in the order from left to right. Data are representative of 2 independent experiments (n = 4). Box plots: median and 1$^{st}$/3$^{rd}$ quartiles, whiskers: 1.5x inter-quartile range (**c, h**). Error bars: means ± s.d. *$p < 0.05$, **$p < 0.01$, ***$p < 0.001$, ****$p < 0.0001$; one-way ANOVA with Tukey's multiple comparisons (**e**) and unpaired two-tailed *t*-test (**i**).

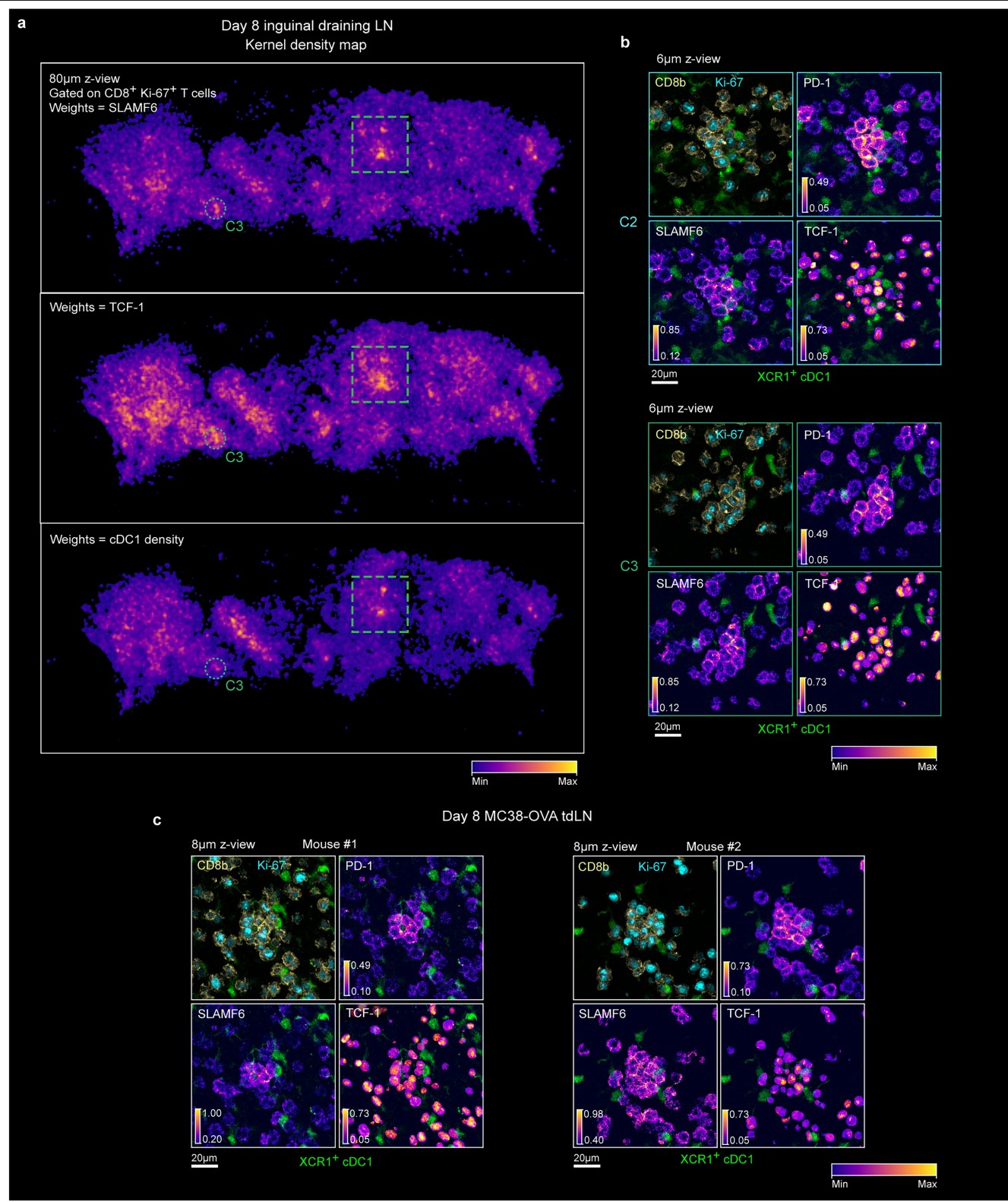

**Extended Data Fig. 3 | Distinct foci of polyclonal CD8+SLAMF6+T$_{SL}$ in Day 8 tdLN. a**, Kernel density map of activated polyclonal CD8+ T cells in Day 8 tdLN. Density was estimated using a Gaussian kernel of 6 μm bandwidth, weighted on SLAMF6 (top), TCF-1 (middle) or cDC1 density (bottom). Green dashed box indicates close-up region corresponding to Fig. 2b,c, while dotted circle (C3) corresponds to close-up images shown in (**b**) **b**, Close-up images of C2 (top) and C3 (bottom) and the relative individual stained protein markers. **c**, Representative close-up images of activated polyclonal CD8+ T cells in Day 8 tdLN harvested from MC38-OVA tumour-bearing mice. Data shown are representative of two independent experiments (n = 5 for KP-OVA, n = 4 for MC38-OVA).

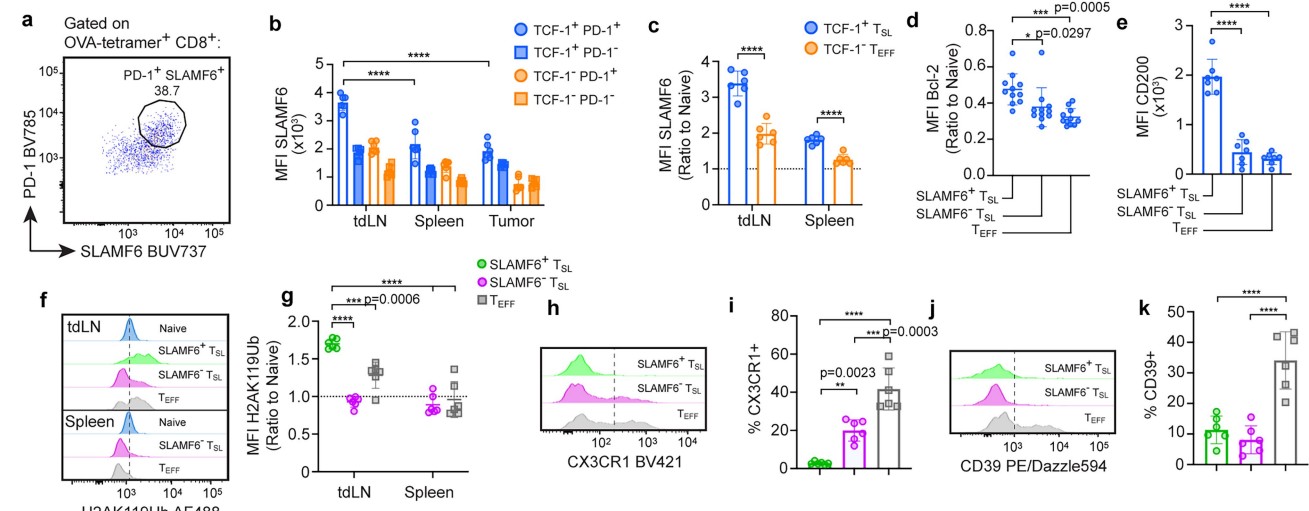

**Extended Data Fig. 4 | Characterization of SLAMF6+T$_{SL}$ across tissues.**
**a**, Flow cytometry plot showing correlation of PD-1 and SLAMF6 protein expression on OVA-tetramer+ CD8+ T cells from Day 10 tdLN. **b**, Mean SLAMF6 fluorescence intensity on different subsets of OVA-tetramer+ cells across different tissues. SLAMF6 expression was particularly enriched in TCF-1+ T$_{SL}$ residing in tdLN, as shown by the normalized SLAMF6 expression (normalized to naïve TCF-1+PD-1-CD8+ T cells) in (**c**). **d**, Bcl-2 expression (normalized to naïve TCF-1+PD-1-CD8+ T cells) and **e**, mean fluorescence intensity of CD200 expressed on each subset of OVA-tetramer+ CD8+ T cells. Data from 2 independent experiments (n = 6) for (**b-c**), 3 independent experiments (n = 11) for (**d**), and 2 independent experiments (n = 7) for (**e**). **f-g**, Relative expression of

H2AK119Ub by different subsets of OVA-tetramer+ cells from different tissues, shown in representative histograms (**f**) and as mean fluorescence intensities (normalized to naïve TCF-1+PD-1-CD8+ T cells) (**g**). **h-k**, Quantification of CX3CR1 and CD39 expressions on OVA-tetramer+ subsets in the tdLN. Histogram panels showing the relative expression across subsets (**h, j**) and the proportions of positive gated events (**i, k**) based on thresholds shown as dashed lines in the histogram panels. Data from 2 independent experiments (n = 6). Error bars: means ± s.d. *p < 0.05, **p < 0.01, ***p < 0.001, ****p < 0.0001; unpaired two-tailed *t*-test (**b, c**) and one-way ANOVA with Tukey's multiple comparisons for all others.

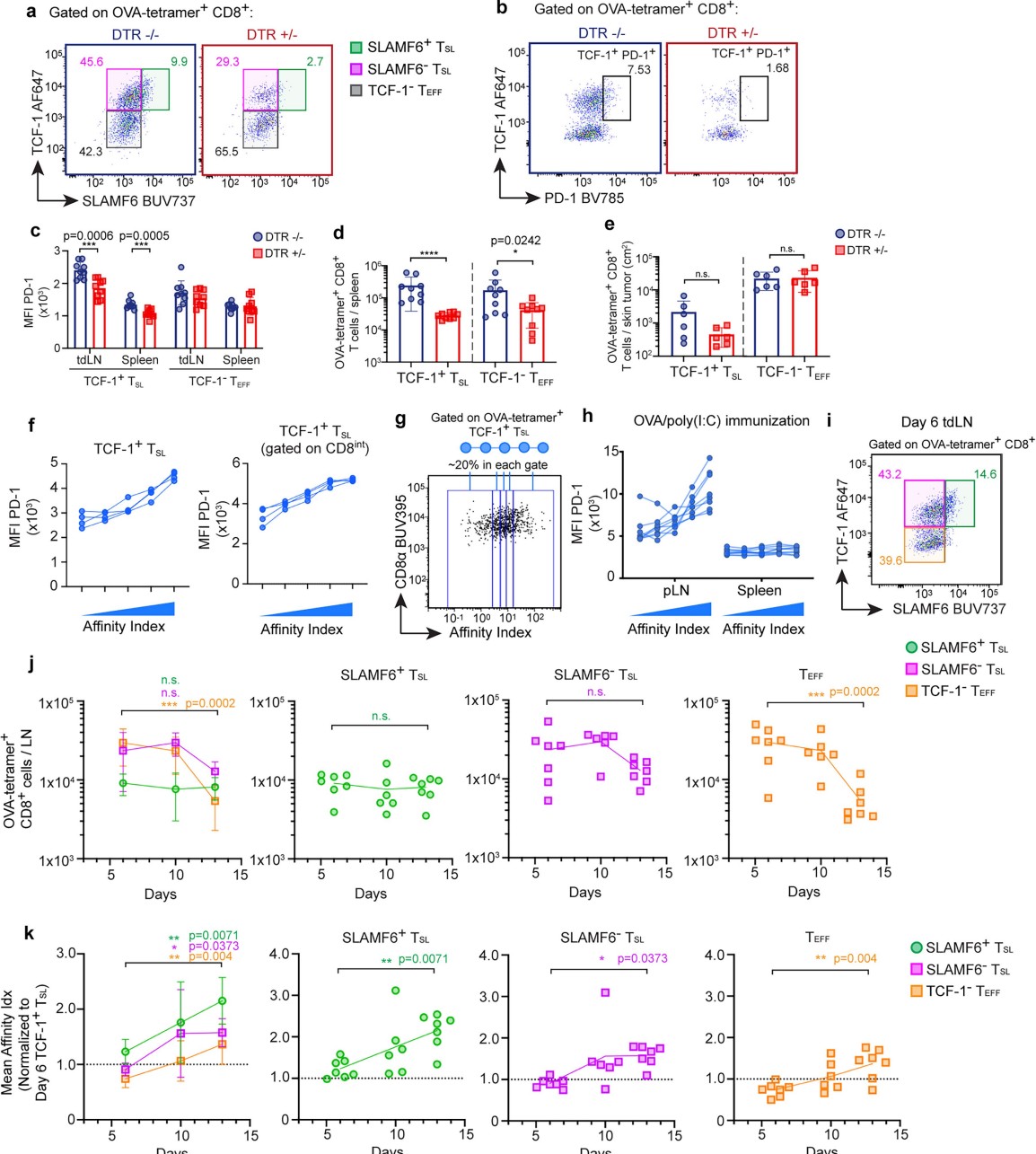

**Extended Data Fig. 5 | Late antigen presentation niche driven expansion and TCR affinity evolution of $T_{SL}$ in the draining lymph node. a**, Gating strategy to quantify SLAMF6$^+$ $T_{SL}$ subset in Fig. 3d. **b–e**, Representative flow cytometric plots showing the loss of PD-1+ OVA-tetramer+ CD8+ T cells after diphtheria toxin depletion of cDC1 (**b**) and quantification of mean PD-1 intensity (**c**) in Day 10 tdLN. **d–e**, Quantification of OVA-tetramer+ CD8+ T cells recovered from Day 10 spleens (**e**) and 1 cm$^2$ tumour-containing skin (**g**). Data from 3 independent experiments (n = 9 per group). **f**, Mean fluorescence intensity of PD-1 expression on OVA-tetramer+ TCF-1+ $T_{SL}$ across different bins of imputed affinity indices (as defined in Fig. 3e), gated on the entire sub-population (left panel) or along a band of cells with intermediate CD8α expression (~50% of the sub-population, right panel). **g**, Flow cytometry plot showing gating strategy of binning OVA-tetramer+ TCF-1+ $T_{SL}$ in Day 5 draining popliteal lymph node of OVA immunized mice, based on imputed TCR affinity indices. Mean fluorescence intensity of PD-1 in the dLNs and spleens is shown in (**h**). Data pooled from 2 independent experiments (n = 8). **i**, Gating strategy for OVA-tetramer+ CD8+ T cell subsets in (**j, k**). The absolute count of each subset (**j**) and the average imputed TCR affinity indices (normalized to Day 6 TCF-1+ $T_{SL}$) (**k**) shown combined (left) or as individual subsets. Data from 2 independent experiments (n = 7 for Day 6, Day 13; n = 6 for Day 10). Error bars: means ± s.d. *p < 0.05, **p < 0.01, ***p < 0.001, ****p < 0.0001; unpaired two-tailed *t*-test for (**c-e**), one-way ANOVA with Tukey's multiple comparisons for all others.

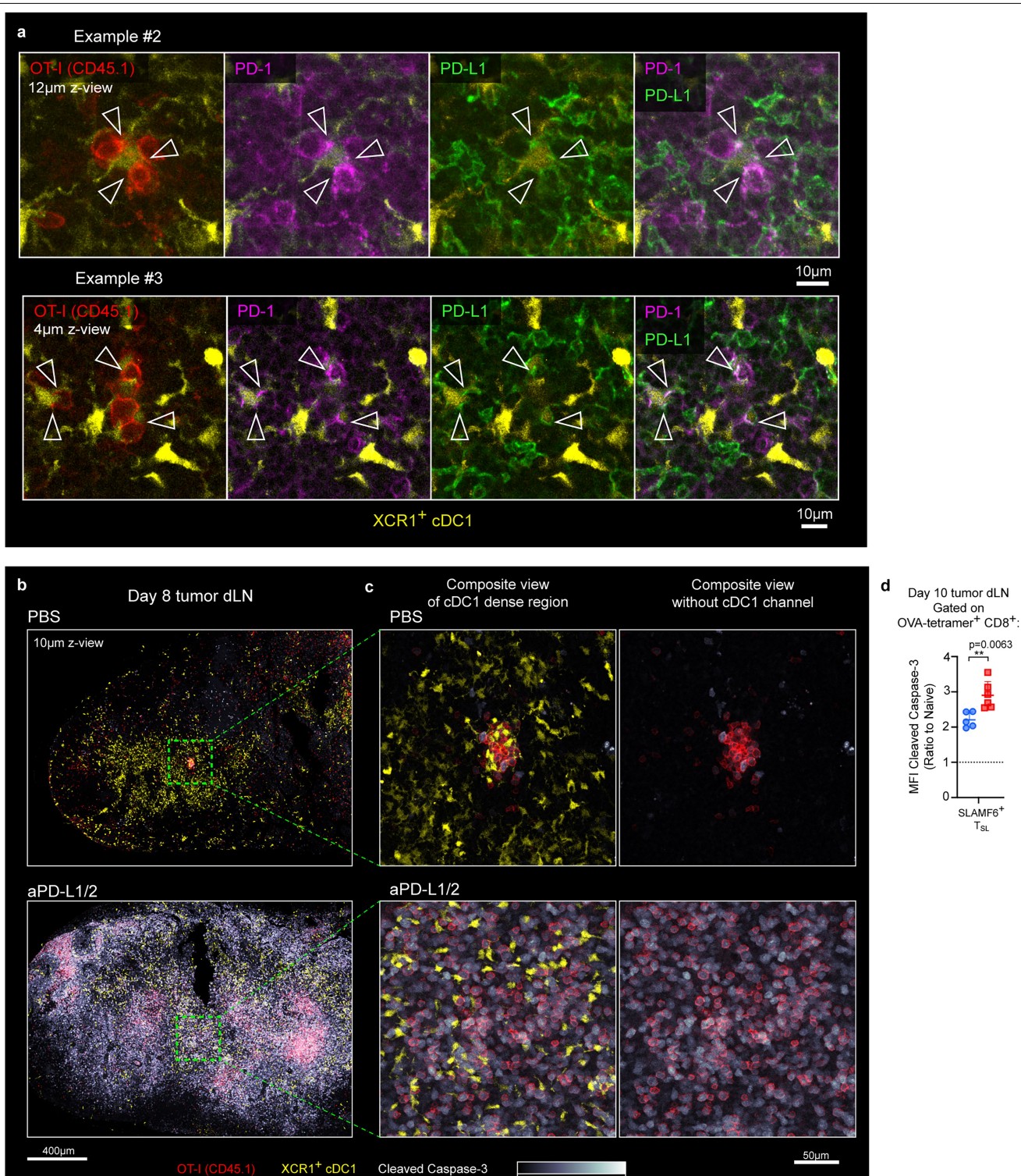

**Extended Data Fig. 6 | PD-1 checkpoint blockade induces apoptosis of CD8+T cells in the tdLN. a**, Multi-panel images showing thin cross-sections (12 μm and 4 μm z-thickness) of two examples of clustering OT-I cells and the co-localization of PD-1 (magenta) on OT-I.CD45.1 (red) with PD-L1 (green) expressed on cDC1 (yellow), as in Fig. 4b. Arrows indicate polarization and punctate microcluster formation of PD-1 and their co-localization with PD-L1 blending into white pixels. Data representative of at least 2 independent experiments (n = 4). **b**, Cross-section of Day 8 tdLN from mice treated with PBS or anti-PD-L1/PD-L2 blocking antibodies following the experimental scheme from Fig. 4c. Cleaved caspase-3 staining indicates increased apoptosis of the lymph node cells. Dotted box denotes the close-up of cDC1-dense region shown in (**c**), displayed with (left) or without cDC1 channel (right). Data are representative of 3 independent experiments (n = 5 for PBS; n = 6 for aPD-L1/2). **d**, Relative expression of cleaved caspase-3 of OVA-tetramer+ SLAMF6+ $T_{SL}$ recovered from Day 10 tdLN treated with PBS (blue) or anti-PD-L1/2 (red) as measured by flow cytometry. Expression level is normalized to CD44- TCF-1+ PD-1- naïve CD8+ T cell population. Error bars: means ± s.d. **p < 0.01; unpaired two-tailed $t$-test.

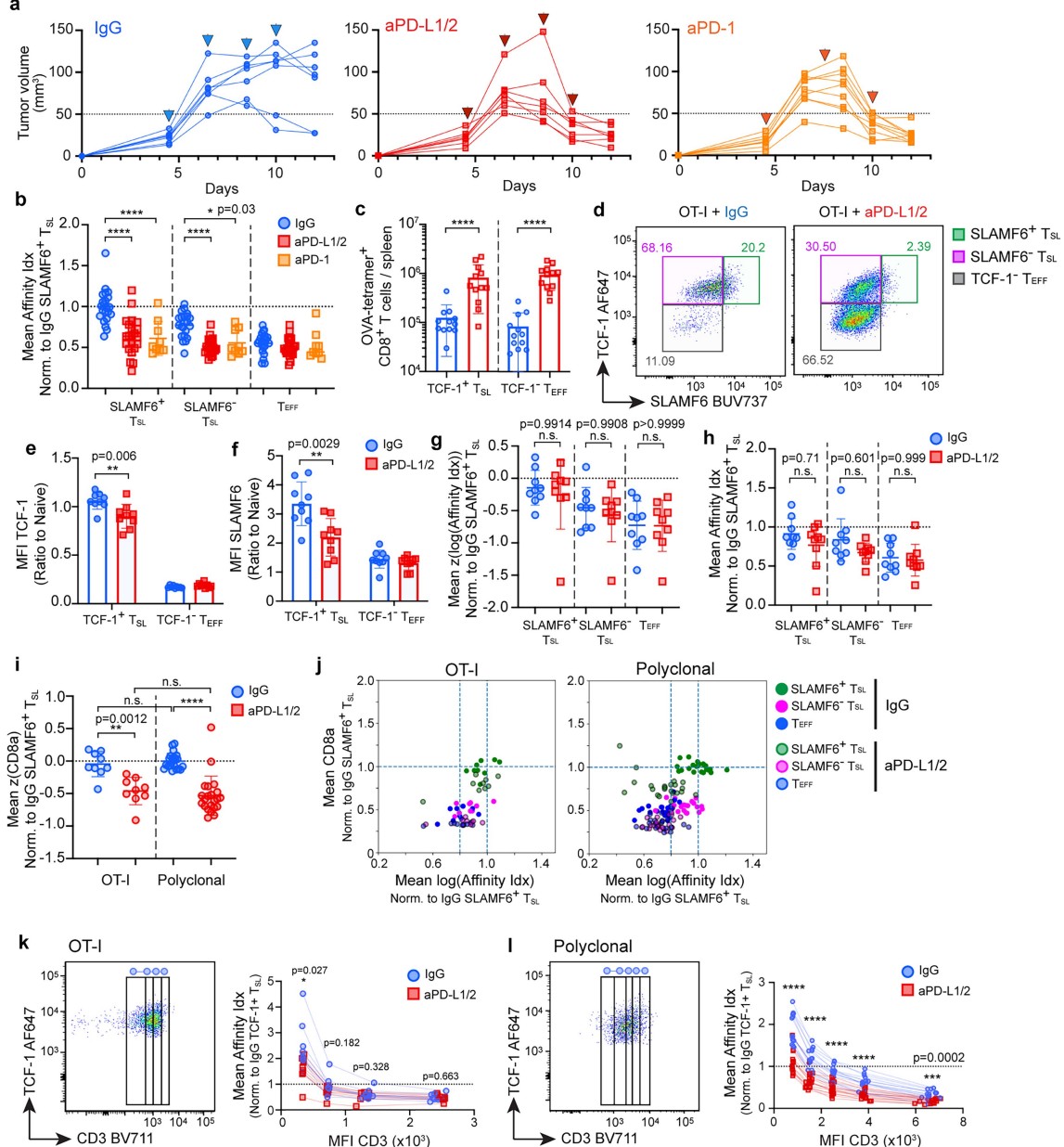

**Extended Data Fig. 7 | PD-1 checkpoint blockade reduces the frequency of high affinity CD8+ T$_{SL}$ in the tdLN. a**, Tumour growth curve in mice treated with control IgG, anti-PD-L1/PD-L2 or anti-PD-1. Arrows indicate checkpoint antibody injection. Data from 2 independent experiments (n = 7 for IgG and anti-PD-L1/PD-L2; 9 for anti-PD-1). **b**, Mean imputed affinity index of each subset of OVA-tetramer+ CD8+ T cells from Day 12 tdLN normalized to IgG-treated SLAMF6+ T$_{SL}$ from the same experimental cohort. Data pooled from 7 independent experiments for IgG and anti-PD-L1/2 treatment (n = 22 per group), 2 independent experiments for anti-PD-1 group (n = 9). **c**, Quantification of OVA-tetramer+ TCF-1+ T$_{SL}$ and TCF-1- T$_{EFF}$ recovered from Day 12 spleen. Data from 4 independent experiments (n = 12 per treatment group). **d-i**, Tumour-bearing mice adoptively transferred with 2×10³ congenic CD45.1+ naïve OT-I.Rag1⁻/⁻ cells 1 day prior to tumour induction and treated with checkpoint antibody as shown in Fig. 5a. **d**, Representative flow cytometry plots showing changes in TCF-1 and SLAMF6 expression of monoclonal OT-I cells from Day 12 tdLN. Gates denote subsets used in (**g, h**). **e-f**, Mean fluorescence intensity of TCF-1 (**e**) and SLAMF6 (**f**) of OT-I cells normalized to endogenous naïve (CD44- TCF-1+ PD-1-) CD8+ T cells. **g-h**, Standardized log transformed imputed TCR affinity index (**g**) and mean imputed affinity index (**h**) of each treatment group, normalized to IgG-treated SLAMF6+ T$_{SL}$ from the same experimental cohort. Each data point represents the mean subset value from each animal. **i**, Standardized CD8α expression of SLAMF6+ OT-I T$_{SL}$ compared with polyclonal SLAMF6+ T$_{SL}$ data from Fig. 5b–i, normalized to the mean CD8α expression of IgG-treated SLAMF6+ T$_{SL}$ from the same experimental cohort. Data in (**d-i**) pooled from 2 independent experiments (n = 9 per treatment group). **j**, Scatter plots showing the relationship between CD8a expression and log(Affinity index) of each subset from the monoclonal OT-I experiments (left panel, data from **d-i**) and polyclonal experiments (right panel, data from Fig. 5b–i). Control groups are shown in solid circles and aPD-L1/2 treated groups in shaded circles. Note the partially reduced affinity index in SLAMF6- T$_{SL}$ and TCF-1- T$_{EFF}$ from OT-I cells and the corresponding lower CD8a expression in those subsets. Checkpoint treatment (shaded circles) led to slight reduction in CD8a expression of SLAMF6+ T$_{SL}$ (green) but similar affinity indices in OT-I (left panel), while substantial drop in affinity indices was observed in the polyclonal OVA-tetramer+ cells (right panel). Data for (**j**) were pooled from 2 independent experiments (n = 9 per treatment group) for OT-I and 7 independent experiments (n = 22 per treatment group) for polyclonal dataset. **k-l**, TCR affinity indices gated based on TCR (CD3) expression for OT-I (**k**) and polyclonal (**l**) TCF-1+ T$_{SL}$ from Day 12 tdLN. Data for (**k, l**) were pooled from 2 independent experiments (n = 9 per treatment group) for OT-I and 5 independent experiments (n = 15 per treatment group) for polyclonal dataset. Error bars: means ± s.d. *p < 0.05, **p < 0.01, ***p < 0.001, ****p < 0.0001; one-way ANOVA with Tukey's multiple comparisons in (**b, i**), unpaired two-tailed *t*-test in all others.

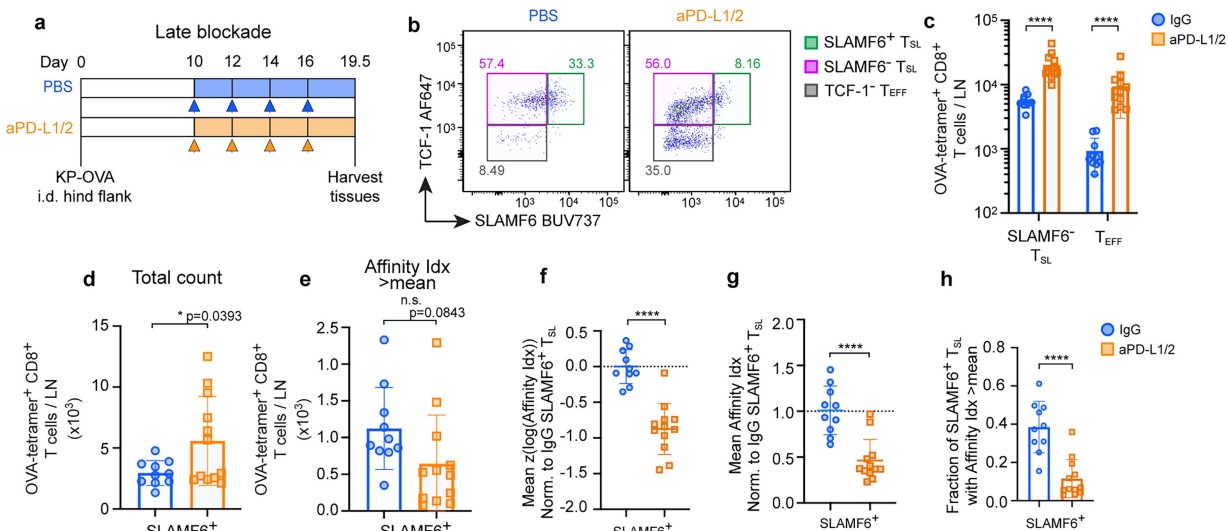

**Extended Data Fig. 8 | PD-1 checkpoint blockade at late time point similarly disrupts accumulation of high affinity SLAMF6+T$_{SL}$ in the tdLN. a**, Experimental scheme illustrating the checkpoint treatment strategy of late PD-1 ligand blockade initiated at Day 10. **b**, Representative flow cytometry plots of TCF-1 and SLAMF6 expression on OVA-tetramer+ CD8+ T cells in the tdLN at Day 19.5 and the gating strategy defining the three major T$_{SL}$ and T$_{EFF}$ subsets. The total numbers of each subset recovered from the tdLN are shown in (**c,d**). **e**, Total number of "high affinity" SLAMF6+ T$_{SL}$ (imputed affinity index >mean of control (PBS-treated) SLAMF6+ T$_{SL}$ from the same experimental cohort) in the tdLN. Standardized log transformed imputed TCR affinity index (**f**) and mean imputed affinity index (**g**) of each treatment group, normalized to PBS-treated SLAMF6+ T$_{SL}$ from the same experimental cohort. **h**, Fraction of "high affinity" SLAMF6+ T$_{SL}$ derived from the quantification of cell numbers in (**e**) divided by total SLAMF6+ T$_{SL}$ count (**d**). Data pooled from 3 independent experiments (n = 10 (PBS-treated) and n = 12 (aPD-L1/2-treated)). Error bars: means ± s.d. *p < 0.05, ****p < 0.0001; unpaired two-tailed $t$-test.

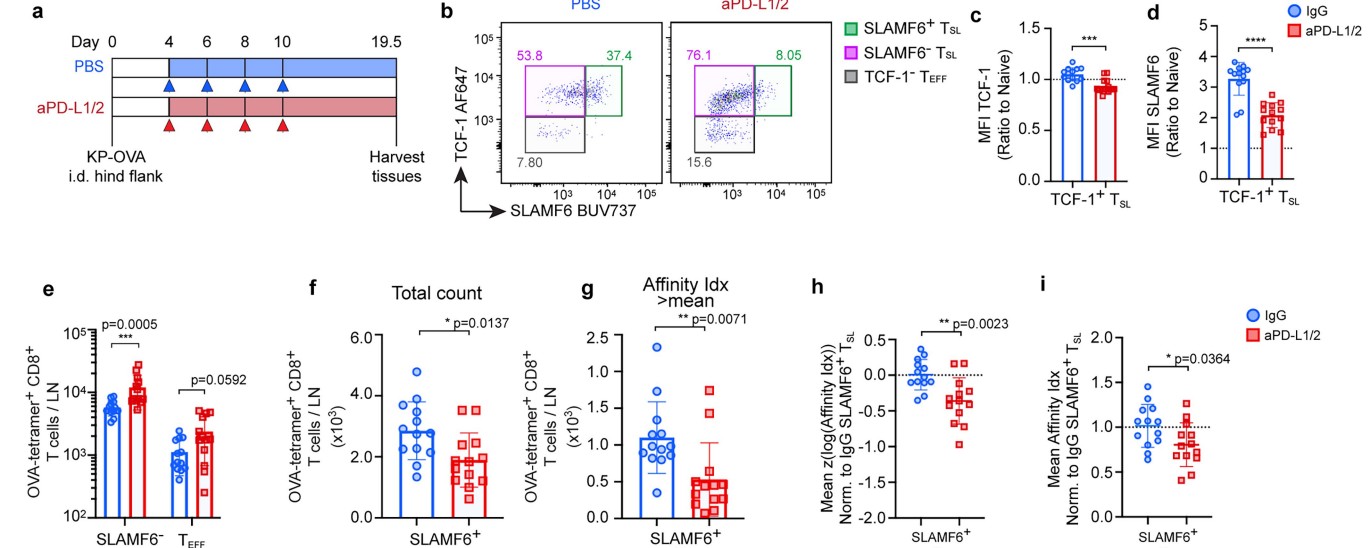

**Extended Data Fig. 9 | Persistence of high affinity SLAMF6+T$_{SL}$ loss after cessation of checkpoint blockade. a**, Experimental scheme illustrating the checkpoint treatment strategy allowing 9-10 days of recovery after treatment had stopped at Day 10. **b**, Representative flow cytometry plots of TCF-1 and SLAMF6 expression on OVA-tetramer+ CD8+ T cells in the tdLN at Day 19.5 and the gating strategy defining the three major T$_{SL}$ and T$_{EFF}$ subsets. Mean fluorescence intensity of TCF-1 (**c**) and SLAMF6 (**d**) of each subset of OVA-tetramer+ CD8+ T cells, normalized to naïve (CD44- TCF-1+ PD-1-) CD8+ T cells. The total numbers of each subset recovered from the tdLN are

shown in (**e, f**). **g**, Total number of "high affinity" SLAMF6+ T$_{SL}$ (imputed affinity index >mean of control (PBS-treated) SLAMF6+ T$_{SL}$ from the same experimental cohort) in the tdLN. Standardized log transformed imputed TCR affinity index (**h**) and mean imputed affinity index (**i**) of each treatment group, normalized to PBS-treated SLAMF6+ T$_{SL}$ from the same experimental cohort. Data in (**b-i**) are pooled from 4 independent experiments (n = 13 per treatment group). Error bars: means ± s.d. *p < 0.05, **p < 0.01, ***p < 0.001, ****p < 0.0001; unpaired two-tailed *t*-test.

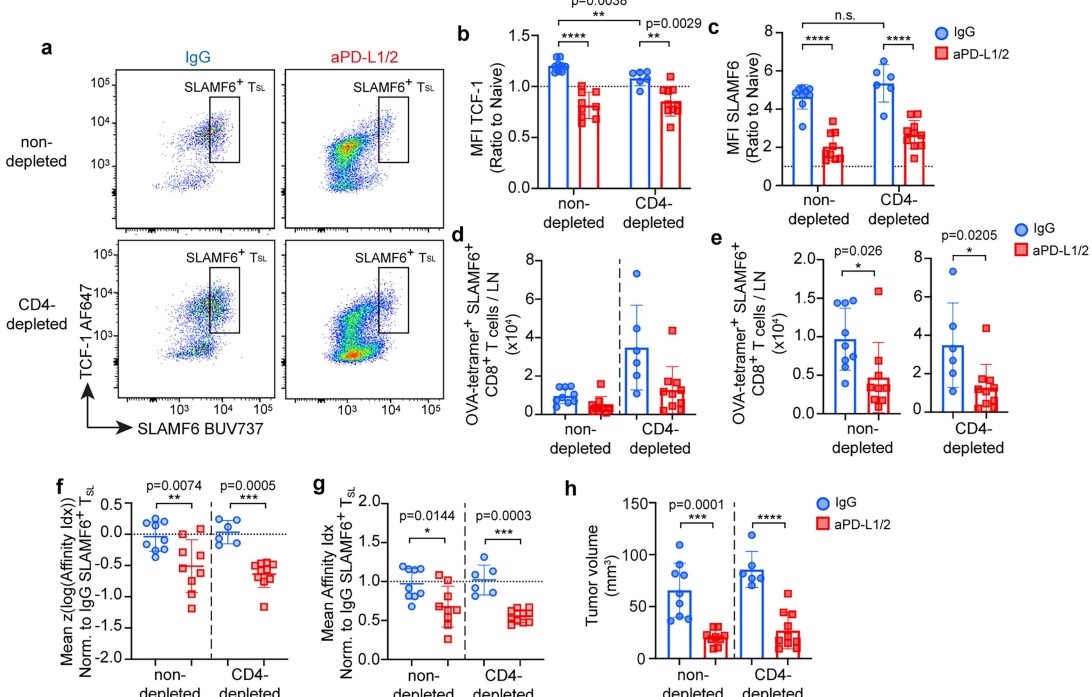

**Extended Data Fig. 10 | CD4+ T cells are not required to induce CD8+ phenotypic and clonal repertoire shift after checkpoint blockade.**
**a**, Representative flow cytometry plots of OVA-tetramer+ CD8+ T cells from Day 12 tdLN of mice pre-treated with or without anti-CD4 depleting antibodies, and with control IgG or anti-PD-L1/PD-L2 according to Fig. 5a experimental scheme. **b, c**, Mean fluorescence intensity of TCF-1 (**b**) and SLAMF6 (**c**) of OVA-tetramer+ TCF-1+ $T_{SL}$ normalized to naïve (CD44- TCF-1+ PD-1-) CD8+ T cells. **d, e**, Quantification of OVA-tetramer+ SLAMF6+ $T_{SL}$ from Day 12 tdLN (**d**) and re-scaled for each group (non-depleted and CD4-depleted) to show the relative extent of SLAMF6+ $T_{SL}$ reduction after checkpoint treatment (**e**).

**f, g**, Standardized log transformed imputed TCR affinity index (**f**) and mean imputed affinity index (**g**) of each treatment group, normalized to non-CD4-depleted, IgG-treated SLAMF6+ $T_{SL}$ from the same experimental cohort. Each data point represents the mean subset value from each animal. **h**, Tumour volume measured at the experimental end point at Day 12. Data from 3 independent experiments (n = 9 for non-depleted IgG and aPD-L1/2; n = 6 for CD4-depleted IgG; n = 10 for CD4-depleted aPD-L1/2). Error bars: means ± s.d. *p < 0.05, **p < 0.01, ***p < 0.001, ****p < 0.0001; unpaired two-tailed *t*-test in (**e**), one-way ANOVA with Tukey's multiple comparisons in all others.

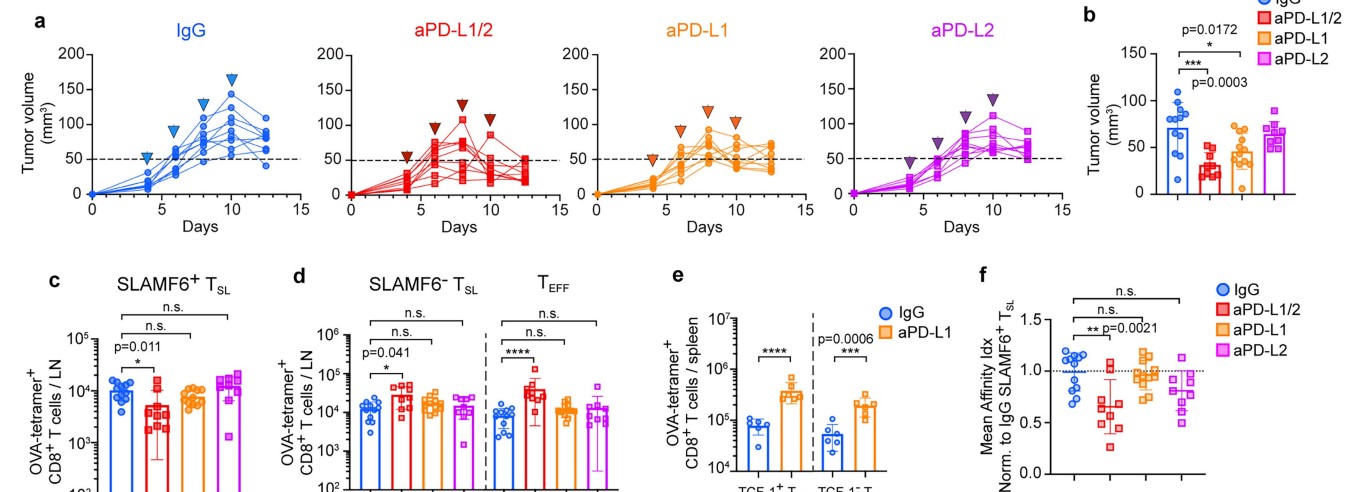

**Extended Data Fig. 11 | PD-L1 blockade alone does not result in dramatic loss of high affinity SLAMF6+T$_{SL}$. a,** Tumour growth curve in mice treated with control IgG, anti-PD-L1/PD-L2, anti-PD-L1 only or anti-PD-L2 only. Arrows indicate checkpoint antibody injection. Tumour volume measured at experimental end point at Day 12 for each group is shown in (**b**). **c, d,** Number of OVA-tetramer+ SLAMF6+ T$_{SL}$ (**d**), SLAMF6- T$_{SL}$ and TCF-1- T$_{EFF}$ (**e**) recovered from Day 12 tdLN. **e,** Number of OVA-tetramer+ TCF-1+ T$_{SL}$ and TCF-1- T$_{EFF}$ recovered from the spleen of mice treated with control IgG or anti-PD-L1 only at Day 12. **f,** Mean imputed affinity index of each treatment group, normalized

to IgG-treated SLAMF6+ T$_{SL}$ from the same experimental cohort. Each data point represents the mean subset value from each animal. Data shown in (**a**) are from 3 independent experiments (n = 9 per treatment group), (**b-d, f**) are pooled from 4 independent experiments for control IgG, anti-PD-L1 treatment groups (n = 12 per group), and 3 independent experiments for anti-PD-L1/PD-L2 and anti-PD-L2 groups (n = 9 per group). Data in (**e**) are from 2 independent experiments (n = 6 per group). Error bars: means ± s.d. *p < 0.05, **p < 0.01, ***p < 0.001, ****p < 0.0001; unpaired two-tailed *t*-test in (**e**), one-way ANOVA with Tukey's multiple comparisons in all others.

# Reporting Summary

## Statistics

For all statistical analyses, confirm that the following items are present in the figure legend, table legend, main text, or Methods section.

| n/a | Confirmed | |
|---|---|---|
| ☐ | ☒ | The exact sample size (*n*) for each experimental group/condition, given as a discrete number and unit of measurement |
| ☐ | ☒ | A statement on whether measurements were taken from distinct samples or whether the same sample was measured repeatedly |
| ☐ | ☒ | The statistical test(s) used AND whether they are one- or two-sided *Only common tests should be described solely by name; describe more complex techniques in the Methods section.* |
| ☐ | ☒ | A description of all covariates tested |
| ☐ | ☒ | A description of any assumptions or corrections, such as tests of normality and adjustment for multiple comparisons |
| ☐ | ☒ | A full description of the statistical parameters including central tendency (e.g. means) or other basic estimates (e.g. regression coefficient) AND variation (e.g. standard deviation) or associated estimates of uncertainty (e.g. confidence intervals) |
| ☐ | ☒ | For null hypothesis testing, the test statistic (e.g. *F*, *t*, *r*) with confidence intervals, effect sizes, degrees of freedom and *P* value noted *Give P values as exact values whenever suitable.* |
| ☒ | ☐ | For Bayesian analysis, information on the choice of priors and Markov chain Monte Carlo settings |
| ☒ | ☐ | For hierarchical and complex designs, identification of the appropriate level for tests and full reporting of outcomes |
| ☒ | ☐ | Estimates of effect sizes (e.g. Cohen's *d*, Pearson's *r*), indicating how they were calculated |

*Our web collection on statistics for biologists contains articles on many of the points above.*

## Software and code

Policy information about availability of computer code

| Data collection | Flow cytometry data were collected using Facs Diva v8.0.3 (BD Biosciences). Confocal microscopy data using Leica LAS X software (v3.5.7 for SP8, v4.6.0 and v4.8.0 for STELLARIS). |
|---|---|
| Data analysis | Imaging processing and analysis pipeline comprised the use of custom Python-based scripts (available on Github: https://github.com/jlhor/3d-imaging-pipeline), open source (Fiji ImageJ v1.54f) and commercial softwares (Imaris v10.0.0). Flow cytometry data were analyzed using FlowJo v10. Statistical tests were performed in Graphpad Prism v9. Linear modeling was performed using R v4.2.0. Graphs were plotted in Prism v9 and edited for appearance using Adobe Illustrator 28.5. Animations were generated using napari-animation package and edited with Davinci Resolve 18.6. |

The following Python packages were used for data visualization:
matplotlib 3.6.3
mpl_scatter_density 0.8
seaborn 0.13.2
shapely 2.0.6
scikit-image 0.21.0
scikit-learn 1.3.2
KDEpy 1.1.9
napari 0.4.18
napari-animation 0.0.7

Packages for cell segmentation:

```
stardist 0.9.1
cellpose 3.0.7
```

For manuscripts utilizing custom algorithms or software that are central to the research but not yet described in published literature, software must be made available to editors and reviewers. We strongly encourage code deposition in a community repository (e.g. GitHub). See the Nature Portfolio guidelines for submitting code & software for further information.

## Data

Policy information about availability of data

All manuscripts must include a data availability statement. This statement should provide the following information, where applicable:
- Accession codes, unique identifiers, or web links for publicly available datasets
- A description of any restrictions on data availability
- For clinical datasets or third party data, please ensure that the statement adheres to our policy

The representative imaging dataset shown in Fig. 1 can be accessed via the Zenodo repository (https://zenodo.org/records/15599322). Due to the large file sizes, other imaging dataset will be made available upon request from the corresponding authors. Source data are provided with this paper.

## Research involving human participants, their data, or biological material

Policy information about studies with human participants or human data. See also policy information about sex, gender (identity/presentation), and sexual orientation and race, ethnicity and racism.

| | |
|---|---|
| Reporting on sex and gender | N/A |
| Reporting on race, ethnicity, or other socially relevant groupings | N/A |
| Population characteristics | N/A |
| Recruitment | N/A |
| Ethics oversight | N/A |

Note that full information on the approval of the study protocol must also be provided in the manuscript.

# Field-specific reporting

Please select the one below that is the best fit for your research. If you are not sure, read the appropriate sections before making your selection.

☒ Life sciences        ☐ Behavioural & social sciences        ☐ Ecological, evolutionary & environmental sciences

For a reference copy of the document with all sections, see nature.com/documents/nr-reporting-summary-flat.pdf

# Life sciences study design

All studies must disclose on these points even when the disclosure is negative.

| | |
|---|---|
| Sample size | No statistical methods were used to determine sample size. The number of mice per group within each independent experiment was determined based on similar experiments in the field, and on the practicality of conducting the animal study taking into account the total number of test conditions involved. In imaging experiments, 2-3 mice per group/condition were used. In flow cytometry experiments, 3-6 mice per group/condition were used. The exact numbers and statistical tests detailed in the figure legends and methods. |
| Data exclusions | No data were excluded from analysis. |
| Replication | Replicate experiments were performed with the number of independent experiments (minimum of 2) as detailed in the figure legends and methods sections. |
| Randomization | Age-matched littermate mice were used to control for litter, cage, and age effects. Mice were randomly assigned for experimental groups (but no specific randomization protocol was used). |
| Blinding | Investigators were not blinded to group allocation during data collection and analysis, as most of the readouts are quantitative and not subjective. Tumor injection, treatment and measurement were performed by one person and not blinded. |

# Reporting for specific materials, systems and methods

We require information from authors about some types of materials, experimental systems and methods used in many studies. Here, indicate whether each material, system or method listed is relevant to your study. If you are not sure if a list item applies to your research, read the appropriate section before selecting a response.

## Materials & experimental systems

| n/a | Involved in the study |
|---|---|
| ☐ | ☒ Antibodies |
| ☐ | ☒ Eukaryotic cell lines |
| ☒ | ☐ Palaeontology and archaeology |
| ☐ | ☒ Animals and other organisms |
| ☒ | ☐ Clinical data |
| ☒ | ☐ Dual use research of concern |
| ☒ | ☐ Plants |

## Methods

| n/a | Involved in the study |
|---|---|
| ☒ | ☐ ChIP-seq |
| ☐ | ☒ Flow cytometry |
| ☒ | ☐ MRI-based neuroimaging |

# Antibodies

| | |
|---|---|
| Antibodies used | All antibodies used including their conjugated fluorophores, clones, vendors, catalog numbers and dilutions used are listed in Supplementary Table 1. |
| Validation | All antibodies are validated per the data sheets available on the manufacturers' websites. For imaging experiments, validation was performed by assessing the immunostaining and their expected localization on the cells (membrane, cytoplasmic, nuclear) and the cell types as identified by other markers, on published immunohistochemistry data, and whenever possible, compared their expression level with the data obtained using flow cytometry (also presented in the extended data figures).<br><br>The following antibodies (clone, target species, application) were used and validation was performed as stated:<br><br>Rat monoclonal anti-PD-1 AF647 (RMP1-30, mouse, imaging), validation data from manufacturer's website: https://www.biolegend.com/en-us/products/alexa-fluor-647-anti-mouse-cd279-pd-1-antibody-12480<br><br>Rat monoclonal anti-PD-1 purified (RMP1-30, mouse, imaging), validation data from manufacturer's website: https://www.biolegend.com/en-us/products/purified-anti-mouse-cd279-pd-1-antibody-455<br><br>Rat monoclonal anti-CD45.1 AF594 (A20, mouse, imaging), validation data from manufacturer's website: https://www.biolegend.com/en-us/products/alexa-fluor-594-anti-mouse-cd45-1-antibody-13424<br><br>Rat monoclonal anti-CD45.2 AF594 (104, mouse, imaging), validation data from manufacturer's website: https://www.biolegend.com/en-us/products/alexa-fluor-594-anti-mouse-cd45-2-antibody-13446<br><br>Rat monoclonal anti-CD8b AF488 (YTS156.7.7, mouse, imaging), validation data from manufacturer's website: https://www.biolegend.com/en-us/products/alexa-fluor-488-anti-mouse-cd8b-ly-3-antibody-17365<br><br>Mouse monoclonal anti-Ly-108 AF647 (13G3, mouse, imaging), validation data from manufacturer's website: https://www.bdbiosciences.com/en-us/products/reagents/flow-cytometry-reagents/research-reagents/single-color-antibodies-ruo/alexa-fluor-647-mouse-anti-mouse-ly-108.561547<br><br>Mouse monoclonal anti-Ly-108 RY586 (13G3, mouse, imaging), validation data from manufacturer's website: https://www.bdbiosciences.com/en-us/products/reagents/flow-cytometry-reagents/research-reagents/single-color-antibodies-ruo/ry586-mouse-anti-mouse-ly-108.753676<br><br>Rat monoclonal anti-Ki-67 eFluor450 (SolA15, mouse, imaging), validation data from manufacturer's website: https://www.thermofisher.com/antibody/product/Ki-67-Antibody-clone-SolA15-Monoclonal/48-5698-82<br><br>Rabbit monoclonal anti-TCF-1 AF555 (C63D9, mouse, imaging), validation data from manufacturer's website: https://www.cellsignal.com/products/antibody-conjugates/tcf1-tcf7-c63d9-rabbit-mab-alexa-fluor-555-conjugate/17404<br><br>Rabbit monoclonal anti-TCF-1 AF594 (C63D9, mouse, imaging), validation data from manufacturer's website: https://www.cellsignal.com/products/antibody-conjugates/tcf1-tcf7-c63d9-rabbit-mab-alexa-fluor-594-conjugate/35972<br><br>Rabbit monoclonal anti-BATF AF647 (D7C5, mouse, imaging), validation data from manufacturer's website: https://www.cellsignal.com/products/antibody-conjugates/batf-d7c5-rabbit-mab-alexa-fluor-647-conjugate/47914<br><br>Rabbit monoclonal anti-BATF AF555 (D7C5, mouse, imaging), validation data from manufacturer's website: https://www.cellsignal.com/products/primary-antibodies/batf-d7c5-rabbit-mab/8638. Additional validation by comparing with AF647-conjugated anti-BATF antibody.<br><br>Rabbit monoclonal anti-NFAT1 AF647 (D43B1, mouse, imaging), validation data from manufacturer's website: https://www.cellsignal.com/products/antibody-conjugates/nfat1-d43b1-xp-rabbit-mab-alexa-fluor-647-conjugate/14201<br><br>Rabbit monoclonal anti-cleaved caspase-3 (Asp175) AF555 (D3E9, mouse, imaging), validation data from manufacturer's website: https://www.cellsignal.com/products/antibody-conjugates/cleaved-caspase-3-asp175-d3e9-rabbit-mab-alexa-fluor-555-conjugate/9604 |

Goat polyclonal anti-PD-L1 (polyclonal, mouse, imaging), validation data from manufacturer's website: https://www.rndsystems.com/products/mouse-pd-l1-b7-h1-antibody_af1019

Rabbit polyclonal anti-XCR1 (polyclonal, mouse, imaging), validation data from manufacturer's website: https://www.lsbio.com/antibodies/xcr1-antibody-aa244-322-wb-western-ls-c763561/789657. Additional validation performed on lymph node tissues from XCR1-venus reporter mice (heterozygous XCR1-venus knock-in for presence of XCR1 protein, and homozygous knock-in for absence of XCR1).

Rat monoclonal anti-CD3 BV711 (17A2, mouse, flow cytometry), validation data from manufacturer's website: https://www.biolegend.com/en-us/products/brilliant-violet-711-anti-mouse-cd3-antibody-10022

Armenian Hamster monoclonal anti-CD27 BV650 (LG.3A10, mouse, flow cytometry), validation data from manufacturer's website: https://www.biolegend.com/en-us/products/brilliant-violet-650-anti-mouse-rat-human-cd27-antibody-14413

Rat monoclonal anti-PD-1 BV785 (29F.1A12, mouse, flow cytometry), validation data from manufacturer's website: https://www.biolegend.com/en-us/products/brilliant-violet-785-anti-mouse-cd279-pd-1-antibody-9874

Rat monoclonal anti-CD200 PE-Cy7 (OX-90, mouse, flow cytometry), validation data from manufacturer's website: https://www.biolegend.com/en-us/products/pecyanine7-anti-mouse-cd200-ox2-antibody-19211

Rat monoclonal anti-CD44 AF700 (IM7, mouse, flow cytometry), validation data from manufacturer's website: https://www.biolegend.com/en-us/products/alexa-fluor-700-anti-mouse-human-cd44-antibody-3406

Rat monoclonal anti-PD-L1 BV421 (10F.9G2, mouse, flow cytometry), validation data from manufacturer's website: https://www.biolegend.com/en-us/products/brilliant-violet-421-anti-mouse-cd274-b7-h1-pd-l1-antibody-7250

Rat monoclonal anti-PD-L2 PE-Dazzle 594 (TY25, mouse, flow cytometry), validation data from manufacturer's website: https://www.biolegend.com/en-us/products/pedazzle594-anti-mouse-cd273-antibody-15630

Rat monoclonal anti-Vα2 PE-Cy7 (B20.1, mouse, flow cytometry), validation data from manufacturer's website: https://www.biolegend.com/en-us/products/pe-cyanine7-anti-mouse-tcr-va2-antibody-15019

Rat monoclonal anti-CD45.1 AF488 (A20, mouse, flow cytometry), validation data from manufacturer's website: https://www.biolegend.com/en-us/products/alexa-fluor-488-anti-mouse-cd45-1-antibody-3103

Rat monoclonal anti-CD45.1 AF700 (A20, mouse, flow cytometry), validation data from manufacturer's website: https://www.biolegend.com/en-us/products/alexa-fluor-700-anti-mouse-cd45-1-antibody-3392

Rat monoclonal anti-CD45.2 AF488 (104, mouse, flow cytometry), validation data from manufacturer's website: https://www.biolegend.com/en-us/products/alexa-fluor-488-anti-mouse-cd45-2-antibody-3106

Rat monoclonal anti-CD45.2 AF647 (104, mouse, flow cytometry), validation data from manufacturer's website: https://www.biolegend.com/en-us/products/alexa-fluor-647-anti-mouse-cd45-2-antibody-3107

Mouse monoclonal anti-Bcl-2 AF488 (BCL/10C4, mouse, flow cytometry), validation data from manufacturer's website: https://www.biolegend.com/en-us/products/alexa-fluor-488-anti-bcl-2-antibody-6346

Mouse monoclonal anti-T-bet PE-Cy7 (4B10, mouse, flow cytometry), validation data from manufacturer's website: https://www.biolegend.com/en-us/products/pe-cyanine7-anti-t-bet-antibody-8328

Rat monoclonal anti-MHC-II (I-A/I-E) AF700 (M5/114.15.2, mouse, flow cytometry), validation data from manufacturer's website: https://www.biolegend.com/en-us/products/alexa-fluor-700-anti-mouse-i-a-i-e-antibody-3413

Armenian Hamster monoclonal anti-CD11c PE-Cy7 (N418, mouse, flow cytometry), validation data from manufacturer's website: https://www.biolegend.com/en-us/products/pe-cyanine7-anti-mouse-cd11c-antibody-3086

Rat monoclonal anti-SIRPα AF488 (P84, mouse, flow cytometry), validation data from manufacturer's website: https://www.biolegend.com/en-us/products/alexa-fluor-488-anti-mouse-cd172a-sirpalpha-antibody-14089

Mouse monoclonal anti-XCR1 PE (ZET, mouse, flow cytometry), validation data from manufacturer's website: https://www.biolegend.com/en-us/products/pe-anti-mouse-rat-xcr1-antibody-10217

Mouse monoclonal anti-XCR1 AF647 (ZET, mouse, flow cytometry), validation data from manufacturer's website: https://www.biolegend.com/en-us/products/alexa-fluor-647-anti-mouse-rat-xcr1-antibody-10402

Armenian Hamster monoclonal anti-CD80 PerCP-Cy5.5 (16-10A1, mouse, flow cytometry), validation data from manufacturer's website: https://www.biolegend.com/en-us/products/percp-cyanine5-5-anti-mouse-cd80-antibody-4275

Mouse monoclonal anti-CX3CR1 BV421 (SA011F11, mouse, flow cytometry), validation data from manufacturer's website: https://www.biolegend.com/en-us/products/brilliant-violet-421-anti-mouse-cx3cr1-antibody-11852

Mouse monoclonal anti-CX3CR1 BV605 (SA011F11, mouse, flow cytometry), validation data from manufacturer's website: https://www.biolegend.com/en-us/products/brilliant-violet-605-anti-mouse-cx3cr1-antibody-12110

Rat monoclonal anti-CD39 PE/Dazzle™ 594 (Duha59, mouse, flow cytometry), validation data from manufacturer's website: https://

www.biolegend.com/en-us/products/pe-dazzle-594-anti-mouse-cd39-antibody-16385

Mouse monoclonal anti-H-2Kb bound to SIINFEKL APC (25-D1.16, mouse, flow cytometry), validation data from manufacturer's website: https://www.biolegend.com/en-us/products/apc-anti-mouse-h-2kb-bound-to-siinfekl-antibody-7882

Rat monoclonal anti-CD8α BUV395 (53-6.7, mouse, flow cytometry), validation data from manufacturer's website: https://www.bdbiosciences.com/en-us/products/reagents/flow-cytometry-reagents/research-reagents/single-color-antibodies-ruo/buv395-rat-anti-mouse-cd8a.563786

Mouse monoclonal anti-Ly-108 BUV737 (13G3, mouse, flow cytometry), validation data from manufacturer's website: https://www.bdbiosciences.com/en-us/products/reagents/flow-cytometry-reagents/research-reagents/single-color-antibodies-ruo/buv737-mouse-anti-mouse-ly-108.741893

Rat monoclonal anti-CD24 BUV737 (M1/69, mouse, flow cytometry), validation data from manufacturer's website: https://www.bdbiosciences.com/en-us/products/reagents/flow-cytometry-reagents/research-reagents/single-color-antibodies-ruo/buv737-rat-anti-mouse-cd24.612832

Rat monoclonal anti-CD4 BV786 (RM4-4, mouse, flow cytometry), validation data from manufacturer's website: https://www.bdbiosciences.com/en-us/products/reagents/flow-cytometry-reagents/research-reagents/single-color-antibodies-ruo/bv786-rat-anti-mouse-cd4.740844

Rat monoclonal anti-Ki-67 AF488 (SolA15, mouse, flow cytometry), validation data from manufacturer's website: https://www.thermofisher.com/antibody/product/Ki-67-Antibody-clone-SolA15-Monoclonal/53-5698-82

Rabbit monoclonal anti-TCF-1 AF647 (C63D9, mouse, flow cytometry), validation data from manufacturer's website: https://www.cellsignal.com/products/antibody-conjugates/tcf1-tcf7-c63d9-rabbit-mab-alexa-fluor-647-conjugate/6709

Rabbit monoclonal anti-BATF AF647 (D7C5, mouse, flow cytometry), validation data from manufacturer's website: https://www.cellsignal.com/products/antibody-conjugates/batf-d7c5-rabbit-mab-alexa-fluor-647-conjugate/47914

Rabbit monoclonal anti-BATF AF555 (D7C5, mouse, flow cytometry), validation data from manufacturer's website: https://www.cellsignal.com/products/primary-antibodies/batf-d7c5-rabbit-mab/8638. Additional validation by comparing with AF647-conjugated anti-BATF antibody.

Rabbit monoclonal anti-cleaved caspase-3 (Asp175) AF488 (D3E9, mouse, flow cytometry), validation data from manufacturer's website: https://www.cellsignal.com/products/antibody-conjugates/cleaved-caspase-3-asp175-d3e9-rabbit-mab-alexa-fluor-488-conjugate/9603

Rabbit monoclonal anti-Ubiquityl-Histone H2A (Lys119) AF488 (D27C4, mouse, flow cytometry), validation data from manufacturer's website: https://www.cellsignal.com/products/antibody-conjugates/ubiquityl-histone-h2a-lys119-d27c4-xp-rabbit-mab-alexa-fluor-488-conjugate/26498. Antibody chosen based on Kang et al., 2024: 10.1126/science.adl4492.

# Eukaryotic cell lines

Policy information about cell lines and Sex and Gender in Research

| | |
|---|---|
| Cell line source(s) | KP1233 (from T. Jacks lab) and MC38 (from M. Meier-Schellersheim lab) were mouse derived tumor cell lines. 293T adherent (Takara #632180, gift from T. Jacks lab) cells were used for lentiviral production. |
| Authentication | No authentication was performed beyond that provided by the laboratories who gifted the cell lines, or by the vendors. |
| Mycoplasma contamination | 293T lines were tested negative for mycoplasma. Tumor cell lines were not tested for mycoplasma. |
| Commonly misidentified lines (See ICLAC register) | No commonly misidentified cell lines were used. |

# Animals and other research organisms

Policy information about studies involving animals; ARRIVE guidelines recommended for reporting animal research, and Sex and Gender in Research

| | |
|---|---|
| Laboratory animals | Species, strains and age information of mice used in this study are detailed in the methods section. Mice within individual experiments were age and sex matched.<br><br>All mice were housed in 12-hour light-dark cycle with ad libitum access to food and water.<br>CD45.2 (C57BL/6J) and B6.GFP (C57BL/6-Tg(UBC-GFP)30Scha/J) mice were purchased from the Jackson Laboratory (strain numbers 000664 and 004354 respectively); CD45.1 (B6.SJL Ptprca), OT-I.CD45.1 (B6(Ly5.1)-[Tg]TCR OT-I-[KO]RAG1), OT-I.CD45.2 (C57BL/6NAi-[Tg]TCR OT-1-[KO]RAG1) were obtained from the NIAID-Taconic exchange program (strain numbers 8478, 300 and 175, respectively). XCR1-DTR (B6.Cg-Xcr1tm2(HBEGF/Venus)Ksho) and XCR1-venus (B6.Cg-Xcr1tm1Ksho) transgenic mice15 were kind gifts from Tsuneyasu Kaisho (RIKEN-Yokohama and Osaka University). OT-I.GFP mice were cross-bred from OT-I.CD45.2 and B6.GFP and maintained as a homozygous strain in our laboratory. The majority of mice employed were female and aged 6-16 weeks at the beginning of experiments, with a small number of experiments performed in male mice. All mice were bred and maintained under |

specific pathogen-free conditions at an American Association for the Accreditation of Laboratory Animal Care (AAALAC)-accredited animal facility within NIAID and were used under study protocol LISB-4E approved by NIAID Animal Care and Use Committee (NIH).

| | |
|---|---|
| Wild animals | No wild animals were used in this study. |
| Reporting on sex | The majority of mice used for tumor studies were female mice, with a small number of experiments performed using male mice. No significant difference was observed between sex. |
| Field-collected samples | No samples collected from the filed were used in this study. |
| Ethics oversight | All protocols were approved by NIAID Animal Care and Use Committee (NIH) at an American Association for the Accreditation of Laboratory Animal Care (AAALAC)-accredited animal facility within NIAID . |

Note that full information on the approval of the study protocol must also be provided in the manuscript.

# Plants

| | |
|---|---|
| Seed stocks | *Report on the source of all seed stocks or other plant material used. If applicable, state the seed stock centre and catalogue number. If plant specimens were collected from the field, describe the collection location, date and sampling procedures.* |
| Novel plant genotypes | *Describe the methods by which all novel plant genotypes were produced. This includes those generated by transgenic approaches, gene editing, chemical/radiation-based mutagenesis and hybridization. For transgenic lines, describe the transformation method, the number of independent lines analyzed and the generation upon which experiments were performed. For gene-edited lines, describe the editor used, the endogenous sequence targeted for editing, the targeting guide RNA sequence (if applicable) and how the editor was applied.* |
| Authentication | *Describe any authentication procedures for each seed stock used or novel genotype generated. Describe any experiments used to assess the effect of a mutation and, where applicable, how potential secondary effects (e.g. second site T-DNA insertions, mosiacism, off-target gene editing) were examined.* |

# Flow Cytometry

## Plots

Confirm that:

☒ The axis labels state the marker and fluorochrome used (e.g. CD4-FITC).

☒ The axis scales are clearly visible. Include numbers along axes only for bottom left plot of group (a 'group' is an analysis of identical markers).

☒ All plots are contour plots with outliers or pseudocolor plots.

☒ A numerical value for number of cells or percentage (with statistics) is provided.

## Methodology

| | |
|---|---|
| Sample preparation | Lymphocytes were isolated from lymph nodes and spleens and made into single-cell suspensions using a syringe plunger and 100μm or 70μm cell strainers (MACS SmartStrainer, Miltenyi Biotec). ~3x106 cells were used in subsequent staining steps for flow cytometry analysis. Dendritic cell isolation was performed as described previously. Briefly, lymphoid tissues were sliced into small fragments using a scalpel blade and incubated in a digestion mix of collagenase type III (Worthington, 1mg/ml) and DNase I (20ug/ml) and vigorously mixed for 25 min at room temperature, followed by addition of 0.1M EDTA solution at 1/10 digestion volume for 5 min to dissociate lymphocytes from dendritic cells. Tissue debris were filtered out by passing the cell suspension through a 70μm nylon mesh.<br><br>T cells from skin tumor samples were isolated as described previously. Briefly, a 1cm2x1cm2 tumor-containing skin patch was harvested into collagenase III (Worthington, 3mg/ml), finely chopped with scissors and incubated at 37°C for 90 min before pressing through 70μm cell strainers. For spleen and tumor samples, cells were also treated with red blood cell lysis buffer prior to staining. Cell counts in LN and spleen were determined using an automated Cellometer T4 cell counter (Nexcelom Bioscience).<br><br>For detection of polyclonal OVA-specific CD8+ T cells, cells were first incubated with PE-conjugated H-2Kb-SIINFEKL tetramer (1:100, NIH Tetramer Core) for 20 min at 37°C, washed, followed by cell surface marker staining for 25 min at 4°C. Mouse BD Fc Block (1:200; BD Biosciences) was also included during cell surface marker staining step. A fixable LIVE/DEAD near-infrared staining dye was used for determining cell viability. For detection of intracellular proteins, stained cells were further treated with fixative and stained for antibodies against intracellular proteins using Foxp3/transcription factor staining buffer kit per the manufacturer's protocol (eBioscience). For tumor samples, CountBright Plus Absolute beads (Thermofisher) were added prior to sample acquisition. Samples were acquired using a BD Fortessa (BD Biosciences) and analyzed using Flowjo 10 (Treestar). |
| Instrument | BD Fortessa was used for flow cytometry data acquisition. |
| Software | FlowJo v10 |

| Cell population abundance | Frequencies of cell populations are indicated on flow cytometry plots and their absolute count quantified. |
| --- | --- |

| Gating strategy | Antigen-specific lymphocytes were gated based on LIVE/DEAD stain, CD3 and CD8a, and then gated using the appropriate tetramers. PD-1+ gate was used to gate on activated lymphocytes and filter out rare, contaminating population. Gating strategy for further subsetting are shown in Figures and Extended Data Figures. |
| --- | --- |

☒ Tick this box to confirm that a figure exemplifying the gating strategy is provided in the Supplementary Information.

