## [Peer Review file · Nature]

Inhibitory PD-1 axis maintains high avidity stem-like CD8+ T cells

Corresponding Author: Professor Ronald Germain

Version 0:

Reviewer comments:

Referee #1

(Remarks to the Author)

Hor et al. conducted an in-depth investigation into the mechanisms and dynamics of CD8+ T cell priming to tumor antigens within the tumor-draining lymph node (tdLN) during the early stages of tumor implantation. By combining advanced imaging techniques with flow cytometry, the authors provide key insights into T cell-dendritic cell (DC) dynamics and the origins of an anti-tumor CD8+ T cell response. Utilizing the KP-Ova (NSCLC) model implanted subcutaneously, they monitored OVA-specific OT-1 T cell responses and made several significant observations:

1. T Cell Interactions: The authors convincingly show that the TSL (PD-1^{int} SLAMF6⁺ TCF1⁺) cells engage in tight interactions shortly after tumor implantation. These cells are largely Ki67⁺, indicating active proliferation. They further differentiate these cells from OT-1 cells that have downregulated TCF1, which they refer to as Teff cells. Interestingly, the authors find that the effector-like cells in the tumor express the highest levels of PD-1, a pattern not observed in tissues less likely to contain tumor antigens. This cross-tissue comparison adds depth to the study.
2. Role of cDC1s in TSL Maintenance: Their data suggest that cDC1s play a crucial role in sustaining TSL cells in the tdLN. When cDC1s are deleted, TSL cell numbers drop by approximately fourfold, while Teff cell numbers remain unaffected.
3. TCR Affinity of TSL and Teff Cells: Using a tetramer-binding assay, they propose that TSL cells have higher-affinity TCRs than Teff cells, indicating that the TSL pool maintains the highest-affinity tumor-reactive T cells.
4. Effect of PD-1 Blockade: Blocking PD-1 disrupts cDC-TSL clusters and promotes Teff differentiation, but also appears to reduce TCR affinity.
5. Differential Effects of PD-L1/PD-L2 Blockade: Blocking PD-L1 or PD-L2 has a more modest effect on tumor growth and does not result in a loss of high-affinity TSL cells. This distinction between PD-1 and PD-L1/PD-L2 blockade may have important clinical implications, particularly since more than half of patients who initially respond to ICB later develop acquired resistance within 1-3 years.

Overall, this study represents a high-quality, rigorous contribution to the field. Given the clinical implications, comparative analyses like this are vital and will undoubtedly be of broad interest. However, the findings raise several important questions and suggestions for further clarification and exploration:

Major Comments:

1. Intratumoral TSL: The authors identify TSL cells in the tdLN in Figure 2d but do not provide adequate identification of these cells within the tumor. While the intratumoral TSL cells seem to express lower Slamf6 levels than in the tdLN, the small subset of TCF1^{hi} SLAMF6^{int} cells in the tumor represents the intratumoral TSL population that appears to have been overlooked by the authors. It would be important to incorporate a deeper analysis of this subset within the tumors, comparing its location relative to cDCs and assessing its TCR affinity. Additionally, the lack of analysis of this population in the context of PD-1 or PD-L1/L2 blockade is a notable omission. Including this population in the primary experiments would help to elucidate its potential as a local source of Teff and terminally exhausted T cells within the tumor.

The authors should also acknowledge terminally exhausted T cells, as this would conceptually enhance the study. Incorporating markers like CD101, CXCR6, and CX3CR1 into flow cytometric analyses would help to distinguish effector T cells from more terminally exhausted subsets, as intratumoral PD-1^{hi}/TCF1^{lo} cells are likely heterogeneous. Although the

authors may argue that most time points are early after tumor implantation, and terminally exhausted cells are likely scarce, better delineation of intratumoral CD8 T cells is essential.

Furthermore, it would be beneficial to confirm that Teff cells are indeed effector cells. Could the authors assess functional markers such as IFN γ , TNF α , GzmB, and transcription factors like T-bet and Eomes in these populations to ensure their definitions are correct, particularly as they are PD-1 $^{\text{lo}}$ in the tdLN and other tissues?

2. TCR Affinity and Repertoire Measurements: The authors interpret their tetramer-binding data to suggest that TSL cells have higher-affinity TCRs than Teff cells. However, this assay actually measures TCR avidity, which incorporates both TCR affinity and the higher-order interactions between TCR complexes. This interpretation becomes more complex in the context of ICB, where a reduction in "affinity" is observed in expanding Teff populations. If TSL cells are clonal precursors to the newly generated Teff cells, shouldn't the TCR affinity be similar? The observed differences between TSL and Teff tetramer binding could indicate a drop in TCR avidity in Teff cells or the emergence of low-affinity clones that infiltrate the tumor, or both. The authors need to examine the TCR repertoire more closely. Single-cell TCR sequencing of tdLN and intratumoral T cells could provide crucial distinctions between these models. Especially since previous studies have shown that anti-PD-1 therapy leads to an influx of new TCR clones into tumors, it would be critical for the authors to integrate this knowledge into their analysis.

It would also be valuable for the authors to incorporate findings from Dr. Andrea Schietinger's work (<https://doi.org/10.1084/jem.20201966>). The Schietinger lab found that the higher affinity antigens induced more T cell exhaustion, but the lower affinity maintained more effector-like cells that expressed higher levels of Tcf7 and other genes in TSL cells. This is somewhat at odds with the present manuscript's interpretation albeit this manuscript focuses on the tdLN whereas the Schietinger study focused on intratumoral cells. Hence, the need to compare the cells within the two sites as described in #1 above. Use of such APL models could also really help to strengthen the present manuscript's claims, and doing so would be highly encouraged to increase the relevance of the conclusions, but this Reviewer understands that this would be a large undertaking and may not be feasible.

3. Clinical Relevance: The finding that anti-PD-L1 or PD-L2 monotherapy may better preserve TSL cells compared to anti-PD-1 monotherapy is intriguing. Do clinical data support this distinction in terms of T cell responses or durability of anti-tumor immunity? While anti-PD-1 and anti-PD-L1 therapies yield comparable responses as monotherapies, is there any longer-term data on progression-free survival (PFS) or overall survival suggesting that anti-PD-L1 might offer more durable benefits? Can the authors find any evidence of this from clinical studies?

Minor Points:

1. Tumor Models: While the authors used a subcutaneous model of lung cancer, it is surprising they did not examine more relevant models where tumors originate in the lung. The lung-specific LV-OVA model developed by Dr. Tyler Jacks would be particularly well suited for these studies.

2. Ki67+ TSL Cells: The data in Figure 1 show that most TSL cells are Ki67+, yet they are not all rapidly dividing because their cell cycle is held in check by PD-1. Could the authors explore further which phase of the cell cycle these cells are in? Do the authors have any idea what allows for the spontaneous "release" of the TSL from PD-1-repression to allow for the differentiation of TSL into Teff and terminal Tex cells?

Referee #2

(Remarks to the Author)

The authors investigate the differentiation of stem-like CD8+ T cells in tumor-draining lymph nodes (tdLN), and the effect of PD-1 checkpoint blockade on the fate of these cells. Using 3D imaging techniques, the authors identify a population of stem-like CD8+ T cells (Tsl) (expressing TCF-1, PD-1 and SLAMF6) in the tdLN, and find that PD-1 expression by these cells correlates with high "TCR affinity index" for the model tumor antigen, consistent with the premise that cells with stronger reactivity to tumor antigens will have higher PD-1 expression. The authors explore the hypothesis that PD-1 expression protects these stem-like cells from premature differentiation into terminally differentiated effector cells (Teff), finding supportive evidence for this by observing a loss in frequency and reduced TCR affinity index in SLAMF6+ Tsl, following anti-PD-1 or PD-L1/2 blockade.

These studies are well conducted and certainly help define the stem-like population(s) that develop in tumor-draining lymph nodes, and the authors raise the interesting and important hypothesis that checkpoint blockade may essentially "burn-out" the most avid stem-like CD8+ T cells, leading to trade off enhanced short-term efficacy of tumor control with loss of the high affinity stem-like pool. Although the approaches are elegant and, for the most part informative, there are numerous issues that dampen enthusiasm.

1) A general concern is that the authors examine quite early time points but interpret the data as analysis of "late" or "prolonged" antigen presentation. As the authors show, the same subsets - in similar proportions - are seen in the context of protein/adjuvant immunization (Ext. Fig. 2), which would presumably be a model of acute priming, against a non-replicating

source of antigen – hence the argument that these cells are characteristic of “prolonged” antigen presentation is questionable. Furthermore, the authors premise leads them to interfere with the process during early stages of immune priming – day 4 (Figs. 4,5) and day 5 (Fig. 3): these studies may have little to do with the stated intent of this study which was to address the maintenance and role of the “reservoir” of stem-like antigen-specific CD8+ T cells in tdLNs (e.g., lines 23-27 and 67-72). To address that important issue, it would be more meaningful to study later time points, when the initial response is complete, and the maintenance of stem-like cells is relevant for sustaining the anti-tumor response.

2) The authors propose that T cells with higher affinity TCRs are preferentially recruited into the PD-1^{hi} Tsl population (Fig. 3), but the idea seems at odds with the finding that there is a similar distribution of TCF-1+/SLAMF6+, TCF-1+/SLAMF6- and TCF-1-ve populations among OT-I and polyclonal OVA/Kb specific cells (Figs. 1c, 2d). Wouldn't the expectation be that OT-I would align with one group (e.g. PD-1^{hi}, TCF-1+/SLAMF6+ if, as might be expected, OT-I represents a high affinity clone?). There also appears to be a similar decline in “TCR affinity index” among both OT-I and polyclonal OVA/Kb reactive cells when the authors examine SLAMF6+ Tsl, SLAMF6- Tsl and Teff cells, respectively (IgG controls, Ext. Fig. 8, panels b versus h). These data certainly do not support the idea of distinct TCR repertoires in those three subsets, since OT-I found in all of them, and differences in apparent TCR affinity occur even with a monoclonal population. This sounds more like “functional avidity” changes, that have been reported with TCR transgenic cells in distinct activation states (e.g., PMID: 11477407).

3) To explore the last point further, it will be important for the authors to examine OT-I cells in the same assays as used for polyclonal cells in Fig. 3e-j. If OT-I cells show similar correlations between PD-1 expression level and “affinity index”, the authors need to rethink their hypotheses: one implication of such a result would be that PD-1 blockade does not lead to loss of a subset of Tsl with high affinity TCRs, but rather leads to modulation of apparent TCR affinity (and, perhaps, differentiation of such cells), while preserving cells with that TCR in the repertoire.

4) The authors hypothesize that PD-1 blockade causes “terminal differentiation to the effector state or death of the most avid anti-tumor stem-like cells” (lines 37-38, with similar statements throughout). If the first part of this prediction were correct, one might predict that PD-1 (or PD-L1/2) blockade would lead to loss of high affinity index cells in the Tsl pool – and appearance of such cells in the Teff pool. In other words, the most avid cells would be “chased” from the stem-like to the effector pool. There appears to be minimal evidence for such a transition in Fig. 5g.

5) The authors are correct to note that PD-1 (or PD-L1/2) checkpoint blockade leads to reduced frequencies of SLAMF6+ Tsl (Fig. 5b) but their own data show that the largest element in this effect is expansion of the SLAMF6- Tsl and Teff (Fig. 5c). Numerically, the decline in numbers of SLAMF6+ Tsl is about 2-fold (Fig. 5c). While potentially important, it is not clear that the authors can justifiably say that this population has been “lost” – it is more accurate to say that other populations have been expanded. This limits the impact of the studies, since it argues against the idea that a key stem-like population is irretrievably lost by checkpoint blockade – rather the data indicate that this population is (at least transiently) outnumbered.

6) Developing the previous point, it would be useful for the authors to show what happens when they examine the tdLN many days after the last anti-PD-1 (or anti-PD-L1/2) treatment: Is there, in fact, evidence that the SLAMF6+ Tsl population is permanently lost, or do these cells “bounce back” following decay of the blockade antibodies? This reviewer appreciates that the experimental design may be challenging (if the tumor either overgrows or is eliminated, interpretation would be difficult), but this would seem to be critical to test a fundamental element in the novelty and impact of the study. If the SLAMF6+ Tsl cells do recover in tdLN (in mice that maintain their tumors), this may suggest elements of the phenotypic changes reported (e.g., reduced TCF-1 or SLAMF6 expression levels – Fig. 5d,e) may in fact be reversible. Such a finding would limit the clinical significance of the report.

7) A potential consequence of checkpoint blockade would be that, by unleashing improved CD8+ T cell effector functions, tumor antigen specific CD8+ T cells start killing antigen bearing DC. Hence, just like the authors' studies in which cDC1 were eliminated (Fig. 3c,d), checkpoint blockade would cause (partial) loss of SLAMF6+ Tsl, due to abrupt loss of antigenic restimulation. Have the authors examined this, and if so, how do they control for that interpretation?

(Remarks on code availability)

Well outside my area of expertise.

Referee #3

(Remarks to the Author)

To the author,

Congratulations on a very compelling manuscript and highly relevant findings.

In the manuscript titled “PD-1 controls differentiation, survival, and TCR affinity evolution of stem-like CD8+ T cells” by Jyh Liang Hor et al. the investigators apply 3D multiplex immunofluorescence imaging to study stem-like CD8+ T cells in the spatial ecosystem of the tumor draining lymph node.

Summary of the key results:

The manuscript describes multiple novel and paradigm shifting findings that generally justify a publication in “Nature”. 1st they demonstrate that in contrary to the prevailing view, continued TCR signaling by high affinity TCRs does not necessarily lead to terminal TEFF differentiation and exhaustion. 2nd they elucidate, at least partly, the mechanisms by which dDC1 in the tdLN counterbalance TCR signaling by providing PD-1-stimulating signals. 3rd they describe a novel mechanism of high

affinity TCR T cell selection, in which prolonged antigen presentation and PD1-stimulating signals (PD-L1/2 expression) by dDC1 cells in the tdLN promote the survival and expansion of high affinity TCR TSL T cell clones. 4th they show that PD1/PD-L1/2 blockade interferes with this process resulting in progressive loss of high affinity TCR T cell. These findings have major implications on our understanding of high affinity TCR T cell selection and might influence our clinical application of PD1/PD-L1/2 blocking antibodies.

Originality and significance: The findings are of highest originality and significance.

Overall, the manuscript is written very well. That data presentation is adequate and supports the message. Statistics are used appropriately.

There are some remaining questions and there might be space for improvement at certain point, I would like to discuss point-by-point in the following:

Line 90: Could you please provide data or references supporting the statement that day8 represents the “peak expansion and differentiation stage”.

Figure 1: It is stated that all data shown are representative of at least 3 independent experiments, with n=2-3 per experiment. Could you please provide quantification of T-DC clusters etc. from all animals with arrow bars to demonstrate consistency of findings.

Figure 1g: Quantification of nuclear NAFT positive/negative OT-1 cells in correlation to distance tDCs and cluster distant areas of the tdLN distance would help to support the statement.

Line 271-273: It is stated the high affinity TSL (most likely) give rise to high affinity TEFF based on the finding that the TCR affinity of TEFF lacks behind. This is rather weak data to support this mechanistically very important finding. Since the OT-1 input is polyclonal, it should be possible to trace T cell clones by TCR sequencing. At this stage and later, looking at the effects of PD1/PD-L1/2 blockade, the data set would very much profit from an experiment tracing the faith of the polyclonal T cell pool. Like in the experiment resulting in Figure 3 i-j, one could utilize T cells (TSL and TEFF) from the tdLN at different timepoint for TCR sequencing to understand the clonality of the repertoire. One could additionally sequence T cells from peripheral blood and tumor to understand distribution. Ideally one would perform scRNASeq including VDJ-Seq from all cells of the tdLN, blood, tumor. This would additionally generate data to support the mechanism of TEFF promotion within the T-DC clusters. Like this one could follow how specific TSL clones transform into TEFF as well as their distribution tdLN vs. peripheral blood vs. tumor.

Line 304-305 As outlined above, scRNASeq combined with VDJ-Seq at different timepoints could provide significant additional information on cell-cell interaction and mechanistic insights. One could perform velocity analyses tracing the transformation of cell states. How is PD-L1 and PD-L2 expression on DCs regulated? What is inducing the T-DC clusters? Do T cell derived signal, e.g. INFg, induce PD-L1 and PD-L2 expression? Are other co-stimulatory / inhibitory signals involved in the regulation of the stem-like state?

Line 352: It is stated that the effects of PD-1 blockade on the TSL clonal repertoire is examined. However, this is only supported by changes in the “affinity index”. Again, the manuscript would strongly benefit for TCR sequencing data. One would expect an oligomerization of clonal diversity due to selection of high affinity clones, which is screwed by PD1/PD-L1/2 blockade.

Discussion:

Line 466-469: Please elaborate on how a partial blockade of the PD-1 pathway could look like in reality and how that would fit with the proposed mechanism. Is selection of high affinity T cells in patients a timely limited process or consciously ongoing and how would you time “partial blockade”. Is there any evidence from patients that TCR affinity decreases during checkpoint blockade, supporting the hypothesis that high affinity TCR T cells experience AICD?

Please discuss the implications of your findings on cancer vaccine approaches. Multiple studies have demonstrated induction of mutation specific T cell by either peptide or mRNA neoepitope vaccines. As of today, it is unclear, whether vaccines should be combined with PD1/PD-L1/2 blockade and in which sequence. The presented data would argue to hold checkpoint blocked during the vaccination phase to allow high affinity T cells to be induced and selected.

Version 1:

Reviewer comments:

Referee #1

(Remarks to the Author)

The revised manuscript is improved and more thorough. The authors addressed many of my concerns with thoughtful explanations and additional data. I especially appreciate the new analysis of T cell subsets. However, I remain unconvinced by their approach to assessing changes in TCR affinity between TSL and Teff cells. As noted previously, the tetramer binding assay used does not directly measure affinity—it measures avidity—yet the manuscript continues to use the term "affinity" broadly and inaccurately. This was also noted as a point of concern by Rev. 2.

Their analysis is simply based on averaging tetramer MFI normalized to CD3 levels, and while this is more rigorous to account for total surface TCR levels, it still remains a measure of avidity, not affinity. Given that changes in "affinity" are central to the authors' proposed model for how checkpoint blockade affects T cells, this issue must be addressed more rigorously.

That the authors see similar subsets in OT-1 T cells that have the exact same affinity TCR again, speaks to the point that the authors are observing changes in TCR avidity in the different subsets of cells, not affinity.

Thus, at a bare minimum, the authors need to revise the terminology to avidity when this is discussed to ensure a more accurate description of the assays and results.

The authors go through more rebuttal on this point for Reviewer 2 and then introduce CD8 levels and how that can affect TCR affinity. They try to normalize the tetramer staining to CD8 levels and while this Reviewer appreciates and can see why they are examining this third factor in TCR avidity (not affinity), the fact that the authors are normalizing against two variables makes one feel even less confident.

In my original review, I suggested the inclusion of TCR sequencing (e.g., scTCR-seq) to assess clonality across TSL and Teff subsets. This remains critical. Without evidence of clonal overlap or distinction, it is not possible to determine whether observed differences reflect changes in true affinity or simply changes in clonal TCR composition and avidity. If the same clones are present in both subsets, then the observed differences are clearly due to avidity. If distinct clones dominate each subset, and this is enhanced by ICB, then a shift in affinity may be inferred. That the authors interpret the increased in affinity over time is due to the selection of higher affinity clones would be directly observed in a scTCR-seq analysis too. It is fine if technical barriers prevent the tetramer staining comparison across tDLNs and tumors, but this should not be true for scTCR seq as this is done routinely with small numbers of cells and within tumors and tDLNs. The technical barriers to performing TCR repertoire analysis are relatively low, and not incredibly costly, thus, this reviewer feels this data is really important to support the authors' conclusions.

Referee #2

(Remarks to the Author)

The authors have done an excellent job of extending their studies further and resolving what appeared to be contradictory findings. There is still a concern, however, about the overall framing of this work as an investigation into the CD8+ T cells which constitute a reservoir of anti-tumor cells in the draining lymph nodes (dLNs) (related to reviewer comment 2.1).

While it was valuable that the authors clarify their meaning of "late" (i.e. the "initial late differentiation phase" after priming), this does not address a central concern with overall interpretation and context of these studies.

The authors write that "Emerging evidence suggests that tumor-draining lymph nodes (tdLNs) serve as reservoirs from which freshly generated TSL-like cells or their immediate progeny continuously replenish the tumor microenvironment (TME) and provide a source of functionally competent effector cells during checkpoint therapy [refs] 3-6." (lines 62-65) and go to say that investigating the characteristics of cells in the reservoir is a central focus of this study (e.g., lines 30-33, 67-73). Yet, as they acknowledge in the rebuttal, their work "does not represent a much later period during which T cells are expected to experience exhaustion/dysfunction due to excessive TCR signaling in the tumor microenvironment." But that IS the central point of studies such as those in references 3 and 4 – which indicated that dLNs were a key reservoir of cells that entered the tumor long after induction of local exhaustion (in those studies, months after tumor induction). While this reviewer appreciates that different tumor models will have different kinetics, the current studies investigate cells involved in the initial response to tumor – it is not clear that any of these findings relate to T cells that contribute to the long-term establishment of a reservoir of tumor-specific CD8+ T cells in the dLN.

That does not mean the work is without considerable merit – and, again, the other responses by the authors resolved key concerns – but the text needs to be revised to reflect the fact that the early time points studied preclude simple extrapolation from the data shown to the characteristics of the durable reservoir of anti-tumor T cells maintained in draining LNs. Casting this study as an exploration of the properties of the anti-tumor reservoir is, at a minimum, premature.

Referee #3

(Remarks to the Author)

To the authors of the manuscript: "PD-1 controls differentiation, survival, and TCR affinity evolution of stem-like CD8+ T cells"

Thank you for sufficiently addressing and clarifying all raised points. I do understand that large scale scRNASeq/scTCRSeq experiments, though most likely mechanistically informative, are beyond the scope of this manuscript. I do not have additional point to be addressed and would support the publication of the revised version of the manuscript.

Version 2:

Reviewer comments:

Referee #1

(Remarks to the Author)

I want to sincerely thank the authors for thoughtfully addressing my follow-up comments and for their clear understanding of the nuances involved—details that may appear subtle on the surface but significantly impact interpretation.

That said, I remain somewhat surprised by the continued use of the term affinity rather than adopting a more cautious framing around avidity, especially given that the data do not directly demonstrate affinity changes. While I fully agree that demonstrating this definitively would require biochemical measurements beyond the scope of the current study—and thus was not expected or asked for—I had hoped for a more conservative interpretation in the manuscript. I appreciate the authors' thoughtful discussion of the distinction between TCR affinity and avidity, and their acknowledgment of the nuances involved. While the proposed use of the term "imputed affinity" is an improvement, I still find it somewhat surprising that they prefer to include a caveat around this term rather than simply use avidity, which more accurately reflects the measurements performed and where the strongest evidence lies. That said, my intent is not to dwell on this point or engage in back-and-forth debate, as I remain a strong supporter of the manuscript overall. However, I do feel the authors' insistence on using this terminology may invite more scrutiny and raise unnecessary doubts about the study's conclusions, rather than strengthening its overall conclusions. I am comfortable with the inclusion of imputed affinity, provided the authors clearly articulate its limitations and explicitly state that their assays primarily reflect changes in TCR avidity as they mention in their response.

Similarly, as the current title makes a strong claim—that PD-1 controls TCR affinity evolution—I would respectfully encourage the authors to reconsider whether this wording is fully supported by the data presented. The study does not directly measure TCR affinity, nor does it characterize repertoire shifts following PD-1 blockade. In the interest of clarity and accuracy, a revised title that more precisely reflects the study's actual observations would help strengthen the overall impact and interpretation of the work.

Version 3:

Reviewer comments:

Referee #1

(Remarks to the Author)

This reviewer is fine with the revisions.

Response to the Reviewers' Comments

Reviewer 1

Hor et al. conducted an in-depth investigation into the mechanisms and dynamics of CD8+ T cell priming to tumor antigens within the tumor-draining lymph node (tdLN) during the early stages of tumor implantation. By combining advanced imaging techniques with flow cytometry, the authors provide key insights into T cell-dendritic cell (DC) dynamics and the origins of an anti-tumor CD8+ T cell response. Utilizing the KP-Ova (NSCLC) model implanted subcutaneously, they monitored OVA-specific OT-1 T cell responses and made several significant observations:

1. T Cell Interactions: The authors convincingly show that the TSL (PD-1^{int} SLAMF6⁺ TCF1⁺) cells engage in tight interactions shortly after tumor implantation. These cells are largely Ki67⁺, indicating active proliferation. They further differentiate these cells from OT-1 cells that have downregulated TCF1, which they refer to as Teff cells. Interestingly, the authors find that the effector-like cells in the tumor express the highest levels of PD-1, a pattern not observed in tissues less likely to contain tumor antigens. This cross-tissue comparison adds depth to the study.
2. Role of cDC1s in TSL Maintenance: Their data suggest that cDC1s play a crucial role in sustaining TSL cells in the tdLN. When cDC1s are deleted, TSL cell numbers drop by approximately fourfold, while Teff cell numbers remain unaffected.
3. TCR Affinity of TSL and Teff Cells: Using a tetramer-binding assay, they propose that TSL cells have higher-affinity TCRs than Teff cells, indicating that the TSL pool maintains the highest-affinity tumor-reactive T cells.
4. Effect of PD-1 Blockade: Blocking PD-1 disrupts cDC-TSL clusters and promotes Teff differentiation but also appears to reduce TCR affinity.
5. Differential Effects of PD-L1/PD-L2 Blockade: Blocking PD-L1 or PD-L2 has a more modest effect on tumor growth and does not result in a loss of high-affinity TSL cells. This distinction between PD-1 and PD-L1/PD-L2 blockade may have important clinical implications, particularly since more than half of patients who initially respond to ICB later develop acquired resistance within 1-3 years.

Overall, this study represents a high-quality, rigorous contribution to the field. Given the clinical implications, comparative analyses like this are vital and will undoubtedly be of broad interest. However, the findings raise several important questions and suggestions for further clarification and exploration:

We appreciate that the reviewer finds our work to be important and of broad interest to the field. We have since performed additional experiments and substantially expanded on the questions raised by the reviewer in our revised manuscript.

Major:

R1.1: Intratumoral TSL: The authors identify TSL cells in the tdLN in Figure 2d but do not provide adequate identification of these cells within the tumor. While the intratumoral TSL cells seem to express lower Slamf6 levels than in the tdLN, the small subset of TCF1hi SLAMF6int cells in the tumor represents the intratumoral TSL population that appears to have been overlooked by the authors. It would be important to incorporate a deeper analysis of this subset within the tumors, comparing its location relative to cDCs and assessing its TCR affinity.

We thank the reviewer for raising the important question of the relative contributions of LN and intratumoral SLAMF6-int T_{SL} as the progenitors of effector and exhausted T cells.

We note that intratumoral SLAMF6-int TCF-1+ T_{SL} (as indicated in **Fig. 2d**, also designated as SLAMF6- T_{SL} in the main text) have been extensively characterized in both human (*Siddiqui et al., 2019: 10.1016/j.immuni.2018.12.021*; *Miller et al., 2019: 10.1038/s41590-019-0312-6*) and mouse (*Schenkel et al., 2021: 10.1016/j.immuni.2021.08.026*, *Escobar et al., 2023: 10.1016/j.ccell.2023.08.001*, *Lan et al., 2024: 10.1038/s41590-024-01843-8*) studies, as well as in a study that directly compared this subset in the tumor-draining LN (tdLN) and tumor microenvironment (TME), and in both mouse and human lung cancer samples (*Connolly et al., 2021: 10.1126/sciimmunol.abg7836*).

The spatial mapping of TCF-1+ progenitors in the tumor has also been described in several human studies, where these subsets were reported to associate with MHC-II+ DC niches (*Jansen et al., 2019: 10.1038/s41586-019-1836-5*), within “stem-immunity hubs” containing mregDC (that would include cDC1) (*Magen et al., 2023: 10.1038/s41591-023-02345-0*, *Chen et al., 2024: 10.1038/s41590-024-01792-2*), within tumoral tertiary lymphoid structures (*Im et al., 2023: 10.1073/pnas.2221985120*); as well as in DC-rich tumor region in mice (*Lan et al., 2024: 10.1038/s41590-024-01843-8*). These studies collectively suggested that tumor-infiltrating TCF-1+ CD8+ progenitors preferentially localize to sites of antigen presentation in the TME.

Regarding the TCR affinity of the intratumoral T_{SL}, a recent study by *Lan et al. (2024: 10.1038/s41590-024-01843-8)* showed that low avidity T_{SL} were unable to persist/self-renew in the TME due to suboptimal TCR signaling and experienced accelerated terminal effector differentiation. As will be discussed below in **R1.5**, our method of determining affinity index

of the T cells does not allow for direct comparison between LN and tumor-infiltrating T cells, due to the additional enzymatic digestion (90 min at 37°C) performed on tumor samples that could alter TCR and CD8 expression levels.

As such, we do not believe that additional examination of this specific issue that has been studied and reported by numerous groups would add substantially to the value of our manuscript.

Instead, our manuscript follows on recent reports that the tdLN serves as a source of continuous TCF-1+ T_{SL} generation that replenish the TME, as these progenitor cells would become exhausted/dysfunctional in the TME due to persistent antigen stimulation (*Connolly et al., 2021: 10.1126/sciimmunol.abg7836, Schenkel et al., 2021: 10.1016/j.immuni.2021.08.026, Li et al., 2022: 10.1084/jem.20210749, Rahim et al., 2023: 10.1016/j.cell.2023.02.021*). **The primary focus of our manuscript is thus centered on how these more stem-like precursors of T_{SL} progenitors are initially generated in the tdLN prior to replenishing the TME.**

Our findings revealed that a T_{SL} population enriched in high SLAMF6 expression (SLAMF6-hi T_{SL}, designated as SLAMF6+ T_{SL} in the main text) is uniquely found in the tdLN. To further emphasize the distinct properties of these LN SLAMF6+ T_{SL} subset compared to the more widely published SLAMF6- T_{SL} (that are associated with more limited self-renewal capacity as reported by *Schenkel et al., 2021: 10.1016/j.immuni.2021.08.026*), in the revised manuscript we have now provided additional evidence for the high degree of stemness of these LN-resident SLAMF6-hi T_{SL}. We stained for histone 2A lysine 119 ubiquitination (H2AK119Ub), which has recently been identified as a key epigenetic marker for stem-like/quiescent T cells (*Kang et al., 2024: 10.1126/science.adl4492*), and found that the LN SLAMF6+ T_{SL} expressed the highest level of H2AK119Ub as compared to SLAMF6- T_{SL} and TCF-1- T_{EFF} found in both the tdLN and the spleen (updated **Extended Data Fig. 4f,g**).

Additionally, the transcription factor LEF1 may play a potential role in maintaining the stem-like state in T cells (unpublished by *Schietinger et al.*). Therefore, we also stained for LEF1 and found that LEF1 expression correlated with SLAMF6 expression and is enriched in the LN-associated SLAMF6+ T_{SL} (**Fig. R1**).

Together, given the unique localization of SLAMF6+ T_{SL} with true stemness markers in the tdLN, we believe that a key contribution of this manuscript lies in elucidating the function and the mechanism by which these SLAMF6+ T_{SL} can be sustained and expanded in the tdLN.

Figure R1. LEF1 staining of T_{SL} and T_{EFF} subsets.

a-b, Relative expression of LEF1 on different subsets of OVA-tetramer+ cells from the tdLN and spleen at Day 10, shown in representative histograms (**a**) and in mean fluorescence intensities (normalized to naïve TCF-1+ PD-1- CD8+ T cells) (**b**). Data from 1 experiment (n=3). Error bars: means \pm s.d. ***p<0.001, ****p<0.0001; one-way ANOVA with Tukey's multiple comparisons.

R1.2: Additionally, the lack of analysis of this population in the context of PD-1 or PD-L1/L2 blockade is a notable omission. Including this population in the primary experiments would help to elucidate its potential as a local source of T_{eff} and terminally exhausted T cells within the tumor.

We have taken the reviewer's suggestion and examined the intratumoral OVA-specific TCF-1-int SLAMF6-int T_{SL} and T_{EFF} populations in the context of anti-PD-L1/2 blockade, which yielded T cells with even lower TCF-1 and SLAMF6 expression after treatment in a similar fashion to their LN counterparts (**Fig. R2**).

We have been unable to determine the relative contribution of tdLN vs. intratumoral T_{SL} in the generation of functional effector cells by direct experiment because the most reasonable experiment, blockade of tdLN egress with FTY720 prior to ICB, results in a marked diminution of T_{SL} even in the tdLN (**Fig. R3**). Additionally, a recent report by Lan *et al.* (2024: 10.1038/s41590-024-01843-8), also employing FTY720 treatment at 6 days after priming, found significant reduction but not complete elimination of T_{SL} in tumor that had a high avidity for antigen. In contrast, in the same study, infiltration of tumor antigen-specific CD8+ T cells into TME containing low avidity antigen was substantially reduced with or without FTY720 treatment, suggesting that the replenishment rate of low avidity TCF-1+ T_{SL} into the TME was low.

Figure R2. Comparison of phenotypic changes in OVA-specific CD8+ T cells in the tdLN and tumor after checkpoint blockade.

a, d, Representative flow cytometric plots of TCF-1 and SLAMF6 expression on OVA-tetramer+ cells recovered from the tdLN (**a**) and tumor (**d**) at Day 12. Quantification of relative TCF-1 (**b, e**) and SLAMF6 (**c, f**) expression of TCF-1+ T_{SL} subset (normalized to naïve TCF-1+ PD-1- CD8+ T cells in the tdLN) in the tdLN (**b, c**) and tumor (**e, f**). Data from 1 experiment (n=3 per treatment group). Error bars: means ± s.d.

Figure R3. FTY720 treatment during early time point abrogates TSL expansion in the tdLN.

a, Experimental scheme of FTY720 and DT treatment on XCR1-DTR mice. **b**, Quantification of OVA-tetramer+ CD8+ TCF-1+ T_{SL} and TCF-1- T_{EFF} cells recovered from each group of mice at Day 10. Error bars: means ± s.d.

We also note that the number of SLAMF6-int T_{SL} was at least 1 order of magnitude greater in the tdLN than in the entire tumor and by at least 2 orders of magnitude greater in the spleen compared to the tumor (**Fig. 3c, d, Extended Data Fig. 5g, h**). While the SLAMF6-int T_{SL} were further amplified after checkpoint blockade (**Fig. 5c**), these cells also exhibited reduced stemness markers including TCF-1 and SLAMF6 (**Fig. 5d, e**). As such, from a quantitative perspective alone, tumor antigen-specific SLAMF6-int T_{SL} in the tdLN and the spleen far outnumbered the SLAMF6-int T_{SL} found in tumor. We agree that investigation into the relative contribution of the site-specific progenitor-like T cells in future studies will help further elucidate the mechanisms by which PD-1 checkpoint blockade drives effector generation and promote effective tumor control.

R1.3: Furthermore, it would be beneficial to confirm that Teff cells are indeed effector cells. Could the authors assess functional markers such as IFN γ , TNF α , GzmB, and transcription factors like T-bet and Eomes in these populations to ensure their definitions are correct, particularly as they are PD-1 lo in the tdLN and other tissues?

The authors should also acknowledge terminally exhausted T cells, as this would conceptually enhance the study. Incorporating markers like CD101, CXCR6, and CX3CR1 into flow cytometric analyses would help to distinguish effector T cells from more terminally exhausted subsets, as intratumoral PD-1 hi /TCF1 lo cells are likely heterogeneous. Although the authors may argue that most time points are early after tumor implantation, and terminally exhausted cells are likely scarce, better delineation of intratumoral CD8 T cells is essential.

We agree with the reviewer that additional studies of the state of effector cells were warranted and we have now stained for CX3CR1, CD39 and GzmB, and also performed *ex vivo* peptide stimulation followed by staining for the intracellular cytokines IFN γ and TNF α to determine if the TCF-1- T_{EFF} in the tdLN are indeed functional effector cells. We confirmed that tdLN TCF-1- PD-1- lo T_{EFF} cells expressed the highest level of CX3CR1, CD39 and GzmB – markers, characteristic of potent cytolytic effector cells (*Miller et al., 2019: 10.1038/s41590-019-0312-6; Zander et al., 2019: 10.1016/j.immuni.2019.10.009; Duhon et al., 2018: 10.1038/s41467-018-05072-0; Simoni et al., 2019: 10.1038/s41586-018-0130-2; Chow et al., 2023: 10.1016/j.immuni.2022.12.001*) (updated **Extended Data Fig. 4h-k, Fig. R4a-d**). The expression of these markers was further amplified with anti-PD-L1/2 checkpoint blockade (**Fig. R4c-d**).

Additionally, unlike in the tdLN where their PD-1 expression was low, in the tumor, CD39 and GzmB were indeed strongly correlated with PD-1 expression in the TCF-1- T_{EFF} subset, further

suggesting that the PD-1-lo TCF-1- cells detected in the tdLN were functional effector cells that were phenotypically comparable to the PD-1+ effector cells found in tumors in many other studies (**Fig. R4e-f**). These data are consistent with the role of antigen (higher in the TME than in the tdLN) in driving elevation of PD-1 expression and with the TCF-1- T_{EFF} in the tdLN with low PD-1 having migrated away from the predominant XCR1+ cDC1 antigen presenting population that occupies the T cell zone (see **Fig. 2d**).

Ex vivo peptide stimulation and staining for IFN γ and TNF α also revealed that TCF-1- T_{EFF} subset exhibited the highest IFN γ and TNF α staining intensity in both the tdLN and tumor, and production of these effector cytokines was further elevated when mice were treated with anti-PD-L1/2 blocking antibodies (**Fig. R5**). These data are also consistent with the view that the TCF-1- PD-1-lo T_{EFF} in the tdLN have acquired an effector phenotype but were not engaging their TCR, resulting in low PD-1 expression and lack of cytokine production while resident in the LN in the absence of ICB.

Altogether, we have demonstrated that TCF-1- T_{EFF} cells in the tdLN are phenotypically and functionally similar to the intratumoral T_{EFF} published elsewhere.

Figure R4. Functional effector molecule expression by T_{EFF}.

a-d, Relative expression of Granzyme B by different subsets of OVA-tetramer+ cells from Day 12 tdLN, shown in as representative histograms (**a**) and as mean fluorescence intensity (normalized to naïve TCF-1+ PD-1- CD8+ T cells) (**b**). The mean fluorescence intensity of CX3CR1 (**c**) and Granzyme B (**d**) of each subset upon anti-PD-L1/2 checkpoint treatment. Data from 1 experiment (n=3). **e-f**, Co-expression of PD-1, CD39 and Granzyme B in TCF-1- T_{EFF} in the tumor. Low PD-1 expression was found in the tdLN T_{EFF} subset as shown in the manuscript. Data from 1 experiment (n=3) Error bars: means \pm s.d.

Figure R5. Ex vivo peptide stimulation of cytokine expression.

Lymphocytes isolated from Day 12 tdLN (a) and tumor (b) were cultured ex vivo in the presence of 10^{-10} M SIINFEKL peptide and Brefeldin A for 5 hours at 37°C and subsequently stained for intracellular IFN γ and TNF α . Flow cytometric gating on activated (CD44+ PD-1+) CD8+ T cells revealed an increased proportion of IFN γ + TNF α + T_{EFF} from animals treated with anti-PD-L1/2, indicating the functional effector states of T_{EFF} recovered from both tdLN and tumor. Data from 1 experiment (n=3). Error bars: means \pm s.d.

R1.4: TCR Affinity and Repertoire Measurements: The authors interpret their tetramer-binding data to suggest that TSL cells have higher-affinity TCRs than Teff cells. However, this assay actually measures TCR avidity, which incorporates both TCR affinity and the higher-order interactions between TCR complexes. This interpretation becomes more complex in the context of ICB, where a reduction in “affinity” is observed in expanding Teff populations. If TSL cells are clonal precursors to the newly generated Teff cells, shouldn’t the TCR affinity be similar? The observed differences between TSL and Teff tetramer binding could indicate a drop in TCR avidity in Teff cells or the emergence of low-affinity clones that infiltrate the tumor, or both. The authors need to examine the TCR repertoire more closely. Single-cell TCR sequencing of tdLN and intratumoral T cells could provide crucial distinctions between these models. Especially since previous studies have shown that anti-PD-1 therapy leads to an influx of new TCR clones into tumors, it would be critical for the authors to integrate this knowledge into their analysis.

Regarding the reduction in “TCR affinity” in the expanding T_{EFF} population after ICB, it should be noted that our plot provides the mean TCR affinity index across the entire subset and considering the exponential increase in the absolute count of T_{EFF} after ICB in the tdLN and the spleen, this relationship is best captured in the linear modeling as shown in Fig. 5i. This quantitative analysis showed that after ICB, “high affinity” T_{EFF} substantially expanded though not to the same magnitude as the lower affinity T_{EFF}. We have further clarified this observation in the main text (lines 379-390) to suggest that the clonal TCR affinity shift after

ICB is likely driven by a combination of increased expansion of lower affinity clones, together with the clones at the higher affinity range that had received excessive stimulation diminishing due to differentiation into effectors or activation induced cell death after blockade of PD-1 inhibition.

Single cell sequencing would not allow us to evaluate affinity, only examine clonality; while this could be of some value in relating effectors in the TME to the stem-like cells in the tdLN, many others have already connected these populations. Notably *Schenkel et al. (2021: 10.1016/j.immuni.2021.08.026)* also reported shared clonality between SLAMF6+ and SLAMF6- stem-like cells, while others have demonstrated shared TCRs between blood and TIL clonotypes after checkpoint therapy (e.g., *Huang et al., 2017: 10.1038/nature22079*, *Wu et al., 2020: 10.1038/s41586-020-2056-8*). Therefore, we believe that the expensive and time consuming sequencing studies needed to gain such clonal information would not add substantially to our message.

R1.5: It would also be valuable for the authors to incorporate findings from Dr. Andrea Schietinger's work (<https://doi.org/10.1084/jem.20201966>). The Schietinger lab found that the higher affinity antigens induced more T cell exhaustion, but the lower affinity maintained more effector-like cells that expressed higher levels of Tcf7 and other genes in TSL cells. This is somewhat at odds with the present manuscripts interpretation albeit this manuscript focuses on the tdLN whereas the Schietinger study focused on intratumoral cells. Hence, the need to compare the cells within the two sites as described in #1 above. Use of such APL models could also really help to strengthen the present manuscripts claims, and doing so would be highly encouraged to increase the relevance of the conclusions, but this Reviewer understands that this would be a large undertaking and may not be feasible.

Our work is not inconsistent with *Schietinger et al.* in that higher affinity cells, although having better killing capacity against tumor cells as reported in the literature, means these cells are likely to experience a faster rate of exhaustion when encountering high antigen levels in the tumor microenvironment (already included in the **Discussion** section at lines 536-539). The extent of antigen presentation by cDC1 in the tdLN, as estimated using PD-1 expression level on T cells, is far lower compared to the tumor sites (see **Fig. 2d** and the **Discussion** section at lines 536-539).

Regarding the use of APL models, as included in the **Discussion** section (lines 504-505), our findings are consistent with the recent work from *Lan et al. (2024: 10.1038/s41590-926)* in which Lewis lung carcinoma cells (LLC) expressing altered LCMV gp33 epitopes were used to examine how high versus low TCR signaling strength influences CD8+ T cell differentiation.

The use of a high TCR strength epitope tumor model resulted in enhanced SLAMF6+ stem-like cell formation, whereas lower TCR strength drove rapid effector differentiation. These observations are fully consistent with our findings.

Additionally, we have data from a preliminary experiment using mixed tumor model (mixing OVA-expressing and non-OVA expressing tumor in 1:1 ratio prior to injection) to lower the antigen abundance in the tumor. Our results showed that less antigen presentation drove a more TCF-1- T_{EFF} dominant phenotype among OVA-specific CD8+ T cells in the tdLN (**Fig. R6a-e**). Furthermore, the affinity index was not appreciably different (**Fig. R6f-g**), suggesting that reduced amount of antigen presentation sites could still drive the selection for high affinity clones, although the number of high affinity cells adopting this phenotype and/or the magnitude of their expansion was limited with less antigen availability. Consistent with the *Lan et al.* study above, this led to a large population of OVA-specific CD8+ T cells proceeding to terminal differentiation at a rate that appears to be much higher than the self-renewal rate

Figure R6. Limited antigen presentation constrains TCF-1+ T_{SL} expansion.

Mice were injected intradermally with either OVA-expressing tumor (KP-OVA) or a 50% mixture of OVA-expressing and non-OVA expressing parental KP tumor (KP-1233). **a**, Representative flow cytometric plots of TCF-1 and SLAMF6 expression of OVA-tetramer+ CD8+ T cells from Day 12 tdLN and the gating strategy to define the three major T_{SL} and T_{EFF} subsets. Total numbers of each subset from the tdLN (**b**) and spleen (**c**). Note the dominance of TCF-1- T_{EFF} when OVA antigen abundance was low. SLAMF6+ T_{SL} differentiation was not impaired as shown in the mean fluorescence intensity plots of relative TCF-1 (**d**) and SLAMF6 (**e**) expression of OVA-tetramer+ TCF-1+ T_{SL} subset. Standardized log transformed TCR affinity index (**f**) and mean affinity index (**g**) of each treatment group, normalized to SLAMF6+ T_{SL} recovered from KP-OVA-injected mice, revealing that evolution of high affinity SLAMF6+ T_{SL} was not impaired with reduced antigen level. Data from 1 experiment (n=3 from each group). Error bars: means ± s.d. *p<0.05, **p<0.01; unpaired two-tailed t-test.

of TCF-1+ stem-like cells. We have included these data for the reviewer and can add them to the extended data if this is deemed important.

Determining the affinity index of T cells in the TME is problematic. As noted by this and the other reviewers, tetramer binding is highly influenced by CD8 co-receptor expression and local TCR density. We have shown that these features do not markedly affect the conclusions we draw about the T cells within the tdLN (see our response to **R2.3** below, and the updated **Extended Data Figure 9j**), but this is not true for cells in the TME. Repetitive, high level TCR engagement with antigen, as indicated by the high PD-1 expression of the T_{EFF} in the TME (**Fig. 2d**), leads to very marked loss of CD8 and TCR expression through internalization and removal by trogocytosis. Further, to extract lymphocytes from the tumor samples as compared to the tdLN, we need to perform enzymatic digestion, and this together with the prolonged incubation (90 min) at 37°C during the digestion step can affect the surface expression of various molecules including CD8 and the TCR. As a result, these effects make the ratiometric test we employ less reliable, preventing a direct comparison of the TCR affinity of the T cells in the tdLN and TME to directly answer the question raised by the reviewer.

R1.6: Clinical Relevance: The finding that anti-PD-L1 or PD-L2 monotherapy may better preserve TSL cells compared to anti-PD-1 monotherapy is intriguing. Do clinical data support this distinction in terms of T cell responses or durability of anti-tumor immunity? While anti-PD-1 and anti-PD-L1 therapies yield comparable responses as monotherapies, is there any longer-term data on progression-free survival (PFS) or overall survival suggesting that anti-PD-L1 might offer more durable benefits? Can the authors find any evidence of this from clinical studies?

We have performed a literature search and examined five clinical studies published in the last few years that compared pembrolizumab and nivolumab (anti-PD-1) versus atezolizumab (anti-PD-L1) in advanced NSCLC patients (*Passiglia et al., 2017: 10.1002/ijc.31136; Weis et al., 2019: 10.1177/1078155219855; Ramagopalan et al., 2021: 10.1001/jamanetworkopen.2021.34299; Alonso-Garcia et al., 2022: 10.3390/ph15050533; Ham et al., 2023: 10.3390/cancers15164198*).

There were no significant difference in the overall survival (OS) and progression-free survival (PFS) between anti-PD-1 and anti-PD-L1 treatments in all of these studies; however, in three of the studies (*Passiglia et al., 2017; Alonso-Garcia et al., 2022; Ham et al., 2023*), the objective response rate (ORR) was significantly higher in patients that underwent anti-PD-1

therapy. The latter observation would conform with the notion that anti-PD-1 treatment unleashes a more robust effector response as compared to anti-PD-L1 monotherapy.

However, it should be noted that our preclinical model shows that while anti-PD-L1 monotherapy better preserves the high affinity SLAMF6+ T_{SL} compartment, it led to less robust T_{EFF} generation and poorer tumor control compared to anti-PD-1 therapy (see **Extended Data Fig. 13**). As such, the comparisons between both types of treatments on the basis of survival and tumor relapse in mixed / distinct patient populations are difficult to interpret.

Minor:

R1.7: Tumor Models: While the authors used a subcutaneous model of lung cancer, it is surprising they did not examine more relevant models where tumors originate in the lung. The lung-specific LV-OVA model developed by Dr. Tyler Jacks would be particularly well suited for these studies.

While we agree that GEM models or orthotopic implantation can have some benefits as compared to s.c. models, we note that *Schenkel et al. (2021:10.1016/j.immuni.2021.08.026)* already used the KP-LucOS spontaneous lung tumor model to document a role for the tdLN in generating TCF-1+ progenitors that replenish the tumor sites. Given that we are focused on events in the draining LN and not the tumor, we do not believe that doing such time consuming studies (it takes months to develop tumors in these mice) would add substantially to our report.

R1.8: Ki67+ TSL Cells: The data in Figure 1 show that most TSL cells are Ki67+, yet they are not all rapidly dividing because their cell cycle is held in check by PD-1. Could the authors explore further which phase of the cell cycle these cells are in? Do the authors have any idea what allows for the spontaneous "release" of the TSL from PD-1-repression to allow for the differentiation of TSL into Teff and terminal Tex cells?

Our data show that SLAMF6+ T_{SL} express high levels of Ki-67 and are in active cell cycle (**Fig. R7**). TCF-1+ T_{SL} can be divided into Ki-67+ and Ki-67- subpopulations, and the Ki-67+ fraction is closely associated with the TCF-1-hi SLAMF6-hi PD-1+ subset (SLAMF6+ T_{SL}). This observation is congruent with our experimental data (**Fig. 3a-d**) demonstrating antigen-driven expansion of T_{SL}. It is also consistent with the data from *Schenkel et al. (2021)* that the SLAMF6+ TCF-1+ subset is associated with higher proliferative capacity and Ki-67 staining.

Additionally, experiments performed using OT-I cells labeled with proliferation dye revealed that the TCF-1+ PD-1+ OT-I cells clustering with cDC1 have extensively divided by Day 7 in the tdLN (see **Extended Data Fig. 1b-f**), suggesting that these cells were in active cell cycle despite retaining their stem-like state. They also have nuclear NFAT localization, indicating that they are also still actively signaling at some level through the TCR (updated **Fig. 1h-i**).

Based on these data, we propose that PD-1 inhibition **attenuates** strong TCR and co-stimulatory signaling and that this attenuation in turn prevents the T cells from being driven into a terminally differentiated state. However, these signals are still sufficient to activate proliferative programs, perhaps by maintaining a minimal necessary level of myc expression that is known to be rapidly lost when TCR signaling ceases (*Heinzel et al., 2017: 10.1038/ni.3598*).

Reviewer 2

The authors investigate the differentiation of stem-like CD8+ T cells in tumor-draining lymph nodes (tdLN), and the effect of PD-1 checkpoint blockade on the fate of these cells. Using 3D imaging techniques, the authors identify a population of stem-like CD8+ T cells (Tsl) (expressing TCF-1, PD-1 and SLAMF6) in the tdLN, and find that PD-1 expression by these cells correlates with high “TCR affinity index” for the model tumor antigen, consistent with the premise that cells with stronger reactivity to tumor antigens will have higher PD-1 expression. The authors explore the hypothesis that PD-1 expression protects these stem-like cells from premature differentiation into terminally differentiated effector cells (Teff), finding supportive evidence for this by observing a loss in frequency and reduced TCR affinity index in SLAMF6+ Tsl, following anti-PD-1 or PD-L1/2 blockade.

These studies are well conducted and certainly help define the stem-like population(s) that develop in tumor-draining lymph nodes, and the authors raise the interesting and important hypothesis that checkpoint blockade may essentially “burn-out” the most avid stem-like CD8+ T cells, leading to trade off enhanced short-term efficacy of tumor control with loss of the high affinity stem-like pool. Although the approaches are elegant and, for the most part informative, there are numerous issues that dampen enthusiasm.

We sincerely thank the reviewer for the constructive feedback and suggestions and have performed the necessary experiments as well as clarified the text to address the reviewer’s concerns.

R2.1: A general concern is that the authors examine quite early time points but interpret the data as analysis of “late” or “prolonged” antigen presentation. As the authors show, the same subsets - in similar proportions - are seen in the context of protein/adjuvant immunization (Ext. Fig. 2), which would presumably be a model of acute priming, against a non-replicating source of antigen – hence the argument that these cells are characteristic of “prolonged” antigen presentation is questionable. Furthermore, the authors premise leads them to interfere with the process during early stages of immune priming – day 4 (Figs. 4,5) and day 5 (Fig. 3): these studies may have little to do with the stated intent of this study which was to address the maintenance and role of the “reservoir” of stem-like antigen-specific CD8+ T cells in tdLNs (e.g., lines 23-27 and 67-72). To address that important issue, it would be more meaningful to study later time points, when the initial response is complete, and the maintenance of stem-like cells is relevant for sustaining the anti-tumor response.

We apologize for the confusion and have clarified in the main text that “late” in this context refers to the initial late differentiation phase (Day 7-9) after priming, by which time activated T cells have undergone proliferation and functional diversification (lines 91-95), and does not represent a much later period during which T cells are expected to experience exhaustion/dysfunction due to excessive TCR signaling in the tumor microenvironment.

In the main text, we have now clarified that the “early” initial priming phase refers to the first 24-36 hours of antigen priming, during which distinct T-DC clustering indicative of antigen presentation has been reported by numerous intravital imaging studies (*Stoll et al., 2002: 10.1126/science.1071065; Miller et al., 2002: 10.1126/science.1070051; Bousso et al., 2003: 10.1038/nl928; Mempel et al., 2004: 10.1038/nature02238*).

In contrast, in this study, the tdLNs from the tumor models were examined at 8 days post-tumor induction (**Fig. 1**), and at 4 days post-immunization for the acute OVA protein immunization model (**Extended Data Fig. 2j, k**). Both time points examined have far extended beyond the first 24-36 hours of initial priming as previously reported, and instead represent later time points when antigen-specific T cells have undergone extensive proliferation and differentiation. Thus, we refer to the T-DC clustering and antigen-signaling events observed during these time points as “late” antigen presentation / differentiation phase.

Nonetheless, the reviewer raises a relevant point. To directly address the question of whether starting PD-1 pathway blockade at an early time point (Day 4) before the initial response is fully matured is the explanation for our observation of a loss of high affinity SLAMF6+ T_{SL}, we have followed the reviewer’s suggestion and performed anti-PD-L1/2 blockade starting from Day 10 (“late ICB”), when antigen-specific cells have already robustly expanded during the primary response and their average TCR affinities are higher (**Fig. 3j**). When the tdLN was harvested at Day 19.5, similar to our prior experiments that initiated the blockade at Day 4 and examined the tdLN at Day 12, we again observed a much lower average TCR affinity index in the SLAMF6+ T_{SL} subset from ICB treated mice. These findings have been added to the main text with a new text paragraph (lines 413-435) and in the new **Extended Data Fig. 10a-h**.

A possible complication of this experiment is that the absolute number of SLAMF6+ T_{SL} significantly decreases in the control group (without ICB treatment) between Day 12 and Day 19.5 post-tumor induction (**Fig. R8**). There was a slight but statistically non-significant drop in TCF-1+ T_{SL} count from Day 10 to Day 13 as well (**Fig. 3i**), likely reflecting the increasing loss of antigen presentation. Various mechanisms may account for the fall in antigen

presentation level in the tdLN, including decreased DC migration (*Schenkel et al., 2021: 10.1016/j.immuni.2021.08.026*) and Treg-mediated interference of mregDC migration through CTLA-4 inhibition (*You et al., 2024: 10.1016/j.ccell.2024.06.014*).

This is also consistent with our observation that antigen presentation level falls substantially in late tdLN (Day 12+) as assessed by an *in vivo* antigen presentation assay involving the adoptive transfer of naïve OT-I labeled with proliferation dye into tumor-bearing mice at various time points and examine their proliferation and activation status in the tdLN 72 hours post-transfer (**Fig. R9**).

Figure R8. Comparison of SLAMF6+ T_{SL} numbers on Day 12 and Day 19.5 tdLN post-tumor induction

a, Enumeration of total SLAMF6+ T_{SL} recovered from the tdLNs at Day 12 and Day 19.5, demonstrating the loss of SLAMF6+ T_{SL} in the tdLN at late time points. The checkpoint blockade groups received treatment on Days 4-10 for the Day 12 cohort (early blockade), or treatment on Days 10-16 for the Day 19.5 cohort (late blockade). Note that SLAMF6+ T_{SL} pool at Day 10 is larger than at Day 4 and therefore the late blockade group would have received treatment at a time when it has more SLAMF6+ T_{SL} compared to the early blockade group. The corresponding data in the new **Extended Data Fig. 10a-h** suggest that most of the expanded cells comprising SLAMF6+ T_{SL} subset in the late blockade group were lower affinity clones. Data pooled from 7 independent experiments for Day 12 cohort (n=3-5 per group per experiment), with anti-PD-1 group from 2 independent experiments, and from 3 independent experiments for Day 19.5 cohort (n=3 per group per experiment). Error bars: means ± s.d.

Figure R9. Loss of antigen presentation capacity in the tdLN over time.

a-c, Tumor-bearing mice were adoptively transferred with 5×10^5 naïve CellTrace Violet-labeled OT-I cells at the indicated time points and tdLN cells recovered 72 hours post-transfer to determine the extent of proliferation as well as expression of activation/inhibitory molecules (**a**). **b**, Quantification of total OT-I cells recovered from the tdLN and **c**, mean PD-1 fluorescence intensity of divided (CTV-lo) OT-I cells showing the progressive loss of antigen-driven expansion and activation at late time points. Data from 1 experiment ($n=2-3$ per time point). Error bars: means \pm s.d. * $p < 0.05$, ** $p < 0.01$; one-way ANOVA with Tukey's multiple comparisons.

In contrast to early ICB, under these conditions the number of SLAMF6+ T_{SL} in late ICB treated mice increased at Day 19.5 relative to the control (non-ICB treated) group in a number of mice (new **Extended Data Fig. 10d**). To determine if these cells contained a dominant high affinity cohort as seen prior to ICB, we calculated the mean TCR affinity index of the control (non-ICB treated) group (see **Methods** section for details) and used that measure to gate on SLAMF6+ T_{SL} with an affinity index higher than this mean value (>mean affinity), which we designated as “high affinity” cells. The proportion of “high affinity” SLAMF6+ T_{SL} was strongly reduced in the late ICB treated group (11.5% vs 38% in control group) (new **Extended Data Fig. 10h**), as previously seen with treatment begun on day 4. The absolute count of “high affinity” SLAMF6+ T_{SL} in the late ICB treated mice was also marginally lower compared to the control group, although this difference did not reach statistical significance due to a few outliers (new **Extended Data Fig. 10e**). We note that the disparity in the numbers may be attributed to the progressive loss of total SLAMF6+ T_{SL} over time with weaker antigen presentation described above, as well as the enhanced antigen recognition

capability conferred to a larger starting pool of antigen-specific SLAMF6+ T_{SL} in the tdLN when ICB was initiated at Day 10 as opposed to Day 4, with lower affinity cells likely also able to receive adequate antigen signaling to promote their survival and expansion. Additionally, it should also be noted that SLAMF6- T_{SL} and TCF-1- T_{EFF} expanded by 1-2 orders of magnitude after late ICB treatment (new **Extended Data Fig. 10c**), as would be expected from removal of PD-1 inhibition.

Taken as a whole, the findings with late ICB treatment reinforces our hypothesis that PD-1 controls the access of antigen-specific T_{SL} to cDC1-mediated antigen presentation, and that with the removal of such PD-1 inhibitory signaling, lower affinity cells are able to compete and expand, while higher affinity cells either transition into effectors or show a decrease in survival due to excessive stimulation as indicated by an increase in apoptosis (cleaved caspase-3 staining) (**Extended Data Fig. 8**).

R2.2: The authors propose that T cells with higher affinity TCRs are preferentially recruited into the PD-1^{hi} T_{SL} population (Fig. 3), but the idea seems at odds with the finding that there is a similar distribution of TCF-1+/SLAMF6+, TCF-1+/SLAMF6- and TCF-1-ve populations among OT-I and polyclonal OVA/Kb specific cells (Figs. 1c, 2d). Wouldn't the expectation be that OT-I would align with one group (e.g. PD-1^{hi}, TCF-1+/SLAMF6+ if, as might be expected, OT-I represents a high affinity clone?). There also appears to be a similar decline in "TCR affinity index" among both OT-I and polyclonal OVA/Kb reactive cells when the authors examine SLAMF6+ T_{SL}, SLAMF6- T_{SL} and T_{EFF} cells, respectively (IgG controls, Ext. Fig. 8, panels b versus h). These data certainly do not support the idea of distinct TCR repertoires in those three subsets, since OT-I found in all of them, and differences in apparent TCR affinity occur even with a monoclonal population. This sounds more like "functional avidity" changes, that have been reported with TCR transgenic cells in distinct activation states (e.g., PMID: 11477407).

Even monoclonal TCR transgenic cells such as OT-I can have highly diverse TCR signaling properties (*Feinerman et al., 2008: 10.1126/science.1158013*) and they also give rise to highly diverse responses at a single-cell level (*Achar et al., 2022: 10.1126/science.abl5311*).

Our experimental data show that OT-I T cells expand rapidly in the tdLN. Given that the abundance of antigen displayed by cDC1 is limited, such expansion leads to competition for antigen and only some of the OT-I cells would have access to cDC1 with sufficient antigen presentation to enable retention of the SLAMF6+ T_{SL} state due to rapid upregulation of PD-1. Thus we get heterogeneity even among monoclonal cells, consistent with strength of signal (in this case due to variation in antigen availability rather than intrinsic TCR affinity) dictating

fate in accord with the recently published APL studies and our mixed tumor experiment cited in our response to **R1.5** above. Further generation of heterogeneous outcomes arises from the granular variation in antigen presentation by individual cDC1, such that as noted in discussing competition, some OT-I would not undergo the acute and high rise in PD-1 needed to retain the stem-like state, but rather behave like a lower affinity cell and transition to an effector state (please see **Fig. R6** and our response to **R1.5** above for the mixed tumor experiment that supports this aspect of T cell fate determination). In short, when antigen level is limited, phenotypically SLAMF6+ T_{SL} cells were still generated but at a much lower number, whereas TCF-1- T_{EFF} cells were instead predominantly expanded in the tdLN and in the spleen (**Fig. R6**).

Regarding the decline in affinity index, the reviewer is correct in that the affinity indices of SLAMF6- T_{SL} and TCF-1- T_{EFF} of monoclonal OT-I cells were indeed lower when compared to SLAMF6+ T_{SL}. This is likely due to the fact that SLAMF6+ T_{SL} express higher level of CD8 co-receptor, which enhances tetramer binding (see **Extended Data Fig. 6a**).

Please refer to the updated **Extended Data Fig. 9j**: while SLAMF6- T_{SL} and TCF-1- T_{EFF} OT-I cells (left panel) express lower CD8a level and hence registered lower affinity indices (solid circles), the affinity indices of these subsets remained stable when treated with anti-PD-L1/2 blockade (light shaded circles) despite slight drop in CD8a expression.

On the other hand, in the polyclonal setting (right panel), treatment with anti-PD-L1/2 substantially lowered the affinity indices of SLAMF6+ T_{SL} and SLAMF6- T_{SL}, while TCF-1- T_{EFF} show the lowest affinity indices of all subsets even in the IgG (control) group. Together, these data suggest that while CD8 expression level can influence tetramer binding and hence affinity index measurement, the dramatically lower affinity indices displayed among the polyclonal OVA-specific T cells (especially SLAMF6+ T_{SL}) after checkpoint blockade cannot be explained through this effect alone.

We have added the interpretation of this new analysis (**Extended Data Fig. 9j**) in the main text (lines 402-409). Please also see the response to the next comment about comparing T cells with comparable CD8 expression to alleviate this concern of an 'avidity' effect.

R2.3: To explore the last point further, it will be important for the authors to examine OT-I cells in the same assays as used for polyclonal cells in Fig. 3e-j. If OT-I cells show similar correlations between PD-1 expression level and “affinity index”, the authors need to rethink their hypotheses: one implication of such a result would be that PD-1 blockade does not lead to loss of a subset of Tsl with high affinity TCRs but rather leads to modulation of apparent TCR affinity (and, perhaps, differentiation of such cells), while preserving cells with that TCR in the repertoire.

To address the reviewer’s point regarding examining OT-I cells in the same assays as used for polyclonal cells in **Fig. 3e-j**, we analyzed the data from **Extended Data Fig. 9d-i** (OT-I) and **Fig. 5** (polyclonal). As shown in **Fig. R10**, although there is indeed a slight correlation between PD-1 MFI and affinity index among TCF-1+ T_{SL} OT-I cells, when corrected for CD8 expression by selectively gating on an intermediate band of CD8 expression to account for

Figure R10. CD8 co-receptor influences tetramer binding and affinity indices

To examine relationships between PD-1, SLAMF6, and affinity index for OT-I vs. polyclonal T cells, TCF-1+ T_{SL} populations from tdLN were divided into five separate compartments based on their affinity indices (see **Fig. 3e, g** for gating reference). **a-b**, OT-I T_{SL} showed slight positive correlation between PD-1 staining and affinity index (**a**) but not between SLAMF6 and affinity index (**b**). To correct for the influence of CD8 co-receptor, a defined CD8a-intermediate band was used to gate on TCF-1+ T_{SL} (**c**), which revealed loss of the positive correlation between PD-1 staining and affinity index (**d**) among OT-I cells (data from updated **Extended Data Fig. 9d-i**). In contrast, polyclonal OVA-tetramer+ CD8+ T cells (data from **Fig. 5b-i**) continued to show a positive correlation between PD-1 and SLAMF6 expression with affinity indices even after correcting for CD8 expression level (**d, e**). Error bars: means ± s.d. *p<0.05, **p<0.01, ***p<0.001, ****p<0.0001; one-way ANOVA with Tukey’s multiple comparisons within each group (OT-I or polyclonal).

the influence of this co-receptor on tetramer binding, such correlation disappeared among OT-I cells while polyclonal OVA-specific CD8+ T cells continued to exhibit a positive correlation between PD-1/SLAMF6 MFI and affinity index. These data argue strongly against changes in functional avidity (although certainly occurring in this situation in part through modulation of CD8 expression) negating our conclusions about the affinity relationships between T_{SL} and effectors or the effects of anti-PD-1 treatment.

R2.4: The authors hypothesize that PD-1 blockade causes “terminal differentiation to the effector state or death of the most avid anti-tumor stem-like cells” (lines 37-38, with similar statements throughout). If the first part of this prediction were correct, one might predict that PD-1 (or PD-L1/2) blockade would lead to loss of high affinity index cells in the T_{SL} pool – and appearance of such cells in the Teff pool. In other words, the most avid cells would be “chased” from the stem-like to the effector pool. There appears to be minimal evidence for such a transition in Fig. 5g.

The reviewer’s prediction is indeed captured in **Fig. 5i** when we performed linear modeling to determine the absolute count of “above average” (or “high”) TCR affinity cells in each subset. While not immediately apparent when looking at the mean affinity index displayed in **Fig. 5g**, the exponential increase in cell count following the proliferative bursts of SLAMF6- T_{SL} and TCF-1- T_{EFF} subsets after checkpoint blockade revealed that the number of “high affinity cells” were indeed increased in these subsets while the “high affinity” SLAMF6+ T_{SL} correspondingly decreased (**Fig. 5i**). We have made changes to the text in the revised manuscript to make this result more explicit (lines 379-390).

R2.5: The authors are correct to note that PD-1 (or PD-L1/2) checkpoint blockade leads to reduced frequencies of SLAMF6+ T_{SL} (Fig. 5b) but their own data show that the largest element in this effect is expansion of the SLAMF6- T_{SL} and Teff (Fig. 5c). Numerically, the decline in numbers of SLAMF6+ T_{SL} is about 2-fold (Fig. 5c). While potentially important, it is not clear that the authors can justifiably say that this population has been “lost” – it is more accurate to say that other populations have been expanded. This limits the impact of the studies, since it argues against the idea that a key stem-like population is irretrievably lost by checkpoint blockade – rather the data indicate that this population is (at least transiently) outnumbered.

Our proposed model is that PD-1 checkpoint blockade led to the loss of high affinity SLAMF6+ T_{SL}, and as indicated in the previous response, this is also revealed in the linear modeling plot from **Fig. 5i**, where the reduction in high affinity SLAMF6+ T_{SL} is also

accompanied by a corresponding increase in the numbers of high affinity SLAMF6- T_{SL} and TCF-1- T_{EFF}. As noted above, we have revised the manuscript text to better explain this effect (lines 379-390).

Furthermore, a recent study demonstrated that during acute LCMV infection, PD-1 KO progenitor CD8+ T cells preferentially upregulate exhaustion signatures, thus further suggesting that PD-1 signaling is required to maintain the stem-like state of these activated T cells (*Chu et al., 2025: 10.1038/s41586-024-08451-4*). This information has been added to the **Discussion** section (lines 514-517) to reinforce our proposed model.

R2.6: Developing the previous point, it would be useful for the authors to show what happens when they examine the tdLN many days after the last anti-PD-1 (or anti-PD-L1/2) treatment: Is there, in fact, evidence that the SLAMF6+ T_{sl} population is permanently lost, or do these cells “bounce back” following decay of the blockade antibodies? This reviewer appreciates that the experimental design may be challenging (if the tumor either overgrows or is eliminated, interpretation would be difficult), but this would seem to be critical to test a fundamental element in the novelty and impact of the study. If the SLAMF6+ T_{sl} cells do recover in tdLN (in mice that maintain their tumors), this may suggest elements of the phenotypic changes reported (e.g., reduced TCF-1 or SLAMF6 expression levels – Fig. 5d,e) may in fact be reversible. Such a finding would limit the clinical significance of the report.

We thank the reviewer for raising this important question and have followed the reviewer’s suggestion and extended the experimental endpoint of the checkpoint blockade study to Day 19.5, which is more than 9 days after the final ICB dose administered to the animals. We observed that TCF-1 and SLAMF6 expression of T_{SL} remain significantly downregulated even at this late time point (new **Extended Data Fig. 11a-d**), suggesting that the phenotypic change in stemness markers after ICB does not rapidly recover. Average TCR affinity indices of SLAMF6+ T_{SL} from ICB treated mice remained significantly lower compared to the control group, although the difference between the two groups was substantially smaller than when examined at Day 12 (new **Extended Data Fig. 11h, i**). This difference was not due to lower CD8 co-receptor expression as noted in **R2.3** (see also **Fig. R11**).

The total number of SLAMF6+ T_{SL} from ICB treated mice also remained significantly lower than the control group, and that this difference was even more pronounced when gated on cells with higher than the mean TCR affinity index of the control group (“high affinity” cells as described in **R2.1** and in the **Methods** section) (new **Extended Data Fig. 11f, g**). Of note, SLAMF6- T_{SL} in the tdLN remained moderately elevated at this late time point, while the strong T_{EFF} differentiation seen earlier has largely dissipated (new **Extended Data Fig. 11e**).

Thus, high affinity SLAMF6+ T_{SL} did not replenish in numbers even when given an additional 7 days to recover. We have added a new paragraph to the main text (lines 437-450) to describe this new experiment (**Extended Data Fig. 11**).

Additionally, we note that the diminished number of SLAMF6+ T_{SL} in the tdLN of ICB treated mice may also reflect the loss of antigen presentation after treatment (see our response to the next point in **R2.7** below). It remains to be determined if the residual “higher affinity” SLAMF6+ T_{SL} retains a higher capacity to survive long-term, or if the lower affinity T_{SL} exit the SLAMF6+ stemness state more rapidly than their higher affinity counterparts.

R2.7: A potential consequence of checkpoint blockade would be that, by unleashing improved CD8+ T cell effector functions, tumor antigen specific CD8+ T cells start killing antigen bearing DC. Hence, just like the authors’ studies in which cDC1 were eliminated (Fig. 3c,d), checkpoint blockade would cause (partial) loss of SLAMF6+ T_{SL}, due to abrupt loss of

antigenic restimulation. Have the authors examined this, and if so, how do they control for that interpretation?

The reviewer is correct in that tumor antigen-specific CD8⁺ T cells can likely eliminate antigen-presenting DCs following checkpoint blockade that converts stem-like cells into effector cells.

However, although the loss of antigen presentation by cDC1 is indeed linked to the decreased expansion of SLAMF6⁺ T_{SL}, our data also revealed reduction in the average affinity of the remaining SLAMF6⁺ T_{SL} after checkpoint blockade. As such, this observation cannot be attributed to the loss of cDC1 alone either due to reduced tumor burden or elimination of cDC1 by the expanded effector T cells in the tdLN, as the TCR affinity of the population would be expected to stay similar or even increase due to the competitive setting with limited antigen presentation.

Our preliminary data in using mixed OVA + non-OVA tumor (please see our response in **R1.5** and **Fig. R6** above) showed that low antigen presentation setting did not impede the formation of high affinity SLAMF6⁺ T_{SL}, given that the affinity indices recovered were not substantially lower than the cells from mice injected pure OVA-expressing tumor, although their expansion had been significantly curbed (**Fig. R6**). As such, reduced antigen presentation alone cannot sufficiently explain the loss of high affinity SLAMF6⁺ T_{SL}.

The added data in the new **Extended Data Fig. 10a-h** when checkpoint blockade was initiated at Day 10 (see **R2.1** above), at a point where the higher affinity cells had much better expanded than at Day 4, further suggest that lower affinity cells show disproportionate representation among T_{SL} relative to high affinity cells during checkpoint treatment.

Regarding the elimination of antigen-presenting cDC1 after checkpoint blockade, interestingly, in our model, TCF-1- T_{EFF} cells expressing cytolytic molecules, e.g., Granzyme B (see our response in **R1.3** and **Fig. R4a-d** above) were mostly excluded from the T cell zone and were away from the cDC1 dense region, where the antigen presentation to SLAMF6⁺ T_{SL} occurred (**Fig. 1d**). Only after PD-1 checkpoint treatment was given did we observe the extensive infiltration of TCF-1- T_{EFF} in these cDC1 regions (**Fig. 4d-f**), where it was possible for these cells to directly interact with the antigen-presenting DCs and potentially eliminate them. Indeed, a previous study has noted that a tumor antigen-specific CD8⁺ population in the dLN mostly acquires the T_{SL} phenotype and only differentiates into T_{EFF} when arriving at the tumor site (*Prokhnevskaya et al., 2023: 10.1016/j.immuni.2022.12.002*). We also note that eliminating antigen-presenting cDC1 would not be expected to promote the activation-

induce cell death we observe, as this requires repetitive TCR engagement that would be absent if the cDC1 were eliminated, yet increased apoptosis of SLAMF6+ T_{SL} is what we see after anti-PD-L1/2 checkpoint therapy (**Extended Data Fig. 8**).

As such, the spatial and temporal segregation of T_{EFF} from cDC1 may reflect an evolved strategy to prevent the killing of antigen-presenting DCs by effector cells such that antigen-driven expansion of T_{SL} can be perpetuated, whereas disruption of PD-1 signaling axis during checkpoint therapy not only drives the terminal differentiation of T_{SL} but also brings the effectors into close contact with DCs, which then forms a positive feedback loop that further reduces T_{SL} expansion.

We agree this issue is a potentially relevant limitation of our study. For space limitation reasons, we have not added a discussion of this complex issue into the main text but we can certainly add some comments to the **Discussion** section if the reviewer deems it important to do so.

Reviewer 3

Congratulations on a very compelling manuscript and highly relevant findings.

In the manuscript titled “PD-1 controls differentiation, survival, and TCR affinity evolution of stem-like CD8+ T cells” by Jyh Liang Hor et al. the investigators apply 3D multiplex immunofluorescence imaging to study stem-like CD8+ T cells in the spatial ecosystem of the tumor draining lymph node.

Summary of the key results:

The manuscript describes multiple novel and paradigm shifting findings that generally justify a publication in “Nature”. 1st they demonstrate that in contrary to the prevailing view, continues TCR signaling by high affinity TCRs does not necessarily lead to terminal TEFF differentiation and exhaustion. 2nd they elucidate, at least partly, the mechanisms by which dDC1 in the tdLN counterbalance TCR signaling by providing PD1-stimulating signals. 3rd they describe a novel mechanism of high affinity TCR T cell selection, in which prolonged antigen presentation and PD1-stimulating signals (PD-L1/2 expression) by dDC1 cells in the tdLN promote the survival and expansion of high affinity TCR T cell clones. 4th they show that PD1/PD-L1/2 blockade interferes with this process resulting in progressive loss of high affinity TCR T cell. These findings have major implications on our understanding of high affinity TCR T cell selection and might influence our clinical application of PD1/PD-L1/2 blocking antibodies.

Originality and significance: The findings are of highest originality and significance.

Overall, the manuscript is written very well. That data presentation is adequate and supports the message. Statistics are used appropriately.

We are very grateful that the reviewer has found our study to be compelling and highly relevant to the field. We have incorporated many of the suggestions of the reviewer in the revised manuscript together with detailed responses here to the other points raised.

There are some remaining questions and there might be space for improvement at certain point, I would like to discuss point-by-point in the following:

R3.1: Line 90: Could you please provide data or references supporting the statement that day8 represents the “peak expansion and differentiation stage”.

We apologize for the confusion and have addressed this point in the revised text to clarify that Day 8 represents a “late” expansion and differentiation stage following initial robust proliferation and functional diversification, to better represent the context (lines 91-95).

R3.2: Figure 1: It is stated that all data shown are representative of at least 3 independent experiments, with n=2-3 per experiment. Could you please provide quantification of T-DC clusters etc. from all animals with arrow bars to demonstrate consistency of findings.

We agree that quantification of the imaging data is important to document the consistency of the experimental findings. We have now included quantitative image analysis of the OT-I subsets in dense cDC1 region for a total of 6 samples from Day 8 tdLN pooled from 3 independent experiments. This analysis has been added to the updated figure as **Fig. 1g**.

Additionally, we have improved our image analysis pipeline for the segmentation of T cells based on CD45.1 congenic membrane markers, and this updated workflow has been incorporated into the **Supplementary Methods** section and the code is now available in the Github repository associated with this manuscript.

R3.3: Figure 1g: Quantification of nuclear NFAT positive/negative OT-1 cells in correlation to distance tDCs and cluster distant areas of the tdLN distance would help to support the statement.

We have also performed a quantitative analysis of nuclear NFAT localization on OT-I cells and their distance to cDC1 on a total of 5 tdLN samples pooled from 2 independent experiments, which demonstrated that OT-I with positive nuclear:membrane NFAT ratio were significantly associated with regions of high cDC1 density in the tdLN. The analyzed data are now shown in the updated **Fig. 1i and Extended Data Fig. 2f-i**. The workflow for quantifying NFAT signal has also been added to the **Supplementary Methods** section and in the associated Github repository.

R3.4: Line 271-273: It is stated the high affinity TSL (most likely) give rise to high affinity TEFF based on the finding that the TCR affinity of TEFF lacks behind. This is rather weak data to support this mechanistically very important finding. Since the OT-1 input is polyclonal, it should be possible to trace T cell clones by TCR sequencing. At this stage and later, looking at the effects of PD1/PD-L1/2 blockade, the data set would very much profit from an experiment tracing the faith of the polyclonal T cell pool. Like in the experiment resulting in Figure 3 i-j, one could utilize T cells (TSL and TEFF) from the tdLN at different timepoint for TCR sequencing to understand the clonality of the repertoire. One could additionally

sequence T cells from peripheral blood and tumor to understand distribution. Ideally one would perform scRNASeq including VDJ-Seq from all cells of the tdLN, blood, tumor. This would additionally generate data to support the mechanism of TEFF promotion within the T-DC clusters. Like this one could follow how specific TSL clones transform into TEFF as well as their distribution tdLN vs. peripheral blood vs. tumor.

We are not quite sure what the reviewer means in referring to “OT-I input is polyclonal” since OT-I cells express a monoclonal transgenic TCR. To clarify, all experiments shown from **Fig. 2-5**, unless otherwise stated, were performed under polyclonal setting without the adoptive transfer of monoclonal OT-I cells.

Regarding scRNA seq + TCR seq, we note that a number of studies have already reported the clonal overlap between tumor draining lymph node, blood, and tumor after checkpoint blockade (e.g., *Huang et al., 2017: 10.1038/nature22079*, *Wu et al., 2020: 10.1038/s41586-020-2056-8*), as well as a recent report by *Pai et al. (2023: 10.1016/j.ccell.2023.03.009)* that performed longitudinal tracking on NSCLC patients undergoing anti-PD-1 treatment. As stated above, it seems beyond the scope of this paper to conduct scRNAseq studies on the effector population given both the existing literature on this issue and our affinity evidence that the highly avid clones are ‘chased’ into the effector population.

R3.5: Line 304-305: As outlined above, scRNASeq combined with VDJ-Seq at different timepoints could provide significant additional information on cell-cell interaction and mechanistic insights. One could perform velocity analyses tracing the transformation of cell states. How is PD-L1 and PD-L2 expression on DCs regulated? What is inducing the T-DC clusters? Do T cell derived signal, e.g. INF γ , induce PD-L1 and PD-L2 expression? Are other co-stimulatory / inhibitory signals involved in the regulation of the stem-like state?

Such sequencing experiments would not be capable of relating the specific state of the cDC1 (XCR1+) DCs that are doing the relevant antigen presentation to the generation and stability of TCF1+SLAMF6+ cells, as the relevant DC would be only a subset of the sequenced cells and currently we have no means of relating isolated dissociated single cells to the DC in the clusters beyond what we have done by direct imaging. In the latter case, the expression of proteins engaged in ligand-receptor interactions have key epitopes obscured by these interactions and many molecules of importance undergo either internalization or trogocytosis, making measurements of expression extremely complex.

Our work and that of others using intravital imaging has shown the formation of T-DC clusters and the critical roles of antigen presentation-TCR engagement along with chemokine guidance (*Castellino et al., 2006: 10.1038/nature04651*) in fostering a ‘stop-signal’ that

keeps the T cells from migrating away from the APC, in part due to integrin affinity enhancement, so we do not think this is an aspect of our work requiring new detailed study. Similarly, there is ample evidence that IFN γ increases PD-1 ligand expression on APC. Full dissection of all costimulatory and inhibitory signals involved in fostering the stem-like state would unfortunately require more than another full manuscript's worth of studies to unravel.

R3.6: Line 352: It is stated that the effects of PD-1 blockade on the TSL clonal repertoire is examined. However, this is only supported by changes in the “affinity index”. Again, the manuscript would strongly benefit for TCR sequencing data. One would expect an oligomerization of clonal diversity due to selection of high affinity clones, which is screwed by PD1/PD-L1/2 blockade.

We agree that scRNA + TCR sequencing will allow examination of clonal repertoire and changes in such repertoires by checkpoint blockade therapy. However, as during the short period of our experiments there is no gross change in the naïve peripheral T cell repertoire, the shift of high affinity clones from the stem-like population to the effector population shows that we are 'chasing' cells of given clonality between the two populations. To directly assess affinity within any clonal populations detected by TCR sequencing at the single cell level, we would need to clone these TCR chains and express them in T cells blanked for endogenous TCR expression to avoid mixed pairing – this is a very expensive and time consuming study to undertake and we do not believe that such experiments would add substantially to the interpretation of the data we present.

We also note that, as already included in the **Discussion** section, “clonal replacement” has been reported in a number of publications examining clinical data from studies involving patients who underwent PD-1 checkpoint therapy (lines 527-529). Substantial changes in the clonal repertoire have been reported, with new emergent clones as well as extinction of existing clones. However, at present there is no evidence that these emergent clones as reported in these publications are tumor antigen-specific.

Discussion:

R3.7: Line 466-469: Please elaborate on how a partial blockade of the PD-1 pathway could look like in reality and how that would fit with the proposed mechanism. Is selection of high affinity T cells in patients a timely limited process or consciously ongoing and how would you time “partial blockade”. Is there any evidence from patients that TCR affinity decreases

during checkpoint blockade, supporting the hypothesis that high affinity TCR T cells experience AICD?

A potential approach would involve a similar concept from “adaptive therapy” used in chemotherapeutic setting for the treatment of prostate cancer (*Zhang et al., 2017: 10.1038/s41467-017-01968-5; Gallaher et al., 2023: 10.1158/0008-5472.CAN-22-2558*), in which the drug dosage is adjusted based on radiological findings and the impact of previous dose on tumor size. Here, the goal is to control tumor growth rather than to eliminate the tumor entirely with strong dose, as doing so would likely select for drug-resistant tumor clones that would be out-competed in the presence of drug-sensitive clones.

We propose that similar strategy could be explored as an immunotherapeutic approach where the dosage is titrated based on the response on tumor growth/shrinkage to minimize the dose of anti-PD-1 administered. In other words, the goal is to drive a sufficiently robust effector response through PD-1 blockade but not at an excessive dosage that it starts culling the high affinity precursors.

As far as we are aware, there are no clinical data that have directly examined TCR affinity during checkpoint blockade. This would require identification of individual patients’ neoantigen epitopes and conducting quantitative antigen presentation assays. We have not been able to find studies that examine the TCR affinities of patients’ tumor antigen-specific T cell clones, although such work could be relevant to examining the extent to which our findings correspond to those in patients treated with ICB.

R3.8: Please discuss the implications of your findings on cancer vaccine approaches. Multiple studies have demonstrated induction of mutation specific T cell by either peptide or mRNA neoepitope vaccines. As of today, it is unclear, whether vaccines should be combined with PD1/PD-L1/2 blockade and in which sequence. The presented data would argue to hold checkpoint blocked during the vaccination phase to allow high affinity T cells to be induced and selected.

Regarding cancer vaccine approaches, neoepitope vaccines have shown increase in T_{SL} formation (e.g., *Keshari et al., 2024: 10.1016/j.celrep.2024.114875*), which is consistent with the notion of antigen-driven T_{SL} expansion as described in the present manuscript. mRNA neoantigen vaccine has also been paired with anti-PD-L1 (*Rojas et al., 2023: 10.1038/s41586-023-06063-y*) although the changes in stem-like compartment and TCR affinity clones remain unexplored.

Response to the Reviewers' Comments

Reviewer 1

The revised manuscript is improved and more thorough. The authors addressed many of my concerns with thoughtful explanations and additional data. I especially appreciate the new analysis of T cell subsets. However, I remain unconvinced by their approach to assessing changes in TCR affinity between TSL and Teff cells. As noted previously, the tetramer binding assay used does not directly measure affinity—it measures avidity—yet the manuscript continues to use the term "affinity" broadly and inaccurately. This was also noted as a point of concern by Rev. 2.

Their analysis is simply based on averaging tetramer MFI normalized to CD3 levels, and while this is more rigorous to account for total surface TCR levels, it still remains a measure of avidity, not affinity. Given that changes in "affinity" are central to the authors' proposed model for how checkpoint blockade affects T cells, this issue must be addressed more rigorously.

That the authors see similar subsets in OT-1 T cells that have the exact same affinity TCR again, speaks to the point that the authors are observing changes in TCR avidity in the different subsets of cells, not affinity.

Thus, at a bare minimum, the authors need to revise the terminology to avidity when this is discussed to ensure a more accurate description of the assays and results.

The authors go through more rebuttal on this point for Reviewer 2 and then introduce CD8 levels and how that can affect TCR affinity. They try to normalize the tetramer staining to CD8 levels and while this Reviewer appreciates and can see why they are examining this third factor in TCR avidity (not affinity), the fact that the authors are normalizing against two variables makes one feel even less confident.

We thank the reviewer for the thorough assessment of the data presented in our manuscript, and we apologize for not having provided both complete justification for our use of the terminology "TCR affinity" and all the associated experimental data we have developed in support of this usage.

The reviewer is absolutely correct that measuring tetramer binding is an estimation of TCR *avidity* (based on the multivalent binding strength of the TCR-pMHC complex), which in itself is a function of TCR affinity (monovalent TCR-pMHC binding), the amount/density of expressed TCRs, as well as the presence of coreceptors (e.g., CD8) and other molecules that can potentially influence binding capacity in a solution-based experiment of this type.

The “Affinity Index” that we have derived by normalizing tetramer MFI to CD3 expression is thus a proxy measurement of the TCR affinity itself and not direct measurement of the ligand binding affinity. However, there is extensive support in the literature for using the term ‘affinity’ as derived from tetramer binding experiments under the appropriate conditions; this has been reported in studies that compare the results from tetramer binding / washout experiments to direct measurements of *affinity* using surface plasmon resonance (*Crawford et al., 1998: 10.1016/S1074-7613(00)80572-5; Krogsgaard et al., 2003: 10.1016/S1097-2765(03)00474-X; Laugel et al., 2007: 10.1074/jbc.M700976200; Zhong et al., 2013: 10.1073/pnas.1221609110*). Indeed, an early paper from the Davis lab, where tetramers were first developed, explicitly uses tetramer dissociation to impute what they call ‘affinity’ – see *Savage et al., 1999: 10.1016/S1074-7613(00)80048-5*.

This said, we think that the reviewer has made an important point and in the interest of greater precision of terminology, we suggest changing our terminology of “Affinity” or “Affinity Index” to “Imputed Affinity” or “Imputed Affinity Index” so that the reader is aware that we have not measured TCR affinity directly at the molecular level. We have also expanded the section *Imputed Affinity Index* in the **Methods** of our manuscript (**lines 831-846**), clarifying that our approach to *impute* affinity “is possible because tetramers elute from labeled T cells in the wash buffer during incubation steps in reasonable proportion of affinity, in particular the off-rate of the interaction. When normalized to the surface TCR expression to account for *avidity* differences among T cells, we can use the measured staining as a proxy for this off-rate and hence, *affinity*. We also further controlled “for secondary factors affecting this off rate such as CD8 co-receptor binding through direct comparison with monoclonal OT-I transgenic cells” and demonstrated that downregulation of CD8 co-receptor cannot account for the shift in the imputed affinity measured during checkpoint blockade when using this approach. When we introduce the term “Imputed Affinity” in the revised manuscript, we have referred the audience to this **Methods** section for a detailed explanation and rationale of the approach. We have also included additional references as cited above to better place our use of this assay in the context of TCR-ligand interaction studies already in the literature and have added the material described below for those interested in more details.

As to why we wish to maintain the use of the term *affinity* in our manuscript, it is clear that factors affecting avidity are changeable and vary temporally in individual T cells, affecting the ‘instantaneous’ capacity for signaling in response to a given display of pMHC ligand. However, affinity does not change in this manner, and among T cells with a physiological range of TCR expression and CD8 density, it is TCR affinity (or catch bond formation capacity affecting off-rate) that sets a limit for sensitivity to antigen. This is of particular relevance in the tumor setting, as T cells with lower affinity will be subject to failure of effector function

in the face of partial loss of pMHC density on tumor targets, for example, heterozygous loss of $\beta 2m$, of genes encoding components of the peptide transport and loading machinery, or partial downregulation of the source protein for bound peptides. Thus, we believe it is important to emphasize this ‘uncorrectable’ loss of T cell reactivity with checkpoint blockade therapy and believe our data support the use of the ‘imputed affinity’ terminology. We also note that recent studies using weak altered peptide ligands that directly change affinity, not avidity, of responding T cells, leads to loss of T_{SL} and favors effector generation, in full accord with our conclusions (*Lan et al., 2024: 10.1038/s41590-024-01843-8*).

Beyond these conceptual issues, we have also more directly addressed the reviewer’s concerns by performing additional analyses to dissect how the key components influencing TCR avidity and tetramer binding factor into our Imputed Affinity Index measurement.

First, we further divided antigen-specific TCF-1+ T_{SL} isolated from Day 12 tdLN into graded sub-populations based on their CD3 expression levels (**Fig. R12A, E**). Monoclonal OT-I transgenic T cells as examined in **Extended Data Fig. 9d-j** were used as controls where TCR affinity remains constant.

Figure R12. Tetramer binding changes after PD-1 checkpoint blockade cannot be explained by altered TCR expression. Antigen-specific TCF-1+ T_{SL} was divided into graded subpopulations based on CD3 (TCR) levels. Representative flow cytometry plots for monoclonal OT-I (**A**) and polyclonal OVA-specific CD8+ T_{SL} (**E**) isolated from Day 12 tdLN. TCR avidity plots showing the relationship between tetramer MFI and CD3 MFI for individual samples from both control IgG-treated and anti-PD-L12 treated groups were shown in (**B**) for OT-I and (**F**) for polyclonal OVA-specific T cells. Imputed Affinity Index plots after normalization of tetramer MFI to CD3 levels were shown in (**C**) for OT-I and (**G**) for polyclonal OVA-specific T cells. The mean Imputed Affinity Indices were further normalized to the IgG-treated (control) group for each experiment to control for batch effects and are shown in (**D**) for OT-I and (**H**) for polyclonal OVA-specific T cells. Data pooled from 2 independent experiments (n=9 from each group) for OT-I and from 5 independent experiments (n=15 from each group) for polyclonal OVA-specific T cells. * $p < 0.05$, **** $p < 0.0001$; unpaired two-tailed t-test.

As can be seen in **Fig. R12B**, the tetramer MFI on OT-I cells is linearly and positively correlated with CD3 levels. This shows that absolute tetramer staining intensity is indeed sensitive to the total amount of TCRs expressed by the antigen-specific T cells as an avidity model predicts. With the exception of a few outlier animals, the tetramer MFI in relationship to CD3 expression was nearly identical for OT-I cells from control IgG treated and anti-PD-L1/2 treated groups. This suggests that anti-PD-L1/2 treatment did not alter the tetramer binding capacity of the monoclonal TCR-expressing OT-I cells *at a given CD3 expression level*.

When normalized to CD3 expression (our “Imputed Affinity Index”) (**Fig. R12C**), the positive relationship shown for tetramer binding vs. CD3 staining disappears. To further control for experimental variations in tetramer binding, we further normalized the mean Imputed Affinity Index to the mean indices of TCF-1+ T_{SL} from IgG-treated (control) group pooled from the same experimental cohort, as we did for most data figures from the manuscript involving quantifying the Imputed Affinity Indices, and found a nearly identical trend (**Fig. R12D**, see “Affinity Index” under **Methods** section for details). There is a mild negative correlation where the highest CD3 expressing cells also registered the lowest Imputed Affinity Index, and conversely, the lowest CD3 expressing cells typically marked the higher range of the Imputed Affinity Index. This is likely caused by non-linear effects of TCR expression on pMHC binding, for example, diminishing returns of the density of TCR molecules in their ability to enhance clustering with pMHC. This also does not preclude other biophysical properties that could affect binding.

What is critical to note is that these two subpopulations at both ends of the CD3 expression range comprise only a very small fraction of the total OT-I T_{SL} and the majority of the cells were situated in between the two sub-compartments (as is also the case for the polyclonal T cells we examine using this assay). The CD3-graded OT-I subpopulations registered very similar Imputed Affinity Indices for control and anti-PD-L1/2 treated groups, as expected for a monoclonal population where our index has corrected for avidity and revealed intrinsic properties of the TCR in terms of ligand binding.

However, the picture is drastically different in the polyclonal setting. Despite also showing a positive correlation between tetramer MFI and CD3 expression, across all CD3 graded segments, polyclonal OVA-specific CD8+ T cells from anti-PD-L1/2 treated group displayed lower tetramer MFI compared to their counterparts in the control group, indicating lower TCR avidity (**Fig. R12F**). When converted into our Imputed Affinity Index measurement by normalizing to CD3 level, we once again observed the Imputed Affinity Indices of the TCF-1+ T_{SL} from anti-PD-L1/2 treated group to be significantly lower than those from the control IgG

group (**Fig. R12G, H**). The inverted relationship between Imputed Affinity Indices and CD3 levels further argue against the hypothesis that the changes in Indices after checkpoint blockade were simply due to TCR downregulation as a result of enhanced exposure to antigen. Parts of **Fig. R12** have been added to the manuscript as **Extended Data Fig. 9k, l** and a summarized description of the above paragraphs in the main text in **lines 371-376**.

Finally, we note that the overall CD3 expression on OT-I transgenic cells is lower than those found in the polyclonal cells, and this lower CD3 expression is also observed on naïve OT-I cells (data not shown). This is likely due to the transgenic induction of the cloned TCRs in RAG knockout cells. Nonetheless, the observed correlation between tetramer binding and CD3 level continues to hold and agrees with the measurement of TCR avidity as defined in the literature.

Altogether, this set of data shows that TCR expression level influences tetramer binding but also establishes that changes in TCR expression-related avidity effects alone cannot explain the reduced Imputed Affinity Indices observed among polyclonal T cells after checkpoint blockade treatment.

Another component integral to TCR avidity is CD8 coreceptor expression, which was already referenced in the main text (*Laugel et al., 2007: 10.1074/jbc.M700976200*) and raised by the reviewer(s) as the main confounding issues with respect to affinity vs. avidity measurements using tetramers. Given that SLAMF6+ T_{SL} show higher CD8 coreceptor expression than other subsets and as such are liable to show greater tetramer binding even when the latter is adjusted for TCR expression, we have previously addressed this in **Extended Data Fig. 9i-j** as well as **Reviewer Figure R10**. In particular, we show that while higher CD8 expression does lead to a higher Imputed Affinity Index assigned to these cells, and that CD8 expression is downregulated upon PD-1 checkpoint blockade, the Imputed Affinity Index of monoclonal OT-I transgenic SLAMF6+ T_{SL} remained unchanged whereas polyclonal OVA-specific SLAMF6+ T_{SL} still demonstrate a reduction in these affinity indices (**Extended Data Fig. 9i-j**, see below). Thus, CD8 downregulation still cannot account for the drop/shift in the Imputed Affinity Index after checkpoint blockade.

While adhesion molecules e.g. LFA-1/ICAM-1 during cell-cell interaction have been proposed to stabilize the formation of immunological synapse and enhance TCR clustering with pMHC (*Dustin, 2014: 10.1158/2326-6066.CIR-14-0161*), this component is largely irrelevant in the pMHC tetramer binding assay performed on single cell suspension that does not involve an adjacent cell partner to measure TCR binding.

Taking into consideration all these factors, and based on the literature definition of TCR avidity being a function of TCR affinity, TCR expression amount/density, coreceptor and adhesion molecule expressions, we find it most reasonable that the Imputed Affinity Index changes that followed PD-1 checkpoint blockade were primarily driven by an altered clonal landscape with a clear shift in the TCR affinity at a population level, and not simply a reduced capacity of clonal TCRs to bind to pMHC complexes due to changes other than such clonal TCR affinity. We hope the referee agrees and finds that our new terminology, extended discussion of how we derive the Imputed Affinity Index, added data, and additional references provide an adequate response to this important issue.

In my original review, I suggested the inclusion of TCR sequencing (e.g., scTCR-seq) to assess clonality across TSL and Teff subsets. This remains critical. Without evidence of clonal overlap or distinction, it is not possible to determine whether observed differences reflect changes in true affinity or simply changes in clonal TCR composition and avidity. If the same clones are present in both subsets, then the observed differences are clearly due to avidity. If distinct clones dominate each subset, and this is enhanced by ICB, then a shift in affinity

may be inferred. That they authors interpret the increased in affinity over time is due to the selection of higher affinity clones would be directly observed in a scTCR-seq analysis too. It is fine if technical barriers prevent the tetramer staining comparison across tdLNs and tumors, but this should not be true for scTCR seq as this is done routinely with small numbers of cells and within tumors and tdLNs. The technical barriers to performing TCR repertoire analysis are relatively low, and not incredibly costly, thus, this reviewer feels this data is really important to support the authors' conclusions.

We appreciate the reviewer's suggestion to perform TCR sequencing to further investigate the shared clonality between stem-like and effector cells as an approach to understand if affinity or avidity is what we are tracking in our experiments. Unfortunately, this sequencing approach will not address this affinity/avidity issue and doing such studies would entail much more complex experiments than proposed by the reviewer. Our data indicate that PD-1 blockade promotes the increased competition/proliferation of lower affinity clones while the higher affinity clones are either driven into hyperactivation-induced cell death or forcibly differentiated into effector cells, all of which follows from a model involving a linear differentiation trajectory from the most stem-like SLAMF6+ T_{SL}, to the intermediate SLAMF6-T_{SL} and finally into the TCF-1- T_{EFF}. This pathway is supported by multiple studies in the literature including some using TCR sequencing to show clonal relationships of the type the reviewer has asked us to generate (e.g., *Rahim et al., 2023: 10.1016/j.cell.2023.02.021*; *Prokhnevskaya et al., 2023: 10.1016/j.immuni.2022.12.002*). Schenkel et al. (2021: *10.1016/j.immuni.2021.08.026*) have previously reported on the shared clonality between SLAMF6+ and SLAMF6- stem-like cells, suggesting that SLAMF6+ T_{SL} clones were not segregated from the rest of the antigen-specific populations. While sequencing studies can reveal clonal relationships among T cells of distinct phenotype in tdLN and in tumors, such TCR sequencing studies do not provide data able to distinguish between changes in avidity vs. affinity after checkpoint treatment. The loss of high affinity clones accompanied by an expansion of lower affinity clones in the T_{SL} compartment would also lead to the observation of shared clonotypes in both the T_{SL} and T_{EFF} compartments, simply due to the fact that T_{SL} are precursors that give rise to T_{EFF}. Our preliminary data from **Reviewer Fig. R6** provided during the previous revision, where mixed OVA-expressing and non-OVA expressing tumors were used to limit the amount of presented antigen, further demonstrated that the expansion of SLAMF6+ T_{SL} was directly controlled by antigen availability. This set of observations is consistent with data from P14 transgenic CD8+ T cells during chronic LCMV infection, where a high level of LCMV antigen is directly correlated with higher amount of TCF-1+ progenitor exhausted cells (*Utzschneider et al., 2020: 10.1038/s41590-020-0760-z*).

Because TCR sequences themselves do not provide direct information on ligand binding affinity, the sequencing analyses suggested are by themselves unable to address the

specific affinity/avidity issue raised by the reviewer. Furthermore, with respect to experimental practicality, the TCR repertoire from each individual animal is distinct and can only be tracked if it is possible to sample SLAMF6+ T_{SL} directly from the blood over time. From our data, SLAMF6+ T_{SL} are mostly confined to the tdLN making such an analysis undoable. At best, we would need to do paired LN/tumor TCR sequencing for multiple animals in control and treatment groups across time, and then deconvolve all the data, which in the end would only show clonal relationships, not affinity changes. This seems an inordinate amount of cost and effort for little gain in terms of the concern that is the focus of the referee.

The reviewer's question could be addressed by performing single cell TCR sequencing on tetramer-bound cells recovered from the tdLN of both control and checkpoint treated mice, followed by either expression and purification of many TCR for plasmon resonance measurements, use in optical tweezer studies, or by cloning and expression in lymphocyte cell lines with their endogenous TCR genes knocked out, followed by dose response analyses. However, performing any of these types of studies on dozens or more of unique TCR sequences is technically challenging and properly performed in only a few labs who would not be able to undertake such studies in collaboration with us in a timely manner. Indeed, such work typically encompasses several years of effort by even the most competent groups working in this area. We agree that such studies would be valuable in the future to more precisely define how checkpoint blockade therapy shifts both the phenotypic and the affinity landscape of the tumor adaptive immune response.

Reviewer 2

The authors have done an excellent job of extending their studies further and resolving what appeared to be contradictory findings. There is still a concern, however, about the overall framing of this work as an investigation into the CD8+ T cells which constitute a reservoir of anti-tumor cells in the draining lymph nodes (dLNs) (related to reviewer comment 2.1).

While it was valuable that the authors clarify their meaning of "late" (i.e. the "initial late differentiation phase" after priming), this does not address a central concern with overall interpretation and context of these studies.

The authors write that "Emerging evidence suggests that tumor-draining lymph nodes (tdLNs) serve as reservoirs from which freshly generated TSL-like cells or their immediate progeny continuously replenish the tumor microenvironment (TME) and provide a source of

functionally competent effector cells during checkpoint therapy [refs] 3-6.” (lines 62-65) and go to say that investigating the characteristics of cells in the reservoir is a central focus of this study (e.g., lines 30-33, 67-73). Yet, as they acknowledge in the rebuttal, their work “does not represent a much later period during which T cells are expected to experience exhaustion/dysfunction due to excessive TCR signaling in the tumor microenvironment.” But that IS the central point of studies such as those in references 3 and 4 – which indicated that dLNs were a key reservoir of cells that entered the tumor long after induction of local exhaustion (in those studies, months after tumor induction). While this reviewer appreciates that different tumor models will have different kinetics, the current studies investigate cells involved in the initial response to tumor – it is not clear that any of these findings relate to T cells that contribute to the long-term establishment of a reservoir of tumor-specific CD8+ T cells in the dLN.

That does not mean the work is without considerable merit – and, again, the other responses by the authors resolved key concerns – but the text needs to be revised to reflect the fact that the early time points studied preclude simple extrapolation from the data shown to the characteristics of the durable reservoir of anti-tumor T cells maintained in draining LNs. Casting this study as an exploration of the properties of the anti-tumor reservoir is, at a minimum, premature.

We appreciate the reviewer’s comment and agree that our previous framing of “late” time points may be potentially misleading since the studies performed in the manuscript did not examine the terminal T cell exhaustion phase that typically occurs at a much later time point.

We have modified the relevant paragraph in the main text in **lines 85-89** to more accurately describe the use of “late” phase in our manuscript:

Naïve antigen-specific T cells undergo intense antigen-driven proliferation and functional diversification after the first 24-36 hours of initial priming. The Day 8 tdLN thus represents a substantially “late” phase of expansion and differentiation distinct from both the initial T cell activation or the terminal exhaustion phase that occurs many days/weeks later in tumor-bearing or chronically-infected hosts.

Nonetheless, we would like to point out that in reference 4 (Schenkel et al., 2021: 10.1016/j.immuni.2021.08.026), the decline in SLAMF6+ stem-like T cells was readily observed from 8 weeks onward in the KP lung adenocarcinoma GEMM model used in that study. The kinetics of SLAMF6+ stem-like cell expansion followed closely the migration of XCR1+ cDC1 to the tdLN, which peaked at 5 weeks in the model used by Schenkel et al. and had begun to diminish in numbers (both cDC1 recruitment as well as Ki-67+ proliferating

SLAMF6+ stem-like cells) by 8 weeks – indicating that the persistence of SLAMF6+ stem-like cells is strongly correlated to the availability of pMHC antigen presentation at the tdLN.

Our intradermal tumor transplant model follows a similar kinetics but at a much more accelerated pace – with SLAMF6+ T_{SL} peaking at 7-10 days and their numbers began to decline at ~14-15 days, as indicated in the response in **R2.1** in our previous revision letter where we extensively discussed the observation.

As such, our findings are not inconsistent with the published data in the literature and we have included in the main text the language noted above to clarify that our data do not represent the “late terminal exhaustion” time points where SLAMF6+ T_{SL} numbers may be substantially diminished in the context of reduced tumor antigen presentation in the tdLN.

Reviewer 3

To the authors of the manuscript: “PD-1 controls differentiation, survival, and TCR affinity evolution of stem-like CD8+ T cells”

Thank you for sufficiently addressing and clarifying all raised points. I do understand that large scale scRNASeq/scTCRSeq experiments, though most likely mechanistically informative, are beyond the scope of this manuscript. I do not have additional point to be addressed and would support the publication of the revised version of the manuscript.

We are grateful for the reviewer’s comments and are glad to hear that the reviewer finds our revised manuscript acceptable for publication.